# Awake ripples enhance emotional memory encoding in the human brain

Haoxin Zhang [1,2,12] ✉, Ivan Skelin[3,4,12], Shiting Ma[1], Michelle Paff[5], Lilit Mnatsakanyan[1], Michael A. Yassa[1,6,7], Robert T. Knight[8,9] & Jack J. Lin [10,11] ✉

Enhanced memory for emotional experiences is hypothesized to depend on amygdala-hippocampal interactions during memory consolidation. Here we show using intracranial recordings from the human amygdala and the hippocampus during an emotional memory encoding and discrimination task increased awake ripples after encoding of emotional, compared to neutrally-valenced stimuli. Further, post-encoding ripple-locked stimulus similarity is predictive of later memory discrimination. Ripple-locked stimulus similarity appears earlier in the amygdala than in hippocampus and mutual information analysis confirms amygdala influence on hippocampal activity. Finally, the joint ripple-locked stimulus similarity in the amygdala and hippocampus is predictive of correct memory discrimination. These findings provide electrophysiological evidence that post-encoding ripples enhance memory for emotional events.

Multiple mechanisms have been proposed to explain the prioritized encoding of emotional experiences[1–3], including the neuromodulatory effects on plasticity and the interplay between the amygdala and the hippocampus[1,4–6]. Several studies have found memory reinstatement during the immediate post-encoding period to be predictive of later memory performance[7,8]. Ripples are transient hippocampal oscillations (80–150 Hz), associated with synchronous neural activation in the hippocampus and the amygdala[9,10], and are implicated in the binding of anatomically distributed memory traces[11]. Behaviorally relevant reactivation of emotional memory occurs during ripples[12], and disruption of post-experience ripples interferes with memory utilization[13]. Based on these findings, we hypothesized that ripples occurring immediately after stimulus encoding (post-encoding) facilitate emotional memory discrimination through coordinated hippocampal-amygdala memory reinstatement or by facilitating the retention of stimulus in working memory. Furthermore, we

hypothesize that either of these processes would result in increased stimulus similarity during post-encoding ripples. Using intracranial electroencephalographic (iEEG) recordings in epilepsy patients during the performance of an emotional encoding and discrimination task, we first confirm behavioral reports of better discrimination memory for arousing stimuli[3]. Next, we demonstrate that the number of ripple events immediately after encoding is associated with both stimulus-induced arousal and the accuracy of later discrimination. Finally, the coordinated post-encoding stimulus similarity across the amygdala and the hippocampus during post-encoding ripples is predictive of later memory discrimination performance, with the amygdala stimulus similarity showing a directional influence on the stimulus similarity in hippocampus. Together, these findings provide evidence that ripples-mediated dynamics in the amygdala and hippocampus provide a mechanism accounting for better remembering of emotional experiences.

[1]Department of Neurology, University of California Irvine, Irvine 92603 CA, USA. [2]Department of Biomedical Engineering, University of California Irvine, Irvine 92603 CA, USA. [3]Krembil Brain Institute, Toronto Western Hospital, Toronto, Ontario M5T 1M8, Canada. [4]Department Center for Advancing Neurotechnological Innovation to Application, Toronto, Ontario M5G 2A2, Canada. [5]Department of Neurosurgery, University of California Irvine, Irvine 92603 CA, USA. [6]Department of Neurobiology and Behavior, University of California Irvine, Irvine 92697 CA, USA. [7]Department of Psychiatry and Human Behavior, University of California Irvine, Irvine 92697 CA, USA. [8]Department of Psychology, University of California Berkeley, Berkeley 94720 CA, USA. [9]Helen Wills Neuroscience Institute, University of California Berkeley, Berkeley 94720 CA, USA. [10]Department of Neurology, School of Medicine, University of California Davis, Sacramento 95817 CA, USA. [11]Center for Mind and Brain, University of California Davis, Davis 95618 CA, USA. [12]These authors contributed equally: Haoxin Zhang, Ivan Skelin. ✉e-mail: haoxinz1@uci.edu; jajlin@ucdavis.edu

# Results

## Memory discrimination is enhanced for emotional stimuli

We performed simultaneous iEEG recordings from the amygdala and the hippocampus in 7 human participants, while performing an emotional memory encoding and discrimination task[14,15] (Methods, Fig. 1a). During the encoding stage, participants were presented with a stimulus (image; stimulus encoding) and asked to rate the stimulus valence as negative, neutral, or positive (post-encoding/response). During the retrieval stage, participants were presented with one of the 3 types of stimuli - Repeats (identical), Lure (slightly different) or Novel (stimuli not seen during encoding) - and classified each stimulus as New or Old.

Memory discrimination is defined as the correct classification of: (1) Repeat stimuli as Old, (2) Novel stimuli as New, or (3) Lure stimuli as New. Participants classified Repeat stimuli and Novel stimuli with high accuracy (Repeat: 89.4 ± 2.4%, Novel: 93.9 ± 1.4%; Fig. 1b). Memory discrimination accuracy was lower for Lure stimuli, relative to both Repeat or Novel stimuli (Lure: 61.5 ± 3.7%; t(6) = 8.36, $p_{Novel vs Lure}$ = 0.0002; t(6) = 6.13, $p_{Repeat vs Lure}$ = 0.0009, two-sided paired t-test), reflecting image similarity induced memory interference. There was a strong negative association between participants' stimulus discrimination ability and lure image pair similarity rating (t(452) = −2.06, p = 0.039, see Methods, Fig. 1c, d). Stimulus-induced arousal (irrespective of valence) was associated with correct Lure discrimination, confirming previous reports[1–3] (t(452) = 1.98, p = 0.047, Fig. 1c, d, Supplementary Fig. 1). Neither the stimulus arousal (t(452) = −0.27, p = 0.785, beta = −0.024) nor valence (t(452) = 1.54, p = 0.126, beta = 0.216) were significantly

associated with correct Repeat discrimination, supporting the selective effect of arousal on correct Lure discrimination. This could be due to the lower difficulty of Repeat trials, as the correct Repeat discrimination performance was already very high (Fig. 1b), limiting the discrimination-enhancing effect of high stimulus arousal. Response times were not significantly associated with the stimulus emotional valence (F(2,18) = 0.290, p = 0.749, one-way ANOVA). The Lure discrimination index (LDI, see Methods) is a procedure used to correct for the general tendency of classifying the stimuli as New[14]. There was no significant effect of valence on LDI (F(2, 18) = 0.980, p = 0.396, one-way ANOVA). The effect of valence on LDI shows a considerable inter-participant variability (Supplementary Fig. 2a), both in the terms of absolute values, as well as the distribution across the valences. The reported relations between the valence and LDI are mixed, including both the higher and lower LDI for emotional stimuli[3,14,15]. LDI was significantly higher for high-arousal stimuli (t(6) = −2.058, p = 0.043, one-tailed paired t-test), reflecting the tendency for classifying the high-arousal stimuli as New.

Correct Lure discrimination was significantly associated with stimulus arousal (t(448) = 15.782, p = 6.15*10⁻⁴⁵) and similarity (t(448) = 50.562, p = 2.99*10⁻¹⁸⁷), while there was no significant association with valence (t(448) = 1.020, p = 0.308). In addition, there was a significant interaction between the arousal and similarity, (t(448) = 10.327, p = 1.44*10⁻²²), reflecting the highest correct Lure discrimination for the high-arousal stimuli of low similarity. There was no other significant interaction between the experimental variables (arousal x valence, similarity x valence, arousal x similarity x valence; Supplementary Table 2).

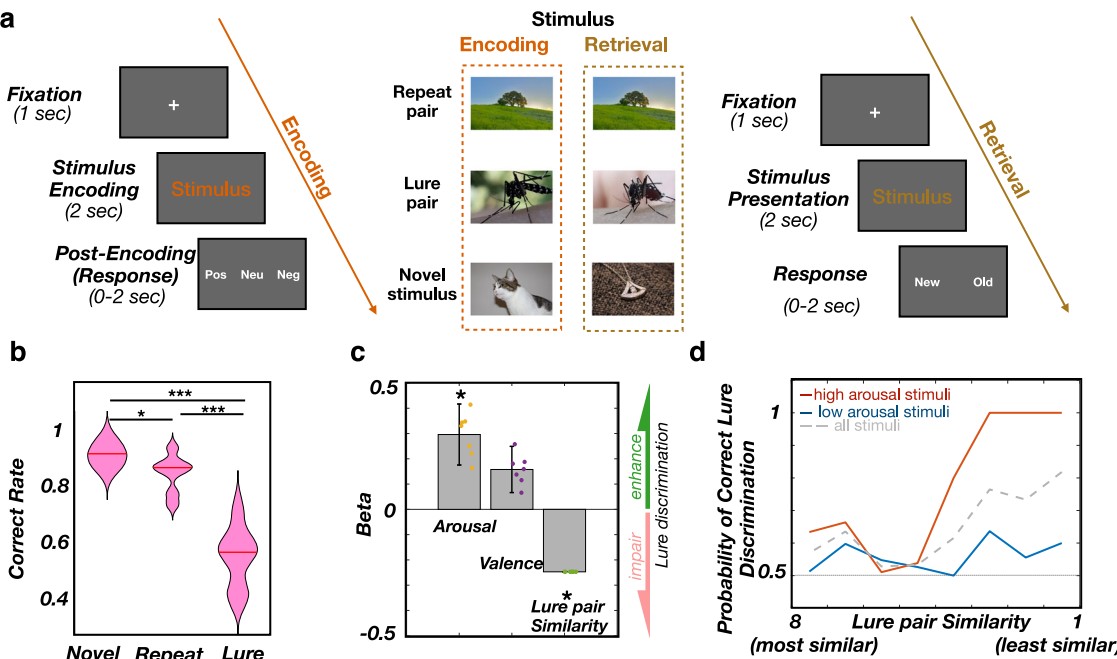

**Fig. 1 | Memory discrimination is more accurate for emotional stimuli. a** Task structure: participants are presented with an image (Stimulus encoding). Following presentation, they rate the valence of the image as negative, neutral, or positive (Post-Encoding/Response). Once all images are presented and rated, participants are presented with 3 types of stimuli - Repeat (identical), Lure (slightly different) or Novel (stimuli not seen during encoding) - and classify each stimulus as Old or New. **b** Correct discrimination is highest for Novel stimuli (93.9 ± 1.4%; median ± SEM), followed by Repeats (89.4 ± 2.4%) and Lures (61.5 ± 3.7%). Two-sided paired t-test: Novel vs. Repeat, *t(6) = 3.33, p = 0.016; Novel vs. Lure, ***t(6) = 8.36, p = 0.0002; Repeat vs. Lure, *** t(6) = 6.13, p = 0.0009. **c** Correct discrimination of Lure stimuli is positively associated with encoded stimulus-induced arousal (*t(452) = 1.98,

p = 0.047, β = 0.3 ± 0.12, two-sided logistic linear mixed-effect model) and valence (t(452) = 1.48, p = 0.137, β = 0.15 ± 0.09, n_participants = 7, two-sided logistic linear mixed-effect model), while negatively associated with lure pair similarity (*t(452) = −2.06, p = 0.039, β = −0.24 ± 0.00, n_participants = 7, two-sided logistic linear mixed-effect model). The beta sign and magnitude indicate effect direction and strength, respectively. Dots correspond to individual participants. Box and bar indicate mean and 95% CI. **d** Probability of correct Lure discrimination as a function of lure pair similarity and stimulus-induced arousal. The solid line shows the actual proportion of New responses (y-axis) as a function of Lure stimulus SI (x-axis) for low arousal (blue) or high arousal stimuli (red). The low/high arousal groups were created using the median split. Source data are provided as a Source Data file.

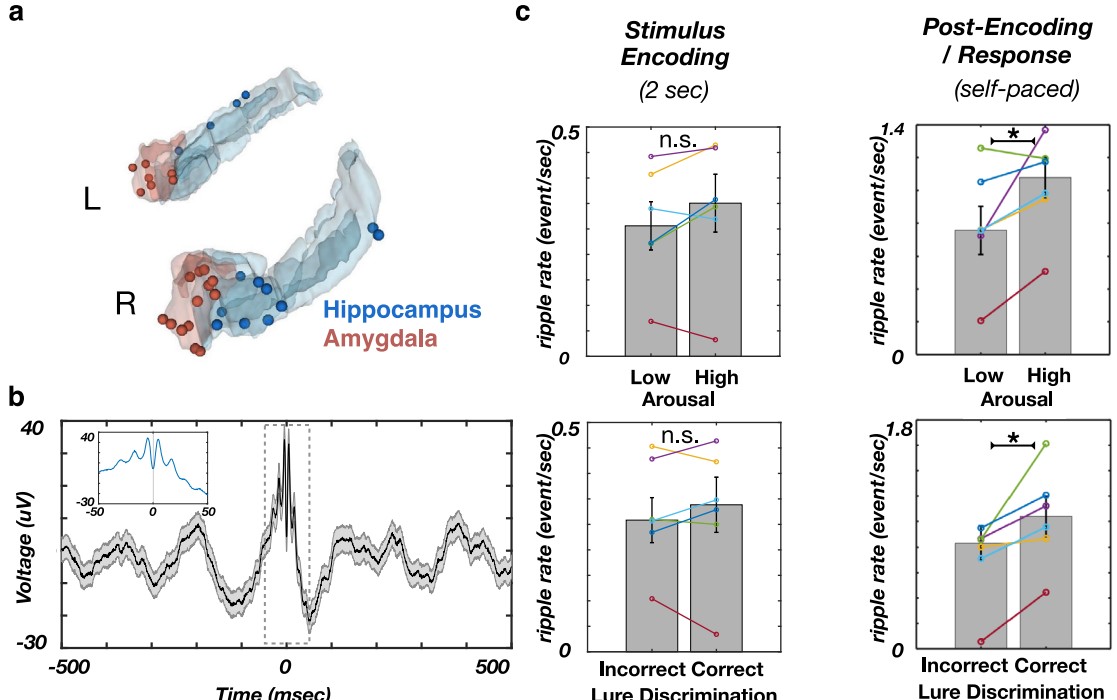

**Fig. 2 | The post-encoding ripple rate predicts the stimulus-induced arousal and memory discrimination. a** Reconstructed locations of hippocampal (blue) and amygdala electrodes (red). **b** The ripple grand average waveform ($n = 4689$ ripples in 6 hippocampal channels, 6 participants). Line and shaded areas represent the mean ± SEM. **c** The ripple rate (events/sec) is significantly higher following encoding of arousing (top right; *$z(5) = -2.0$, $p = 0.046$, Two-sided Wilcoxon signed-rank test) and later correctly discriminated stimuli (bottom right, *$z(5) = -2.2$, $p = 0.028$, Two-sided Wilcoxon signed-rank test). The ripple rate was showing no conditional differences during stimulus encoding (left column, n.s. as non-significant, p's > 0.05, Benjamini-Hochberg correction for multiple comparisons[48]). Box and bar indicate mean ± SEM. Source data are provided as a Source Data file.

## Post-encoding ripples are associated with enhanced discrimination of emotional stimuli

We defined the post-encoding period as the interval between stimulus offset and participants' stimulus valence rating response (Fig. 1a). We tested the association of post-encoding ripple rate (the number of ripple events/second) with the stimulus emotional content (stimulus-induced arousal and valence) and correct Lure discrimination during retrieval. While the behavioral analysis was performed on 7 participants, one participant was excluded from the ripple-based analysis, due to the low number of recorded ripples. The ripple-based analysis was performed on 14 hippocampus and 20 amygdala electrodes, in 6 participants. The locations of electrodes used in the analysis are shown in Fig. 2a and the average ripple waveform is shown in Fig. 2b.

Higher post-encoding ripple rate was associated with stimulus-induced arousal ($z(5) = -1.99$, $p = 0.046$, Wilcoxon signed-rank test, Fig. 2c) and also predicted correct Lure discrimination during retrieval ($z(5) = -2.20$, $p = 0.028$, Wilcoxon signed-rank test, Fig. 2c), but was not associated with stimulus valence (F (2, 15) = 1.88, $p = 0.187$, one-way ANOVA; Supplementary Fig. 3). As the stimulus arousal and correct Lure discrimination are correlated (Fig. 1c), while both being associated with post-encoding ripple rate (Fig. 2c), we tested if the association between the ripple rate and correct Lure discrimination is modulated by the stimulus arousal level. This analysis revealed the main effects of stimulus-induced arousal (F(1,20) = 4.93, $p = 0.038$) and later correct Lure discrimination (F(1,20) = 8.32, $p = 0.009$), with no significant interaction (F(1,20) = 0.26, $p = 0.619$, two-way ANOVA; Supplementary Fig. 4). This result suggests that stimulus-induced arousal and later correct Lure discrimination have an independent association with post-encoding ripples. In addition, we tested if the post-encoding ripple association with stimulus arousal/correct Lure discrimination is limited to specific periods during post-encoding epoch by performing the conditional comparisons of time-resolved

ripple rates (number of ripples/sec). Post-encoding ripple rates were significantly higher for correctly discriminated, relative to incorrectly discriminated Lure stimuli (Supplementary Fig. 5; $p = 0.005$, −400 to −50 msec relative to response time), and for high-arousal, relative to low-arousal Lure stimuli (Supplementary Fig. 5; $p = 0.035$, −780 to −600 msec; non-parametric cluster-based permutation test). To summarize, the post-encoding ripple associations with stimulus arousal/correct Lure discrimination were present during distinct, non-overlapping time windows, suggesting the distinct temporal relation between these variables and post-encoding ripples.

Taken together, these results suggest post-encoding ripples as a potential electrophysiological mechanism for enhanced memory discrimination of arousing stimuli, previously characterized at behavioral level[2,3,16]. Furthermore, the positive associations between ripples and stimulus-induced arousal/later Lure discrimination were present in all individual participants (Fig. 2c). The post-encoding response time (RT) did not differ based on stimulus-induced arousal ($z(5) = 0.7$, $p = 0.2$, $RT_{high-arousal} = 0.8 \pm 0.1$ sec; $RT_{low-arousal} = 0.6 \pm 0.2$ sec) or later Lure discrimination ($z(5) = 0.6$, $p = 0.25$, $RT_{correct} = 0.7 \pm 0.2$ sec, $RT_{incorrect} = 0.7 \pm 0.3$, Wilcoxon signed-rank test). Therefore, the associations between stimulus-induced arousal or correct Lure discrimination and post-encoding ripple rates were unrelated to post-encoding duration.

Associations between ripple rate and stimulus-induced arousal/later correct Lure discrimination accuracy were selective for the post-encoding time window. These relationships were absent for the stimulus encoding or the retrieval task stage ($p > 0.05$, Wilcoxon signed-rank test; Fig. 2c, Supplementary Fig. 6). Two-way ANOVA was used to test if the association between the ripple rate and correct Lure discrimination is task epoch-dependent. The analysis shows significant main effects of task epoch (F (2, 30) = 103.91, $p < 0.001$) and correct Lure discrimination (F(1, 30) = 9.67, $p = 0.004$). In addition, we

observed significant epoch x discrimination interaction (Supplementary Fig. 7; F(2, 30) = 10.97, $p = 0.0003$, two-way ANOVA). Post-hoc comparisons revealed the significantly higher ripple rates during post-encoding epoch for the correctly discriminated Lure stimuli (post-encoding: M(6) = −1.70, $p < 0.001$, 95% CI = [−1.08, −0.47]), with no significant conditional differences during the encoding or retrieval epochs ($p$'s > 0.05; multcompare.m function in Matlab). To summarize, the analysis shows that the correct Lure discrimination is selectively associated with the ripple rate during post-encoding, but not during encoding or retrieval epochs.

There was no significant association between the post-encoding theta power and later correct Lure discrimination (t(261) = 0.187, $p = 0.851$, beta = 0.008, logistic regression). Overall, 30.8 ± 7.4% (mean ± SEM) of Lure trials contained one or more ripples during the post-encoding period. Ripple probability was significantly higher during low theta power periods (Supplementary Fig. 8), consistent with observations of ripple suppression during periods of pronounced theta oscillations[11,13]. In addition, ripples did not overlap with increased broadband gamma power, suggesting that ripples are distinct from non-specific broadband power fluctuations[17] (Supplementary Fig. 8).

## Stimulus similarity is increased during post-encoding ripples

Recent studies suggest that post-encoding memory reinstatement supports successful subsequent memory retrieval[7,8] and ripples are associated with reactivation of pre-established neuronal patterns[18]. We hypothesized that stimulus similarity during the post-encoding ripple windows could enhance later memory discrimination. Distinct neural populations have been proposed to represent individual stimuli, resulting in stimulus-specific high-frequency activity (HFA) patterns[19,20]. We quantified stimulus similarity as the Spearman correlation between HFA power spectral vectors (PSVs), for each combination of the encoding-response time bins from the same trial (see Representational Similarity Analysis in Methods). Next, we computed the average stimulus similarity during ±250 msec around post-encoding ripple peaks. The similarity significance was determined relative to a null distribution, obtained by circular jittering of ripple timestamps. The post-encoding ripple-locked stimulus similarity was stronger for arousing and correctly discriminated stimuli (Supplementary Fig. 10). To assess specific contributions of the amygdala and the hippocampus to this phenomenon, we calculated post-encoding stimulus similarity for each region, relative to ripple peak (Fig. 3a). The significant stimulus similarity period in the amygdala consisted of two intervals, the first starting slightly earlier and overlapping with the stimulus similarity in hippocampus (−105 to −50 msec relative to ripple peak), and a second period following the stimulus similarity in hippocampus (40 to 200 msec relative to ripple peak). The significant similarity period in the hippocampus lasted from −100 to 50 msec (Fig. 3b). These results demonstrated region-specific timing of the post-encoding ripple-locked stimulus similarity in the amygdala and the hippocampus. We then analyzed the association of the post-encoding stimulus similarity with the stimulus-induced arousal and later Lure discrimination. The amygdala, but not the hippocampus, showed a positive association between ripple-locked stimulus similarity and the stimulus-induced arousal (AMY: −80 to −10 msec, $p = 0.035$; HPC: $p > 0.05$, see Methods; Fig. 3c). In contrast, the hippocampus, but not the amygdala, revealed a positive association between ripple-locked stimulus similarity and later correct Lure discrimination (AMY: $p > 0.05$; HPC: −15 to 90 msec, $p = 0.008$, see Methods; Fig. 3c). In addition, the post-encoding ripple-locked similarity in the amygdala was more strongly associated with stimulus arousal than the ripple-locked similarity in the hippocampus (−70 to 20 msec relative to ripple peak, $p < 0.001$, non-parametric cluster-based permutation test, Fig. 3d). In contrast, the ripple-locked similarity in hippocampus was more strongly associated with later correct Lure discrimination than the amygdala ripple-locked similarity (−60 to

10 msec relative to ripple peak, $p = 0.046$, non-parametric cluster-based permutation test, Fig. 3d). In addition, the regional double-dissociation of stimulus-induced arousal and later correct Lure discrimination was tested by comparing the low vs. high stimulus-induced arousal trials and correct vs. incorrect Lure discrimination trials, separately for amygdala and hippocampus. This analysis shows that the association between the stimulus-induced arousal and ripple-locked similarity is significantly stronger than the association between the later correct Lure discrimination and ripple-locked similarity in the amygdala (−42 msec to 0 msec relative to ripple peak, $p = 0.034$). The opposite pattern was present in the hippocampus, where the association between the later correct Lure discrimination and post-encoding ripple-locked stimulus similarity was significantly stronger (−83 msec to 10 msec relative to ripple peak, $p = 0.047$, non-parametric cluster-based permutation test; Supplementary Fig. 9). To summarize, post-encoding ripple-locked stimulus similarity in the amygdala and the hippocampus were associated with reactivation of distinct aspects of encoded stimuli (i.e., the amygdala for stimulus-induced arousal and the hippocampus for later Lure discrimination accuracy).

The stimulus similarity on the trials not containing post-encoding ripples was not significantly different based on the stimulus arousal level or later correct Lure discrimination (non-parametric cluster-based permutation test, $p$'s > 0.05; Supplementary Fig. 11). This result further highlights the role of post-encoding ripples/ripple-locked similarity in consolidation of emotional memories.

The post-encoding ripple-locked neural activity in the hippocampus (−190 to 20 msec, relative to ripple peak) shows the significant stimulus-specific similarity with the activity during stimulus encoding (-500–750 msec following the onset of encoding epoch; non-parametric cluster-based permutation test; $n = 1000$ permutations, $p < 0.05$; Supplementary Fig. 12).The size/timing of significant stimulus similarity during encoding is consistent with previous reports[21,22], but might also be driven by the factors specific to present study, such as the comparison between the encoding and post-encoding ripple activity. The peak ripple-locked stimulus similarity occurred significantly earlier in the amygdala, than in the hippocampus (difference: −18 ± 11 msec, mean ± SEM; t(5) = −3.89, $p = 0.006$, one-tail paired t-test), with the timing difference being consistent across the participants (Supplementary Fig. 13).

## Joint ripple-locked stimulus similarity increase in hippocampus and amygdala

In rodents, the coordinated memory reactivation in the amygdala and hippocampus during sleep ripples is proposed to bind neuronal ensembles encoding emotional and spatial information, respectively[20]. We reasoned that a similar interaction between the amygdala and the hippocampal exists in which cross-regional post-encoding ripple-locked stimulus similarity facilitates later discrimination. We hypothesized that the periods of stimulus similarity in both structures co-occur during the same ripple event and follow a consistent temporal dynamic. To test this, we separately computed ripple-locked joint stimulus similarity for the correctly and incorrectly discriminated stimuli (Methods). A significant joint ripple-locked stimulus similarity in the amygdala and hippocampus was present during the post-encoding period only for correctly discriminated stimuli (Fig. 4a) and was maximal around ripple peaks (Supplementary Fig. 14). Specifically, the amygdala stimulus similarity preceded the hippocampal stimulus similarity by -100 msec. Further, mutual information analysis showed a significant unidirectional influence from the amygdala to the hippocampus before ripple peak (−70 to −30 msec, $p = 0.038$; see Methods; Fig. 4b). Ripple-like events were also detected in the amygdala, similar to recent reports[23]. However, only 5.89 ± 1.82% (mean ± SEM) of hippocampal ripples were accompanied by ripple-like events in the amygdala within the ± 50 msec window. Thus, the joint increases in post-encoding stimulus similarity during hippocampal ripples (Fig. 4)

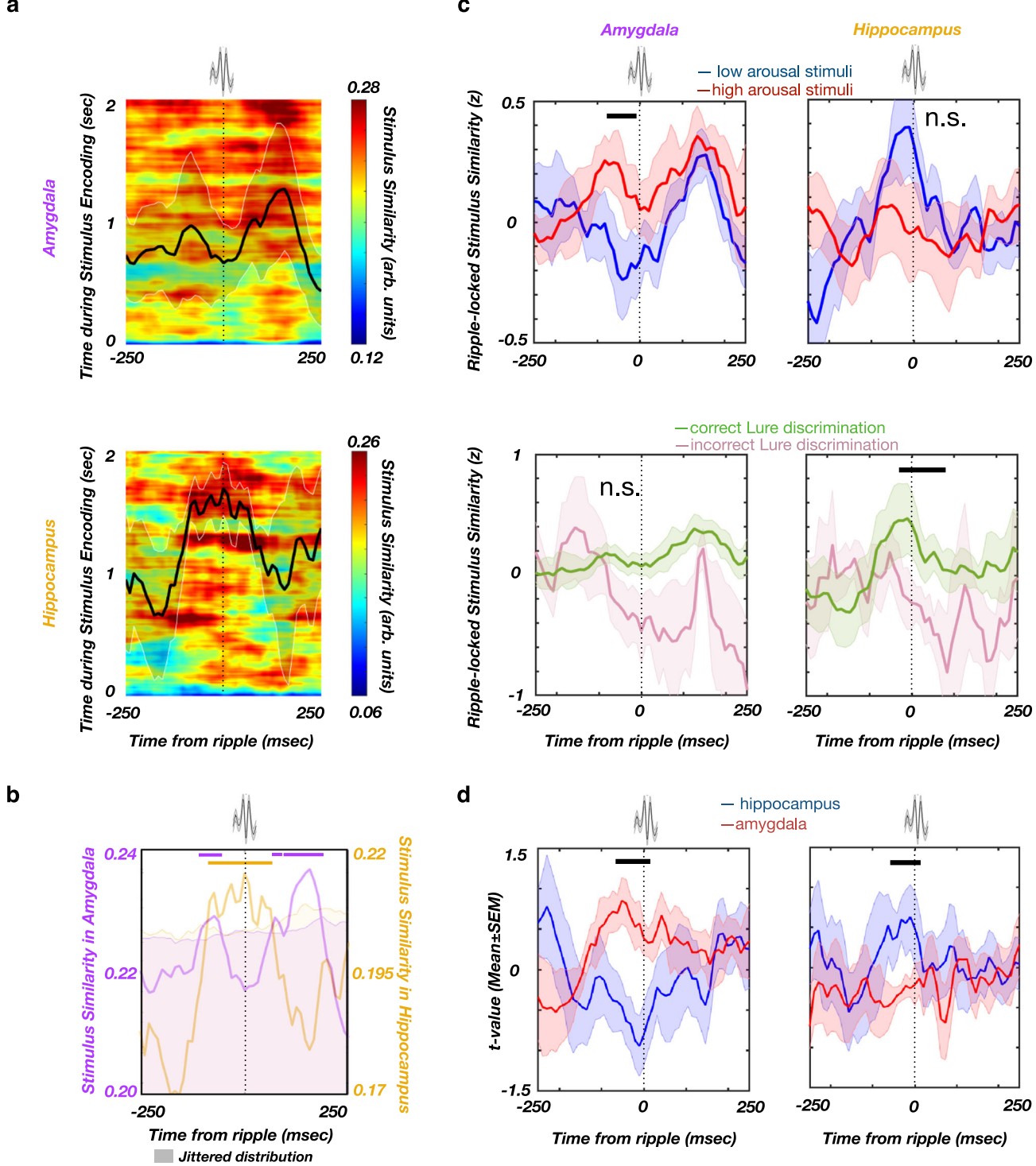

**Fig. 3 | Post-encoding stimulus similarity in the hippocampus and amygdala around ripple. a** Ripple-locked similarity in the amygdala (top) and hippocampus (bottom) during the post-encoding period (line and shaded areas represent the mean ± SEM). **b** Post-encoding stimulus similarity is greatest around the time of ripples as shown by comparison with the null-distribution (within ± 250 msec). Shaded areas denote the null-distribution 95% confidence interval. Similarity in the hippocampus overlaps with ripple peak (orange), while similarity in the amygdala peaks prior to and after the ripples (magenta). **c** Ripple-locked post-encoding stimulus similarity in the amygdala is significantly higher for arousing stimuli (top left, *p* = 0.035, see Methods; two-sided non-parametric cluster-based permutation test) but is not associated with subsequent discrimination (bottom left, n.s. as non-significant, *p* = 0.066). Ripple-locked post-encoding stimulus similarity in the hippocampus is significantly higher for correctly discriminated Lure stimuli (bottom

right, *p* = 0.008, see Methods; two-sided non-parametric cluster-based permutation test) but does not depend on stimulus-induced arousal (top right, n.s. as non-significant, *p* > 0.1). Line and shaded areas represent the mean ± SEM. **d** Double-dissociation between the post-encoding ripple-locked stimulus representation in hippocampus and amygdala. Left: The association between the stimulus arousal and post-encoding ripple-locked stimulus similarity was stronger in the amygdala (−70 to 20 msec relative to ripple peak, *p* < 0.001, one-sided non-parametric cluster-based permutation test). Right: The association between the later correct Lure discrimination and post-encoding ripple-locked stimulus similarity was stronger in the hippocampus (−60 to 10 msec relative to ripple peak, *p* = 0.046, one-sided non-parametric cluster-based permutation test). The line and shaded areas represent the mean ± SEM of the individual participant t-values. Source data are provided as a Source Data file.

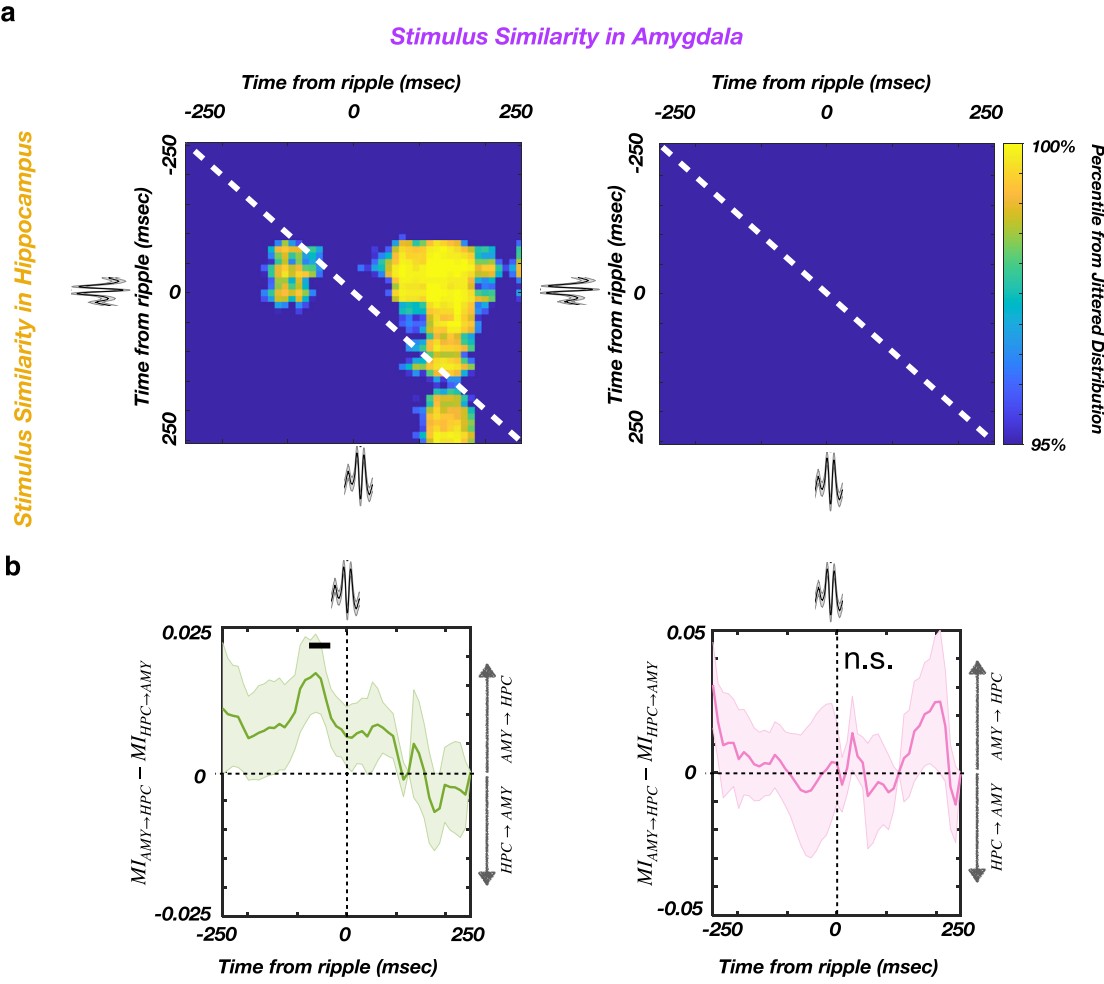

**Fig. 4 | Synchronously increased ripple-locked post-encoding stimulus similarity in the hippocampus and amygdala predicts the correct Lure discrimination. a** The ripple-locked joint stimulus similarity in the hippocampus and amygdala for the correct (left) and incorrect (right) discrimination trials. Significant similarity in the amygdala starts 100 msec prior to the ripple peak, followed by the hippocampus (−50 to 200 msec). There is no significant joint stimulus similarity during incorrect Lure discrimination trials, suggesting that the cross-structure joint stimulus similarity may be required for correct Lure discrimination. **b** Mutual information (abbreviated as MI) difference for the amygdala (abbreviated as AMY) and hippocampal (abbreviated as HPC) stimulus similarity time-courses, during the post-encoding ripple windows (correct Lure discrimination - top, incorrect Lure discrimination - bottom). Positive values denote stronger amygdala to hippocampus (AMY → HPC) directionality. A temporal cluster of significant MI difference (AMY → HPC) is present before ripple peak time (−70 to −30 msec) after encoding of correctly discriminated Lure stimuli (top; $p = 0.038$, see Methods), indicating that hippocampal stimulus similarity is better predictable by amygdala stimulus similarity than vice versa. This effect is present only during the post-encoding period for correctly discriminated Lure stimuli (left), but not for the incorrectly discriminated Lure stimuli (right, n.s. as non-significant). The line and shaded areas represent the mean ± SEM of the individual participant MI difference. Source data are provided as a Source Data file.

were not dependent on coincident presence of ripple-like activity in the amygdala. To conclude, ripple-mediated coordination of post-encoding stimulus representation in the amygdala and the hippocampus promotes later successful discrimination.

## Discussion

Rodent studies have implicated ripples in the retrieval and consolidation of emotional memory. However, it is unclear whether ripples support the memory benefits of emotional experience[24]. Our study reveals an association of higher ripple rate with stimulus-induced arousal and subsequent correct stimulus discrimination, providing direct evidence for ripple-mediated strengthening of emotional memory. Interestingly, the increase in ripples has been shown in rodents after exposure to a novel or reward-associated context[25]. Together, this suggests that ripples may play a general role in the selective enhancement of salient experiences[26]. Notably, such association is specific to the post-encoding period that starts immediately

after memory encoding, when memory retrieval is essential to rate the emotional content of the stimuli. A recent study reported that ripple levels during memory encoding or in a brief post-encoding period (~400 ms) were not predictive of subsequent memory for the presented stimulus[27]. While this corresponds with the lack of significant association between the encoding ripples and correct Lure discrimination in the present study (Fig. 2c), our results differ with respect to the role of post-encoding ripples in subsequent memory (Fig. 2c). This discrepancy could be due to presence of emotional stimuli in the present study, which were associated with higher post-encoding ripple activity (Fig. 2c, Supplementary Figs. 4, 5), or with the requirement for stimulus valence rating in the present study, which might have triggered ripple emergence and indirectly facilitated consolidation. This finding supports theoretical assumptions that ripples mediate both the retrieval of stored representation utilized in decision-making, and the strengthening of the same representation, contributing to memory consolidation[25].

Post-encoding stimulus similarity (or reinstatement) is implicated in memory consolidation[7,8,28,29] and peaks during ripples (Fig. 3). Thus, beside the general role of ripples in memory consolidation, these results also imply ripples as a potential mechanism mediating the effects of arousal on memory consolidation. It should be noted that a subset of trials contained no ripples during the post-encoding period. During these trials ripple amplitudes might have been below our detection threshold. In addition, the ripples might have occurred in the different parts of the hippocampus or in the contralateral hemisphere. The frequency overlap between the broadband gamma (30–300 Hz) and ripples (80–150 Hz) could theoretically result in a circularly inflated stimulus similarity during post-encoding ripples. However, this scenario would not explain the stimulus-specificity of ripple-locked activity (Supplementary Fig. 12), nor the significant association between the post-encoding ripples and stimulus arousal/later correct Lure discrimination (Fig. 3). Therefore, the ripple-locked stimulus similarity during post-encoding epoch is likely functionally relevant for emotional memory consolidation, rather than an epiphenomenon of broadband gamma/ripple frequency overlap.

Next, we aimed to discern the link between the ripple-associated interaction between the amygdala and hippocampus during post-encoding and subsequent memory effect. We found the ripples were accompanied by increased stimulus similarity during the post-encoding period. Specifically, the post-encoding stimulus similarity in the amygdala appears shortly before the ripple peak and shows association with arousing stimuli, while the post-encoding stimulus similarity in the hippocampus appears around the ripple peak and shows associations with correct subsequent memory discrimination. Moreover, the co-occurrence of the post-encoding stimulus similarity in the amygdala and hippocampus during the same ripple events - with the amygdala leading hippocampus by ~100 msec - is predictive of subsequent correct memory discrimination. Similarly, the directional influence from amygdala to hippocampus during encoding was predictive of subsequent memory effect in human participants performing emotional memory task[30]. This finding suggests that the coordinated increase in post-encoding stimulus similarity across the amygdala and hippocampus during ripples is responsible for combining emotional and contextual aspects of the memory[24,31].

Post-encoding ripples, as defined in the study, occur immediately following the stimulus encoding and might represent the initial stage of stimulus memory consolidation. This stage could be particularly relevant in the present task setting, since the stimulus consolidation might be interfered by the presentation of consecutive stimuli. Transient peaks of post-encoding stimulus similarity during ripples (Fig. 3) might strengthen the connectivity between the neurons participating in stimulus representation, both within the hippocampus and in the hippocampus/amygdala circuitry (Fig. 4). In addition, the better discriminability of high arousal Lure stimuli with low within Lure pair similarity (Supplementary Table 2) might reflect the higher encoding fidelity and/or more efficient consolidation of arousing stimuli, as suggested by the higher ripple-locked post-encoding stimulus similarity for arousing stimuli (Figs. 3 and 4). These mechanisms might result in a higher fidelity stimulus representation at retrieval, allowing a more reliable discrimination from the corresponding Lure stimulus.

Ripples are associated with synchronous activation of neuronal ensembles in the hippocampus and connected structures[31–33]. The onset of ripple-locked similarity in the amygdala prior to ripple peak (Fig. 3c) suggests a sequential process whereby amygdala activation triggers the hippocampal ripple, followed by the cascade of ripple-associated plasticity[34,35], resulting in higher probability of later correct Lure discrimination. Similarly, electrical stimulation of the amygdala could potentially induce the hippocampal recruitment equivalent to endogenous ripples and facilitate the consolidation of recently encoded content, as demonstrated by Inman et al.[36]. On the other hand, high frequency electrical stimulation of the hippocampus during

encoding impairs emotional memory consolidation[37], suggesting the disruption of consolidation-related hippocampal dynamics.

The presence of ripple-like activity, coincident with hippocampal ripples, was recently reported in the human amygdala[23] and cortical areas[38–40]. While the presence of ripples in the extrahippocampal structures, especially in the epileptic brain, is still a subject of debate[41], it is possible that the post-encoding stimulus similarity in the amygdala is driven by ripple-like activity. This would be consistent with the co-occurrence of hippocampal ripples and ripple-like activity in temporal cortex during memory retrieval[40]. However, only ~5% of hippocampal ripples were accompanied by ripple-like activity in the amygdala. Regardless of the nature of ripple-like activity in the amygdala, the relatively low coincidence with hippocampal ripples suggests the sufficiency of hippocampal ripples alone for coordinating the joint stimulus representation across the structures (Fig. 4a, Supplementary Fig. 14).

The post-encoding stimulus similarity occurs during the stimulus valence rating, immediately following the stimulus encoding (Fig. 1a). Therefore, the higher post-encoding similarity might be driven by either the retention of stimulus representation in working memory or by memory reinstatement. The interpretation that the post-encoding stimulus representation reflects working memory content is consistent with the proposed role of hippocampal ripples in working memory[13,42,43]. While working memory is traditionally conceptualized as persistence of stimulus-specific activity[44], the post-encoding similarity in the present study is concentrated around ripple peaks (Fig. 3). Therefore, to the extent that post-encoding similarity is driven by working memory, it is consistent with the intermittent representation of working memory content[45]. On the other hand, the stimulus valence rating during post-encoding epoch could also rely on memory retrieval, which was associated with ripple emergence[46]. Regardless of the underlying mechanisms, the post-encoding ripple-locked stimulus similarity is associated with later correct Lure discrimination. This could be due to the contribution of post-encoding ripples to memory consolidation, resulting in a higher fidelity of encoded representation and higher probability of later correct Lure discrimination.

Lure stimulus discrimination is arguably a more difficult cognitive task, relative to recognition of Novel or Repeat stimuli, as reflected by the performance difference between these trial categories (Fig. 1b). While it is possible that post-encoding ripples also support these forms of recognition, it could be obscured by the ceiling effect, as the performance on Novel or Repeat trials might be high regardless of the ripple presence. In addition, a small number of incorrect discrimination Novel or Repeat stimuli would not provide enough statistical power to answer this question in the current setting.

The number of participants in the present study is relatively low, due to paucity of participants with simultaneous amygdala and hippocampus recordings outside of seizure onset zone. However, the effects are consistent across the individual participants (Figs. 1c, 2c and Supplementary Figs. 2a, b, 3, 6, 8b, 9, 14), mitigating the possibility of outliers affecting the study conclusions.

To summarize, both the joint post-encoding similarity and mutual information analyses confirm the predictive validity of directional influence from the amygdala to the hippocampus before ripples on correct Lure discrimination, establishing a link between the amygdala post-encoding stimulus similarity and memory discrimination as a physiological mechanism of emotional memory enhancement. Together, our data support a model wherein the post-encoding stimulus similarity in the amygdala, triggered by emotional stimuli, elicits hippocampal ripple-associated stimulus similarity, which facilitates subsequent memory performance.

## Methods
### Participants
Intracranial electroencephalography (iEEG) recordings were obtained from 7 participants (3 females; mean age ± SD = 33 ± 16), undergoing

presurgical monitoring of epileptic foci at the University of California Irvine Medical Center (UCIMC) Epilepsy Monitoring Unit. The individual participant demographic information is shown in Supplementary Table 1. Only the participants with the correct discrimination rate of Novel trials >= 85% (see Emotional memory encoding and discrimination task) were included in the analysis. The behavioral inclusion threshold (> 85% performance on Novel trials) was used as a sensitive indicator of participants attention level. Correct performance on Novel trials required classifying the stimuli encountered for the first time and with no similarity to previously encountered stimuli as New. One participant was excluded from the behavioral analysis due to low performance on Novel trials (60.5%, z = −2.34), while the rest of the participants performed at much higher level (93.3 ± 1.6%, mean ± SEM). In addition, this participant performed close to the chance level (50%) across all the trial types combined (51.7%, z = −2.43). Electrode placements were determined entirely based on clinical considerations. All the research procedures were approved by the UCI Institutional Review Board and data was collected following informed consent.

## Statistics

All the statistical tests were performed with the individual participant as the unit of analysis. Unless stated otherwise, all the statistical tests (e.g., Wilcoxon signed-rank test, t-test) were two-tailed. The effects of valence, stimulus-induced arousal and similarity on stimulus discrimination (Fig. 1c) were assessed using the logistic linear mixed-effect model (for details, see Behavioral Analysis). The association between post-encoding ripples and stimulus arousal/correct Lure discrimination was tested using ripple rates (number of ripples/sec, Fig. 2; Supplementary Figs. 4–7). Except for the Wilcoxon signed-rank test analysis shown in Fig. 2c, ripple rates were normalized at individual participant level using z-score. Conditional comparisons of ripple rates (correct/incorrect Lure discrimination or high/low arousal; Fig. 2c) were done using the Wilcoxon signed rank test ($p < 0.05$). Associations between the stimulus-induced arousal/later correct Lure discrimination and ripple rate were analyzed using two-way ANOVA (anovan.m function in Matlab, $p < 0.05$; Supplementary Figs. 4 and 6). The epoch-dependence of ripple association with correct Lure discrimination was tested using the two-way ANOVA ($p < 0.05$; Supplementary Fig. 7). Post-hoc tests were done using the multcompare.m function in Matlab. The non-parametric statistical test was used due to non-normal distribution of ripple numbers across the participants. The association between the post-encoding theta power on ripple channels and later correct Lure discrimination was tested using the logistic regression ($p < 0.05$).

Statistical significance of ripple-locked post-encoding stimulus similarity (Fig. 3b) was assessed by comparing the real test statistics with empirical null distribution, obtained using Monte Carlo method (for details, see Representational Similarity Analysis). We implemented the non-parametric cluster-based permutation test[47] to assess the conditional differences (correct/incorrect Lure discrimination or high/low arousal) of post-encoding stimulus similarity (Fig. 3c) and mutual information (Fig. 4b), by randomly shuffling the conditional trial labels 1000 times (for details, see Representational Similarity Analysis). Similarly, the significant temporal windows for the cross structure ripple-locked joint post-encoding stimulus similarity (Fig. 4a) were assessed by comparing to empirical null distribution (for details, see Joint post-encoding stimulus similarity analysis). The correction for multiple comparisons was performed using the Benjamini-Hochberg procedure[48]. To compare the timing of ripple-locked stimulus similarity between the hippocampus and amygdala, the peak similarity timings were computed during post-encoding ripple windows, following the encoding of later correctly discriminated stimuli. Next, the peak similarity timings were compared between the regions using the one-tail paired t-test ($p < 0.05$; Supplementary Fig. 13).

## Emotional memory encoding and discrimination task

The emotional memory encoding and discrimination (EMOP) task consists of encoding and discrimination blocks. During the encoding block (148 trials), each trial consists of a cross fixation (1000 msec), followed by stimulus encoding (2000 msec) and self-paced post-encoding response period (up to 2000 msec). During the post-encoding response period, participants are asked to classify the stimulus emotional valence as either negative, neutral or positive, using the corresponding laptop key. During the retrieval block (290 trials), trial time structure is identical to encoding phase. Following the cross fixation (1000 msec), the participants are presented for 2000 msec with a stimulus identical (Repeat, 54 trials), slightly different (Lure, 97 trials) or unrelated (Novel, 139 trials) to previously encoded stimuli. Next, during the self-paced memory discrimination epoch (up to 2000 msec), participants are asked to discriminate if the presented stimulus was seen during encoding (Old) or not (New). Correct discrimination is defined as classifying the Repeat stimuli as Old and Lure or Novel stimuli as New. The stimuli were selected from the continuous distributions across the valence and stimulus-induced arousal axes (Supplementary Fig. 1). The same set of stimuli was used across participants. In addition, the valence, arousal and lure pair similarity of each stimulus were rated by separate cohorts of healthy participants (also used in Leal et al.[14]). Specifically, a first cohort ($N = 50$, 32 females; age mean ± SD = 22 ± 5) rated the stimulus emotional valence on a continuous scale (range 1–9, with 1 denoting the most negative, 9 the most positive, and 5 neutral valence). Stimuli were assigned in Negative (valence <= 3.5), Neutral (3.5 > valence < 6) or Positive (valence >= 6) groups. Another cohort of healthy participants ($N = 16$, 4 females; age mean ± SD = 23 ± 5) rated the stimulus-induced emotional arousal on a scale 1–9 (1 being the least and 9 being the most arousing). Finally, a third cohort ($N = 17$, 11 females; age mean ± SD = 20 ± 1) examined relative lure pair similarity between the pair of lure images presented during encoding and retrieval stage on the scale 1–8[14]. The rationale for obtaining the categorical ratings from study participants was the need of using the sliding scale for obtaining the continuous ratings, which would introduce systematic difference in response times, depending on the scale distance. The continuous ratings from healthy participants were used for behavioral/neurophysiological correlation based on the: a) better feasibility of continuous behavioral variables for correlation with neural signals and b) high correspondence between the continuous (healthy participants) and categorical (study participants) ratings (~85%; Supplementary Fig. 1b). The high correspondence of stimulus valence ratings obtained from study participants and healthy population suggests the intact emotional processing in study participants (Supplementary Fig. 1).

## Behavioral analyses

To assess the effects of valence, stimulus-induced arousal and similarity on Lure stimulus discrimination, we implemented the logistic linear mixed-effect model.

$$y = \beta X + uZ + \varepsilon \tag{1}$$

In this model, $y$ indicates the responses across the individual Lure discrimination trials (0-Old; 1-New), $X = [x_1, x_2, x_3]^T$ denotes three fixed effect regressors (encoded stimulus valence and arousal as well as similarity between the encoded and Lure stimulus), $Z = [z_1]^T$ denotes random effect regressor (participant identity), $\beta$ and $u$ denote the fixed and random-effect regression coefficients, and $\varepsilon$ denotes the error term. The model includes random intercept to incorporate individual participant differences. We normalized the valence, stimulus-induced arousal and similarity values relative to the scale of 0 to 1. The statistics reported in Fig. 1c corresponds to the fixed-effect coefficients $\beta$. Lure discrimination index (LDI) is defined as the difference in the probability of Lure and Repeat stimuli being classified as

New (p(New|Lure) – p(New|Repeat)). This procedure corrects for the general tendency of classifying the stimuli as New[14]. The effect of valence on LDI was tested using one-way ANOVA ($p < 0.05$). The LDI comparison between the low- and high-arousal stimuli was performed using the one-tailed paired t-test ($p < 0.05$). The association between the valence and response times was tested using one-way ANOVA ($p < 0.05$). The effects of stimulus arousal and valence on the correct discrimination were tested using the linear mixed-effects model (LME; $p < 0.05$).

## Data collection
The behavioral experiment was administered using the PsychoPy2 software[49] (Version 1.82.01). The laptop was placed at a comfortable distance in front of the participant. The iEEG signal was recorded using a Nihon Kohen system (256 channel amplifier, model JE120A) and NeuraLynx ATLAS acquisition system, with an analog high-pass filter (0.01 Hz cutoff frequency) and sampling frequency 5000 Hz.

## Data analysis
The following softwares and packages were used: MATLAB (Version 9.7); Fieldtrip Toolbox; Advanced Normalization Tools(ANTs); Freely Moving Animal Toolbox(FMA); Wavelet Toolbox; Signal Processing Toolbox; Statistics and Machine Learning Toolbox; EEMD package (https://github.com/leeneil/eemd-matlab.git).

## Electrode localization
We localized each electrode using pre-implantation structural T1-weighted MRI scans (pre-MRI) and post-implantation MRI scans (post-MRI) or CT scans (post-CT). Specifically, we co-registered pre-MRI and post-MRI (or post-CT) scans by means of a rigid body transformation parametrized with three translation in x,y,z directions as well as three rotations using Advanced Normalization Tools (ANTs https://stnava.github.io/ANTs/). We implemented a high-resolution anatomical template with the label of medial temporal lobe subfields[14] to guide the localization for individual electrodes. We resampled the template with 1 mm isotropic, and aligned it to pre-MRI by ANTs Symmetric Normalization[50] to produce a participant-specific template. The electrode localization was identified by comparing the participant-specific template subfield area with electrode artifacts (Fig. 2a). The localization results were further reviewed by the neurologist (J.J.L.).

## Preprocessing
The signal preprocessing was done using the custom-written MATLAB code (Version 9.7) and Fieldtrip Toolbox[51]. The 60 Hz line noise and its harmonics were removed using a finite impulse response (FIR) notch filter (ft_preprocessing.m function in FieldTrip). The EEG signal was down-sampled to 2000 Hz, demeaned and high-passed filtered (cutoff frequency 0.3 Hz). The power spectrum density (PSD) was computed using the multitaper method with the Hanning window (ft_freqanalysis.m function in FieldTrip). All the channels were re-referenced to the nearest white matter channel from the same depth electrode, based on the electrode localization results. The interictal epilectic discharges were manually marked by an epileptologist (J.J.L.), using the ft_databrowser.m function in FieldTrip. The channels with severe contamination and trials containing epileptiform discharges were excluded from further analyses.

## Ripple detection
Following the removal of channels with excessive epileptic activity and individual trials containing visually identified interictal epilectic discharges, ripples were detected on the remaining hippocampal channels, using the Freely Moving Animal Toolbox (FMA; http://fmatoolbox.sourceforge.net/). Only the hippocampal channels were used in ripple detection for ripple-based analysis. First, the iEEG traces from the trials used in the analysis were concatenated. Next,

concatenated traces were bandpass-filtered (80–150 Hz, Chebyshev 4th order filter, function filtfilt.m in Matlab). The analytical amplitude was obtained by computing the absolute value of Hilbert-transformed filtered trace (function hilbert.m in Matlab). The analytical amplitude values during periods ± 75 msec around the trial onsets/offsets were set to zero, to avoid the edge effects resulting from concatenating discontinuous traces. Finally, the envelope was z-scored and threshold-based ripple detection was performed on z-scored trace (Supplementary Fig. 15a). Detected events were considered ripples if the z-scored analytical amplitude remained above the lower threshold ($z = 2$) for 20–100 msec and if the peak value during this period exceeded higher threshold ($z = 5$). The prominence of the sharp-wave component in the ripple waveform depends on the optimal electrode position in the hippocampal layers[52]. As the electrode position in human participants can't be optimized post-surgically, this likely accounts for the absence of prominent sharp-wave component in the ripple waveform in this study, similar to other published examples of human ripples[25,27,38,53]. As an additional control analysis, we compared the ripple detection from the hippocampal iEEG signal with the event detection from the synthetic signal of the same spectral characteristics, using the identical detection procedure (see Ripple detection). Only the channels with z-scored number of detected ripples > −2 and number of detected events higher than in the synthetic signal (see Ripple detection from synthetic signal) were used in the analysis. The participant 1 was excluded from the analysis based on the low number of detected putative ripple events (z-score < −2, relative to distribution of detected ripple numbers across the participants, which is not higher than chance level (Supplementary Fig. 16). If the multiple channels from a single participant passed this criteria, a channel with the highest number of detected ripples was selected for further ripple-related analysis. Ripple-locked windows for both the hippocampal and amygdala activity were performed relative to ripple timestamps from the hippocampal channel with the highest number of detected ripples within a given participant. Ripple-like activity in the amygdala was detected using the same algorithm. Coincidence between the hippocampal ripples and ripple-like activity in the amygdala was calculated as the percentage of hippocampal ripples accompanied by amygdala ripple-like activity within the ± 50 msec. Only the Lure trials were used in ripple-based analysis.

## Time-resolved ripple rates
The time-resolved ripple rates (Supplementary Fig. 5) were calculated for individual epochs (encoding, post-encoding) and conditions (low and high stimulus-induced arousal, correct and incorrect Lure discrimination), using the msec bin size and msec step size. The resulting time-resolved ripple rate was smoothed with a Gaussian kernel ($\sigma = 150$ msec) and averaged across the trials. The time-resolved ripple rate was compared based on the low vs. high stimulus-induced arousal and later correct vs. incorrect Lure discrimination contrasts, using non-parametric cluster-based permutation test ($p < 0.05$).

## Ripple detection from synthetic signal
The power spectral density (PSD) was calculated for each hippocampal channel used in ripple detection. Next, the channel-specific filter was applied on a random signal with Gaussian distribution, resulting in a synthetic signal with identical spectral slope as the hippocampal signal. Specifically, the synthetic signal was first transformed to frequency domain by N-point Fourier transform (N denoting the number of datapoints in the signal), followed by multiplication of resulting spectrum by the PSD coefficients and application of inverse Fourier transform, to convert the signal back to time domain. The resulting synthetic signal mimicked the hippocampal channel-specific spectral characteristics. Next, the ripple detection procedure (see Ripple detection) was applied on the synthetic signal. Finally, as an additional control, the numbers of detected events were compared between the

hippocampal iEEG signal and channel-specific synthetic signal. In all the 6 participants used in the ripple-based analysis (participants 2–7), the numbers of detected ripples were higher than the numbers obtained from the participant-specific synthetic signals (Supplementary Fig. 16). Participant 1 was excluded from the analysis based on the two criteria: a) low number of detected putative ripple events (z-score < −2, relative to distribution across the participants) and b) the number of detected events lower than chance level, defined as the number of detected events in synthetic signal (Supplementary Fig. 16).

## Unsupervised decomposition of iEEG signal

To assess the post-encoding stimulus similarity, high-frequency activity (HFA; 30–280 Hz) was used as an indirect measure of local population activity[19,20,54]. This frequency range is relatively broad and the simple bandpass-filtering might cause a disproportionate contribution of lower frequencies within the bandpass range to the overall HFA estimate, due to the 1/f nature of EEG power spectra. To avoid this confound, we applied the Ensemble Empirical Mode Decomposition[20,55] (EEMD; https://github.com/leeneil/eemd-matlab.git), to identify the individual characteristic signal modes within the frequency range of interest. Time series representing each individual mode were normalized across time, resulting in a more balanced sampling across the broadband gamma range. Briefly, the EEMD decomposes a non-stationary signal into its elementary components, referred to as intrinsic mode functions[55] (IMFs; Supplementary Fig. 17). The procedure iteratively applies an empirical mode decomposition algorithm, while adding white noise to prevent the mode mixing[55,56]. Using this approach, decomposition output entirely depends on the signal's intrinsic properties, avoiding prior assumptions[20,55,56]. The resulting IMFs captured several canonical spectral features consistently across participants and anatomical structures (Supplementary Table 3). Finally, the HFA time-series on individual channels were reconstructed by summing the channel-specific IMFs with center frequencies > 30 Hz[20].

## Time-frequency representation of the HFA

The instantaneous spectral power at each time-frequency bin was derived from the reconstructed HFA time series ($x$), using a wavelet transform[57,58]. This approach consists of convolving the time series $x$ with a set of Morlet wavelets, parametrized by a range of cycle numbers ($n = 2, 3, ..., 10$) at a given frequency $f$,

$$P_{f,n}(t) = \left| \psi_{f,n} * x(t) \right|, n = 2, 3, \ldots, 10 \tag{2}$$

with $\psi_{f,n}$ defined as

$$\psi_{f,n} = \frac{1}{B_n \sqrt{2\pi}} e^{-\frac{t^2}{2B_n^2}} e^{j2\pi f t}, where\ B_n = \frac{n}{5f} \tag{3}$$

and computing the geometric average ($\hat{P}(f,t)$) of resulting spectral power at each time-frequency bin:

$$\hat{P}(f,t) = \sqrt[9]{\prod_{n=2}^{10} P_{f,n}(t)} \tag{4}$$

This approach results in a high temporal and frequency resolution, facilitating the detection of narrow-band, transient oscillatory events[57,58]. The wavelet center frequencies were within 30–280 Hz range, with 1 Hz increments. The wavelet cycle number range (2–10) is commonly used[59]. To avoid the edge effects, this procedure was applied on the entire individual recording sessions, and the resulting time-frequency response matrices were segmented into trial epochs (starting −1000 msec prior to stimulus onset and ending 1000 msec after the response time). The power within each trial epoch was then normalized by z-transforming each frequency bin and subtracting the

average pre-trial baseline (−1000 to 0 msec, relative to stimulus onset[59]).

## Representational Similarity Analysis (RSA)

The representational similarity was quantified as the Spearman correlation between the HFA power spectral vectors (PSVs), for each combination of the encoding-response time bins from the same trial[21,46,60,61] (Supplementary Fig. 17). Specifically, the instantaneous spectral power at each frequency was estimated for 100 msec time bins (10 msec step size, 90% overlap), producing the time bin-specific power spectrum vectors (PSV), spanning the encoding (2 sec time window after stimulus onset) and post-encoding response (time window after stimulus offset and before button press) periods:

$$\vec{PSV}_{encoding}(t_1) = \left[ z_1(t_1), \ldots, z_{n_f}(t_1) \right]_{encoding} \tag{5}$$

$$\overline{PSV}_{response}(t_2) = \left[ z_1(t_2), \ldots, z_{n_f}(t_2) \right]_{response} \tag{6}$$

Similar to previous studies[21,22,46,60–62], we computed Spearman's correlation as a measure of PSV similarity between the encoding time $t_1$ and response time $t_2$ for each encoded stimulus,

$$r(t_1, t_2) = \frac{Cov\left( rg_{\overline{PSV}_{encoding}(t_1)}, rg_{\overline{PSV}_{response}(t_2)} \right)}{\sigma_{rg_{\overline{PSV}_{encoding}(t_1)}} \sigma_{rg_{\overline{PSV}_{response}(t_2)}}}, t_1 \in [0, 2], t_2 \in [0, RT] \ sec , \tag{7}$$

with $rg$ representing the ranking operator on the vector $\overline{PSV}$, and $\sigma$ the variance of the vector. This produced a trial-specific two-dimensional similarity matrices, containing all the combinations of encoding ($t_1$) and response ($t_2$) time bins (Supplementary Fig. 15c). The correlation coefficients $r$ were then Fisher transformed, with the resulting coefficients following Gaussian distribution. The region-specific (amygdala and hippocampus) similarity matrices were averaged across trials within individual participants (producing the participant/region-specific similarity maps) and used for group-level statistical analysis. The association strength between the post-encoding ripple-locked stimulus similarity and a) stimulus arousal or b) later correct Lure discrimination was compared between the amygdala and hippocampus. First, the regional t-values were computed, based on *within-participant* comparison between the post-encoding ripple-locked stimulus similarity for low- and high-arousal Lure stimuli or for the correctly or incorrectly discriminated Lure stimuli. The regional t-value time courses were then compared using the non-parametric cluster-based permutation test (1000 permutations, $p < 0.05$).

## Ripple-locked stimulus similarity

Stimulus similarity during individual post-encoding time bins was computed by averaging the bin-specific similarity with the encoding period (200 time bins over 2 sec), resulting in a stimulus similarity time series. To obtain the ripple-locked stimulus similarity, we averaged the stimulus similarity within ± 250 msec around the individual ripple peak times, separately for amygdala and hippocampus (Fig. 3a). To avoid the leakage of encoding epoch activity, only the part of the ripple-locked windows non-overlapping with the encoding epoch was used in the ripple-locked analysis. We next tested whether the post-encoding stimulus similarity is locked to ripples (Fig. 3b), by comparing the grand-average ripple-locked stimulus similarity trace with an empirical null distribution obtained from Monte Carlo simulation. Specifically, we circularly randomly jittered the ripple peak times within ± 500 msec window for 1000 times, obtaining an empirical null distribution of stimulus similarity. Circular jittering denotes the method of event time shuffling within a limited time window. In the context of the present study, the time window is defined by the onset of post-

encoding epoch and the offset of subsequent cross-fixation (Fig. 1a). For example, if the ripple occurred 200 msec after the post-encoding onset and the randomly generated shuffled distance is −500 msec, the assigned shuffled ripple timestamp would be −300 msec prior to offset of the cross-fixation epoch. Next, the stimulus similarity trace around jittered timestamps was averaged within participants and the grand average was calculated across participants. The procedure was repeated for 1000 times, resulting in an empirical null-distribution of stimulus similarity. Regional similarity trace windows exceeding 95th percentile of null-distribution were considered windows of statistically significant ripple-locked stimulus similarity.

To test whether the ripple-locked stimulus similarity is associated with stimulus-induced arousal and later discrimination (Fig. 3c), we first derived the ripple-triggered stimulus similarity, a metric taking the time-locked specificity relative to ripple peak time into account. For every stimulus similarity trace around ripple peak time, we circularly jittered the time as the procedure described above. This results in an empirical null distribution of stimulus similarity (i.e., correlation coefficient) for every time point around ripple. We normalized the real stimulus similarity by z-scoring with mean and standard deviation of the null distribution. We referred to the resulting z-value as ripple-triggered stimulus similarity and it follows Gaussian distribution. We quantified the ripple-locked stimulus similarity difference between the high/low arousal and between correct/incorrect Lure discrimination at every time point by t-test, and corrected for the multiple comparisons using non-parametric cluster-based permutation test. Specifically, we performed the group-level comparisons using paired t-test and identified contiguous time bins with the $p < 0.05$, defined as clusters. The t-values within each cluster were summed as the cluster statistics. We created an empirical null distribution by shuffling the conditional trial labels 1000 times where the maximum cluster statistics was identified for each permutation. It was considered as statistically significant if the real t-sum cluster statistics exceeded the 95% percentile of the null distribution. In addition, regional double-dissociation was tested by computing the region-specific (amygdala and hippocampus) t-value time series, obtained by comparing the low vs. high stimulus-induced arousal trials and correct vs. incorrect Lure discrimination trials within-participant. Next the condition-specific t-values were compared separately for each region, between the stimulus-induced arousal and later correct Lure discrimination, using the non-parametric cluster-based permutation ($p < 0.05$; Supplementary Fig. 9). As an additional control analysis, we averaged the post-encoding ripple times at participant level, using response times as reference points. The average post-encoding ripple times were used to compare the event-locked stimulus similarity on the trials not containing post-encoding ripples (non-parametric cluster-based permutation test, p's > 0.05; Supplementary Fig. 11).

### Stimulus-specific representational similarity
The stimulus-specificity of ripple-locked activity in the hippocampus was determined by first computing the similarity between the activity during encoding epoch on i-th trial ($Enc_i$) and post-encoding ripple window (Ripple) on the same trial ($S_{same} = r(Enc_i, Ripple_i)$). Next, we computed the similarity between other stimuli encoding epochs and ($Enc_{1, 2, …n}$) and post-encoding ripple window on the i-th trial ($S_{diff} = r(Enc_{1, 2, …n}, Ripple_i)$, n denoting the number of different stimuli in the experiment). As the $S_{same}$ might be inflated due to temporal proximity between the $Enc_i$ and $Ripple_i$[63,64], we accounted for the difference in average post-encoding epoch similarity ($S_{avg}$) between the same and different trials. Specifically, $S_{avg}$ was defined as the difference in similarity between the encoding epoch and the entire post-encoding window on the same trial ($S_{same\_avg}$) or different trials ($S_{diff\_avg}$). For each individual stimulus,

the stimulus-specific similarity ($S_{spec}$) was defined as following:

$$Sspec = (Ssame − S_{diff}) − (S_{same\_avg} − S_{diff\_avg}) \qquad (8)$$

Next, the $S_{spec}$ was averaged at participant level and the t-statistics was obtained by comparing the participant-averaged $S_{spec}$ with zero. The similarity null-distribution ($S_{shuff}$) was created by shuffling the stimulus identity (same vs. different) 1000 times and obtaining the t-statistics as described above. Finally, the cluster statistics was performed by comparing the t-values obtained from $S_{spec}$ with the distribution of t-values obtained from $S_{shuff}$ (non-parametric cluster-based permutation test; $n = 1000$ permutations, $p < 0.05$; Supplementary Fig. 12).

### Cross-structure joint ripple-locked stimulus similarity
The cross-structure joint ripple-locked stimulus similarity was obtained by calculating the outer product between the structure-specific stimulus similarity traces (hippocampus and amygdala) during post-encoding ripple windows. The resulting joint stimulus similarity matrices were averaged across the individual ripples for each participant, separately for later correctly or incorrectly discriminated trials. To assess the statistical significance of joint cross-structure stimulus similarity, we performed a Monte Carlo simulation to generate an empirical null distribution by circularly jittering the ripple peak times. The stimulus similarity significance was defined as exceeding the 95% percentile of null distribution (Fig. 4a).

### Dual states analyses
Recorded periods were divided into low- and high-theta (3–10 Hz) or gamma (30–250 Hz) periods, based on the participant-specific power median split, resulting in an equal amount of time assigned to each state. The ripple proportion is defined as the proportion of the total number of ripples occurring during an individual state. The ripple proportion comparisons between the low- and high-theta or gamma periods were performed using one-tailed Wilcoxon signed-rank test ($p < 0.05$; Supplementary Fig. 8b).

### Mutual information
Mutual information (MI)[59,65] is a method for quantifying the amount of information shared between the variables of interest. In electrophysiology, MI is applied to test for the presence and directionality of information flow between the multiple time-series. We applied MI to assess the directional influence between the stimulus similarity in amygdala and hippocampus during the post-encoding ripple windows (Fig. 4b). First, the structure-specific stimulus similarity traces from the amygdala and hippocampus were obtained around each ripple event (±250 msec; see ripple-locked stimulus similarity). Next, we calculated the MI between the amygdala and hippocampal memory stimulus similarity traces, using the 200 msec bin size (10 msec step size), covering the ±250 msec window around ripple peaks. For each time bin, the stimulus similarity was binned into 10 bins (with uniform bin count), consistently across the participants and conditions. The MI between the time series X and Y was defined as

$$MI(X;Y) = \sum_{i}^{n}\sum_{j}^{m} p(x_i, y_j) p(x_i, y_j) − \sum_{i}^{n} p(x_i)p(x_i) − \sum_{j}^{m} p(y_j)p(y_j), \qquad (9)$$

where $p(x_i)$ and $p(y_j)$ represented the marginal probability of signals X and Y, $p(x_i, y_j)$ indicated their joint probability, while m and n represented the numbers of stimulus similarity bins for time series X and Y[59,65]. To test the directionality of information flow, we calculated the time-lagged MI by shifting one time series relative to another across all the time bin combinations. The $MI_{AMY→HPC}$ and $MI_{HPC→AMY}$ at individual time bins were defined as the mean of all the subsequent time-lagged MI bins in the other region[59,66]. We defined the MI

directional influence as the significant difference between the $MI_{AMY \rightarrow HPC}$ and $MI_{HPC \rightarrow AMY}$, assessed using Wilcoxon signed-rank test for each time bin. Correction for multiple comparisons was performed using the non-parametric cluster- based permutation test.

## Data availability

The data generated in this study have been deposited in the Zenodo database (https://zenodo.org/records/10082278). Source data are provided with this paper.

## Code availability

The code generated in this study have been deposited in the Zenodo database (https://zenodo.org/records/10082278).

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

## Acknowledgements

The authors thank all the participants for taking part in the study, as well as the nurses, technicians, and physicians at the UCI Epilepsy Unit. This work was supported by NIH UO1NS108916 to J.J.L., NIH U19NS107609-01 to R.T.K. (subcontract to J.J.L.), NINDS RO1NS21135 to R.T.K., NIA R01AG053555 to M.A.Y. and NIMH R01MH128306 to M.A.Y.

## Author contributions

Conceptualization: H.Z., I.S., J.J.L., R.T.K., M.A.Y. Methodology: H.Z., I.S., S.M., J.J.L., R.T.K. Investigation: H.Z., I.S., J.J.L., R.T.K. Resources: J.J.L, M.P., L.M.; Visualization: H.Z., I.S.; Funding acquisition: J.J.L., R.T.K., M.A.Y. Project administration: J.J.L. Supervision: J.J.L., R.T.K. Writing—original draft: H.Z., I.S.; Writing-review and editing: H.Z., I.S., J.J.L., R.T.K., M.A.Y.

## Competing interests

The authors declare no competing interests.
