## [Peer Review File · Nature Communications]

Awake ripples enhance emotional memory encoding in the human brainReviewer comments, first version:

Reviewer #1 (Remarks to the Author: Overall significance):

Zhang et al. investigated the role of awake ripples for encoding of emotional memories. The authors recorded intracranial EEG in 7 patients from the amygdala and hippocampus. They show that arousing stimuli elicit more ripples and lead to better memory performance in a later test. Furthermore, the authors show that ripples correlated with higher neural measures of stimulus specific similarity with a different time course for amygdala and hippocampus. Finally, the authors show a directional influence of stimulus similarity from the amygdala to the hippocampus. Together, these findings draw a potentially interesting and fine grained picture of how ripples mediate emotional memory encoding.

The paper has several strengths but also a couple of weaknesses which I summarize below.

Strengths: Simultaneous recordings of activity in amygdala and hippocampus. Sophisticated analysis of electrophysiological signals. Correlation of ripples and neural similarity metrics with behaviour. Directional measures of information based connectivity metrics. Coherent set of results. Weaknesses: Low number of patients, i.e. out of the 7 patients only 6 appeared to have yielded usable data. Ripple detection algorithm was based only on ripples only, but appeared to not take into account sharp-waves. Relation between ripples and behaviour could be spuriously driven by arousal. Direct contrast between hippocampal and amygdala similarity traces is missing which is necessary to support double dissociation claim (Figure 3c).

Reviewer #1 (Remarks to the Author: Strength of the claims):

Major concerns

1. The number of patients that contributed data to the ripple based analysis is 6 which is quite low, even for an intracranial EEG study. I do appreciate that the authors show individual subject data, and I also appreciate that a study that records from multiple regions simultaneously will often have low subject numbers but nevertheless an N=6 does seem a bit low for a study of this scope. If the authors would have additional data, or even if the authors would have multiple sessions in that same patients and show that the effects are reliable across sessions that would alleviate that concern a bit.
2. The ripple detection algorithm was based only on the ripple-band filtered data, and hence did not take into account the sharp-wave component of the ripple. To establish the common knowledge, a sharp-wave ripple is composed of two parts (i) the ripple and (ii) the sharp-wave (reflecting a strong input of CA3 to CA1; see Buzsáki 2015 Hippocampus). Therefore, if the authors wish to refer to sharp-wave ripples, evidence for such a sharp-wave needs to be presented or the ripple detection algorithm needs to be adjusted to take into account the sharp-wave (i.e. by checking for an offset at the start of the ripple).
3. Somewhat related to the above point, there is a concern to which extent the ripples detected here are truly physiological ripples or spurious high-frequency events. The authors do present evidence that the ripples are frequency specific and that they occur during periods of low theta which is reassuring, but they do not present evidence that the number of detected events is actually meaningful in a statistical sense. The problem is that the algorithm looks for a specific pattern (i.e. crossing of a threshold in a band-pass filtered signal). Therefore the algorithm would also detect a number of events in a noise pattern, i.e. a simulated signal containing only Brownian noise but no 'true' ripple events. Can the authors show that the number of ripples detected in the physiological data exceeds the number of ripples detected in a simulated signal that does not contain ripples?
4. Since the number of ripples is correlated with both memory performance and arousal it appears possible that the relation between memory and ripples is spurious. For instance, arousal may simply lead to a higher amount of ripples and also better memory performance, but ripples and memory may be totally unrelated. To this end it would be reassuring to see higher rates of ripples for correct vs incorrect memory trials controlled for the level of arousal.

5. The authors report a double-dissociation between amygdala and hippocampus whereby amygdala dissociates between high and low arousing stimuli and hippocampus dissociates between correct and incorrect memory trials. However, this claim is based only on single contrasts that were carried out separately for the amygdala and the hippocampus. For a double dissociation, evidence needs to be shown by way of direct contrasts, that is showing that the difference between high and low arousal trials in the amygdala is stronger than the difference between correct and incorrect trials, and vice versa for the hippocampus (see <https://elifesciences.org/articles/48175>).

Other comments/concerns

6. I was not convinced that the term reinstatement is warranted here. I understand that the analysis was done at the time immediately following the encoding period, where the stimulus is off-screen but it still is in working memory. Therefore, I wonder if anything needs to be re-instated when it hasn't been out of mind. Maybe it would be better use a different term, i.e. post-encoding stimulus similarity.

7. What is circular jittering? Please explain in the methods section.

8. I did not quite understand how different electrodes and time-stamps for SWRs were handled. For instance, where similarity values and SWRs collapsed across electrodes such that an SWR from a hippocampal electrode was used a time stamp for similarity in an electrode in the amygdala? Or were anatomical regions for SWR time stamps and similarity traces used consistently (i.e. hippocampal SWRs for hippocampal similarity, and amygdala SWRs for amygdala similarity).

9. For the similarity analysis Empirical Mode Decomp was used. Why? Wouldn't a simple high-pass filter suffice to reject low frequency components? I understand that the authors write in the methods section that they used EMD to avoid low frequency harmonics, but I don't understand why that would be a problem since the authors are only interested in stimulus specific frequency patterns. I don't think this is critical but I would be interested in a more elaborate justification.

10. Lines 393-394: A Wilcoxon test is a non-parametric test.

11. Lines 492-493: To avoid discontinuities between epochs, epochs were zeroed at beginning an start times. However, this introduces discontinuities itself unless some sort of a windowed approach was used (i.e. Hanning window).

12. For RSA time-bins of 100ms were used (line 550), for Mutual Information time-bins of 200ms (line 631) were used. What is the justification of these different parameters? I would feel more comfortable if these parameters were held constant.

Reviewer #1 (Remarks to the Author: Reproducibility):

To enhance reproducibility the authors should make their code used for the analysis publicly available.

Reviewer #2 (Remarks to the Author: Overall significance):

In this manuscript, Zhang and colleagues report an iEEG study (n=7) recording from the hippocampus and the amygdala during an emotional memory paradigm. They find that arousing events are better remembered and are accompanied by higher ripple numbers during the immediate post-encoding window. Reinstatement analyses suggest what encoding activity is reactivated during post-encoding ripples, with a potentially different role of the hippocampus and the amygdala in this process.

The topic of the paper is interesting and timely, but unfortunately the conclusions are not supported by the statistics, which are flawed in many instances.

The manuscript lacks reference and discussion of highly relevant iEEG/ripple papers published recently, e.g. work from Zaghoul and Malach and Halgren labs for wake ripples and work from the

Fell group on ripples in hippocampus and amygdala.

Reviewer #2 (Remarks to the Author: Strength of the claims):

Major concerns:

- 1) It is unclear whether post-encoding ripples are driven by memory processes or by stimulus-induced arousal. It is likely that these two factors are strongly correlated and the authors would need to disentangle their respective contributions, e.g., via partial correlations or holding one factor constant while examining the impact of the other.
- 2) To make the claim that ripples were predictive of discrimination performance in the post-encoding phase and not the peri-encoding phase, a significant Phase x Discrimination Performance interaction should be demonstrated.
- 3) The statistics in Fig 2c are not convincing. No correction for multiple comparisons is performed. Additionally, the response times post-encoding for high vs low arousal are different. The number of ripples may be higher in high arousal because the window is larger (a mean difference of 200 ms makes the average time window for high arousal trials ~30% larger). The authors should compare ripple rates rather than the raw number of ripple events.
- 4) Likewise, their 'region-specific double dissociation' is no double dissociation. They merely found effect A in region 1 and effect B in region 2. A double dissociation warrants a significant 2x2 interaction. That is, they would need to show that effect A is significantly greater in region 1 than in region 2 and that effect B is significantly greater in region 2 than in region 1.
- 5) The authors state 'A aSWR-locked joint memory reinstatement was present during the post-encoding period only for correctly discriminated stimuli'. To establish the important link between reinstatement and memory, they would need to show that reinstatement is significantly stronger for successful than unsuccessful encoding.
- 6) Overall, the reinstatement results (Fig 3) are not convincing:
 - a) The correlation is not driven by a specific/circumscribed encoding time window, but based on numerous short intervals, which is at odds with established encoding/retrieval reinstatement findings.
 - b) The effects of comparing low/high arousal and successful/unsuccessful memory appear to be driven by decreases in similarity for low arousal and unsuccessful encoding, rather than by an increase for high arousal and successful memory, respectively.
 - c) Apart from shuffling ripple times, an important control would be to shuffle encoding trials in order to ensure reinstatement is event-specific.
 - d) The time window around the ripple event in Fig 3a overlaps with the peri-encoding stage (-250-0 ms). To claim reinstatement of peri-encoding patterns during post-encoding, data from the peri-encoding phase prior to a ripple in post-encoding cannot be treated as the post-encoding phase.
 - e) The authors cannot claim "temporal significant MI difference (AMY→HPC) is present before aSWR peak time (-70 to -30 msec)" in Fig 3e, since cluster-based permutation analysis with cluster-mass is unable to support claims of specific temporal and spatial significance, especially on these time-scales. The same applies to claims of temporal reinstatement profiles in Fig 3b.
- 7) Figure 2c appears to show the absolute number of ripples, which seems identical for the 2 s encoding period and the ~ 0.7 s response period. Ripple occurrences would need to be converted into ripple rates (occurrence/time; see comment 3). On that note, it would be important to explicitly state how many trials had one or more ripples during the response window and how the authors speculate memory processes occur on trials without ripples. Importantly, for the 'aSWR-locked joint memory reinstatement' analysis, are there sufficient trials in which both regions show a ripple?

Other comments:

- 1) How many analysed channels are from the hippocampus vs amygdala? It would be useful to see selected channels for analysis highlighted in Fig 2a.
- 2) Why was 150 events chosen as a cut-off to include channels? Did this result in the exclusion of the 7th participant? If so, what effect did that have on results? In a study with so few participants each excluded participant potentially has a large influence on an effect that is already fragile.
- 3) The statement, "The aSWRs probability was significantly higher during low theta power periods (Extended Data Fig 5), consistent with observations that cholinergic tone promotes theta oscillations and suppresses SWRs" comes out of the blue and does not add to the current question.
- 4) Colour maps should be symmetrical (e.g., Fig 3a).
- 5) One-tailed testing in Fig S9 is not justified.
- 6) The authors may want to stick to SWR and omit the 'a' – there is now a substantial body of literature using SWR for both wake and sleep ripples and new acronyms are somewhat confusing.

Reviewer #2 (Remarks to the Author: Reproducibility):

The authors should consider sharing the code required to reproduce the figures and statistics.

Reviewer #3 (Remarks to the Author: Overall significance):

Here the authors show that the occurrence of awake sharp-wave ripples in humans is correlated with both the arousal triggered by a visual stimulus and its later successful recollection. Further, they show that the joint reinstatement of hippocampal and amygdala intracranial EEG patterns around the time of the ripple is correlated with the successful recollection of emotional stimuli, suggesting a role for SWR in the preferential encoding of arousing stimuli. This is a new and important finding since we know that emotional valence influences memory formation but, to date, the physiological mechanisms by which this could be mediated are relatively unknown.

Reviewer #3 (Remarks to the Author: Impact):

To my knowledge, this is the first paper that explores the question of encoding and retrieval of emotional material using intracranial recordings in humans. It reveals a potential mechanism for the preferential consolidation of emotional memories and paves the way for future studies in human and animal models.

Reviewer #3 (Remarks to the Author: Strength of the claims):

Overall, the work is convincing as presented, although it might lack statistical power due to the scarcity of data (short trials, few ripples, few patients). However, I think this should be considered in light of the difficulty to obtain intracranial human EEG data, and acknowledged as an inevitable drawback that doesn't affect the potential and novelty of the results. I have however a few analysis points that I would like to see addressed.

Major points :

1. I am unfamiliar with the reinstatement method the authors use. However, in the reinstatement analysis for the hippocampus, I am worried about a circular confound: indeed, the similarity measure is calculated over a range of high frequencies that include the ripple range, and then the correlation is calculated at times surrounding pre-detected ripples: won't the correlation coefficient necessarily be high around the ripple times? If so, this invalidates the rest of the results that use this reinstatement measure, for example, the joint reinstatement in Fig 3d.

Minor points :

1. Given the data that is available for analysis in the response time window (less than a second), the data for SWR occurrence is binary (ie either one or zero SWR). I'd suggest it appears more clearly how the authors have accounted for this specificity in their analysis and statistics.
2. It is unclear whether 6 or 7 subjects have been used since 1 subject has been discarded for having too few aSWR. Was the HF data still analyzed in this patient, although most analyses are restricted to SWR occurring time?
3. In figure 3, the significance of the reinstatement has been calculated for all trials with a SWR (a) but when separating trials according to arousal or correctness, the significance of the reinstatements haven't been tested against the null, shuffled hypothesis (but only between the 2 types of trial). I think this is missing.
4. I do not understand the point of the compression analysis in the case of a time-frequency analysis of EEG. My guess is the idea is coming from the fact that neuronal replay during SWR is compressed, which can be calculated in the case of a sequence of activation of individual neurons (duration of replay). How can this apply to iEEG data? Indeed, the authors find that any compression lowers the reinstatement measure, which might be because in that case compressing is just adding noise and has no "physiological" basis. If I am wrong, I would actually be interested in the explanation.

Analysis or examples that would strengthen the claims :

1. Provide a better description of the amygdala activity overall, and more specifically around the time of hippocampal SWR. A recent paper claims that there are "ripples" in the human amygdala during Non-REM sleep (Staresina 2020) and that they coordinate with hippocampal SWR. Have the authors observed such typical HFOs in their data? And therefore, could the joint pattern reinstatement be mediated by a more clearcut amygdala pattern than the wideband HFA? This could be also discussed.
2. Trials are separated a priori according to the 3 mean criteria (valence, arousal, success) and the data analyzed separately for these (Fig 3d, correct/incorrect). It would be interesting if the hippocampus, amygdala or joint responses can predict the success of the response. Trials without SWR could also be included in this analysis that is more "agnostic", and thus, in my eyes, complementary.
3. The authors chose to focus almost exclusively their analysis on the trials with SWR, and use control shuffling within these trials. I see a potential issue with shuffling the ripple time in such a short window (ie response time under 800ms, and considered time window for peri-ripple analysis 500ms!). I suggest making use of the non-SWR trials to develop the analysis and potentially the claims (is there an anti-correlation between the success of the trial and the intensity of the theta power for example?) and also to use as a control. A possibility would be to analyze those non-SWR trials "as if" they were SWR trials. The "fake" event time used to align the correlation coefficient could be the average time at which the ripple occurs in SWR trials. Also, theta power provides you with a non-binary measure (unlike SWR occurrence) with which to correlate other parameters.

Reviewer #3 (Remarks to the Author: Reproducibility):

The paper, including the methods, is very clearly written and would allow for a reproduction of the work should a researcher get access to this type of recordings in humans.

1. Increase the size of the axis and legends in the figures. Some of them are almost unreadable even when zooming in.

Reviewer #4 (Remarks to the Author: Overall significance):

In the current manuscript, Zhang et al. examined whether sharp-wave ripples in the amygdala and hippocampus at various stages of memory processing relate to accurate memory discrimination of emotional stimuli. They report that higher post-encoding aSWR count was related to stimulus-related arousal but not valence. This neural measure was also related to successful memory discrimination and were distinct from non-specific broadband fluctuations in power. The results also show a dissociation in the relationship between the memory reinstatement processes in the amygdala and hippocampus, with the former relating to stimulus-linked arousal and the latter relating to accurate memory discriminations. Finally, post-encoding amygdala reinstatement processes preceded hippocampal reinstatement by ~100ms and this pattern was related to

subsequent memory discrimination, suggesting a directional influence of the amygdala over hippocampal memory reactivation.

I found the results interesting and consistent with prior findings regarding interactions between the amygdala and hippocampus and emotional memory formation. The examination of SWR's represents an important extension of existing work both in rodents (during sleep) and in human work on post-encoding patterns of hippocampal/amygdala connectivity related to emotional memory. The current study helps shed new light on the functional role of awake SWR's in the prioritization of emotional memories, which will be of broad interest to many readers who study learning and memory.

However, the scholarship could be improved by choosing more appropriate references. Namely, the link between some citations is over-generalized and isn't the best reflection of a given datapoint/claim. One example is the statement, "Stimulus-induced arousal (irrespective of valence) was associated with better memory discrimination, confirming previous reports¹⁻³". Two of these are review papers on broad emotional memory effects (not discrimination, per se), and the only empirical one (Szollosi and Racsmany, 2020) is reporting differences in lure discrimination index, which accounts for response biases (see Comment #1); the LDI has, as of yet, not been computed here. In addition, the statement, "...consistent with observations that cholinergic tone promotes theta oscillations and suppresses SWRs^{10,12}" is a big leap and has no grounding in this dataset. The noradrenergic system also plays a similar function and has been heavily implicated in emotional memory processes (e.g., Brown et al., 2005; Sara, 2009). There is no direct evidence in the current study linking these processes to the cholinergic system, so this should be removed or at least moved to the discussion along with a broader consideration of other neuromodulatory candidates. It would also be helpful to cite related work, such as Inman et al. (2018), that shows a relationship between post-encoding amygdala stimulation and memory enhancements.

Reviewer #4 (Remarks to the Author: Impact):

I do think this paper is a strong theoretical contribution to the field of learning and memory. However, my opinion does hinge on additional analyses that will help clarify some of the results. It will also be important for put forth a stronger/clearer interpretation of what post-encoding process is being supported by aSWR in the service of subsequent discriminations (i.e., is about a high fidelity item representation? context? etc.)

Reviewer #4 (Remarks to the Author: Strength of the claims):

1) Emotional memory discrimination paradigms often compute a 'lure discrimination index' (LDI) to correct for potential response biases in old/new memory decisions, such as an over tendency to simply endorse most items as 'New' (e.g., Leal et al., 2014). Here, the commonly reported/used behavioral outcome measure is "correct discrimination", which appears to collapse accurate responses across all three trial types. Please compute the LDI for each subject/emotional valence and report the statistical results so that the current results can be generalized to the broader emotional memory literature.

2) It is unclear when the term "correct/accurate discrimination" specifically refers to Lure trials or to any accurate discrimination ('Old' for Repeats and 'New' for Novel/Lures, as in Figure 1B). This makes it challenging to unpack some of the findings and should be made very clear throughout.

3) If some of the aSWR analyses refer to discrimination accuracy across multiple trial types (presumably Repeat + Lure, because Novel items from encoding don't repeat in phase): Mnemonic discrimination studies use the LDI to index the process of pattern separation, whereas old-item recognition is more often used to index pattern completion. Collapsing these measures together by only assessing "accurate discrimination" as one bin (if indeed correct responses for Repeat + Lure trials are combined) might therefore obscure two opposing memory processes, making it unclear what operations are occurring in the amygdala and hippocampus at retrieval. That could potentially account for null retrieval effect. It is also unclear what the post-encoding period reflects about the quality of memory consolidation. Are SWR's capturing a representation in high fidelity to aid a subsequent discrimination? Why is this process supporting discrimination as opposed to other recognition processes? The specific behavioral significance of the post-encoding aSWR's should be discussed in more detail.

4) The logistic regression results in Figure 1C help reveal how emotional parameters relate to lure discrimination. It would be helpful to perform this same analysis for the 'Repeat' trials to demonstrate that there is: a) specificity to rejecting lures, and b) not driven by response biases. Hypothetically, it could be the case that these results simply reflect endorsements of 'New' during the memory test, in which case it would be important to verify that you don't see the same pattern for incorrect discriminations for 'Repeat' trials.

5) One of the major take-homes from this study is that aSWR's relate to enhanced memory for emotional stimuli. Yet this conclusion isn't fully supported by the current analyses, which show separate main effects for successful memory discrimination and stimulus-related arousal. Although arousal and memory discrimination are correlated, the current results cannot speak to whether these mechanisms/processes simply co-occur and are independent of each other. To argue that these neural processes relate to emotional memory, it would be necessary to identify a memory-by-arousal interaction effect using 2 (Memory Discrimination: successful, unsuccessful) x 2 (Arousal: high, low) repeated-measure ANOVA's. I recommend this approach be used wherever possible in the analyses, including those with valence. If there is also a motivation to make specific predictions regarding lure discrimination versus recognition, then trial type (Type: repeat, lure) should also be modeled.

6) From a conceptual perspective, is it accurate to call the valence rating phase "retrieval"? The rating happens immediately after stimulus presentation, which suggests this is a working memory process (which has also been linked to aSWR's; Jadhav et al., 2012). Further, participants may have already made their valence decisions during image presentation, because they are aware of this instruction and are exposed to many trials. The claim that these neural processes index a retrieval mechanism needs to be clarified and/or alternative mechanisms (e.g., WM) should be considered. The prior work that is cited for post-encoding reactivation (e.g., Sols et al., 2017 and Ben-Yakov et al., 2013) involves a task switch (and perceptual) or a blank screen. Neither of these explicitly pertain to a judgement about the preceding image/stimulus, so the nuances here might matter.

7) Although the RT's didn't differ by arousal or discrimination, it still seems like variability in trial-level RT's need to be addressed. I encourage the authors to use SWR rate (ripple per unit time) instead of a raw count to normalize the number of SWR events based on the actual response window of a given trial (if the goal is to isolate a decision-making process). Furthermore, please report whether the RT's differed by emotional valence.

8) The logistic regression (Fig. 1C) should include all two-way and the three-way interaction term for similarity, valence, and arousal. Similarity is disregarded in all the main brain analyses, but it would be useful to demonstrate that there is no significant interaction with similarity to justify collapsing high and low similarity in the arousal, valence, and discrimination analyses. The results in Figure 1D qualitatively suggest that there might be an interaction; regardless, how do the authors interpret the finding that lure discrimination is best for high arousal trials that are low vs. high in similarity? This seems to be an important effect and warrants more scrutiny.

9) Stimuli. Please report more information about the image stimuli, including the database they were selected from. Furthermore, it seems that the normative emotion ratings were based on a previous study (Leal et al., 2014). If this is true, please cite that paper. One additional issue is that the raters in Leal et al. (2014) reported negative stimuli to be significantly more arousing than positive stimuli; this may confound any interpretations related to the effects being purely driven by arousal vs. valence (i.e., it may be the case that the reported effects are a mixture of both, with arousal being the prevailing factor). This limitation should be addressed in the methods or discussion.

10) Extended Data Figure 1. It's unclear whether these ratings are from the additional samples or from the subjects in the current study. Please specify. It's also unclear in the main text whether the trial types (e.g., defined by valence) are based on the normative ratings or ratings from the current subjects - though it seems to be the former. I'd encourage the authors to instead define the conditions based on the actual subjects' ratings from this study as opposed to the normative rating-related categorizations, especially given that some stimuli do not fit clearly within the predefined categories (e.g., several neutral datapoints would be classified as Positive based on their valence ratings in the Figure).

Reviewer #4 (Remarks to the Author: Reproducibility):

NOTE: Most of my comments related to statistical analyses and appropriateness are described in the "Strengths..." section.

1) Participant exclusions. It is stated that "due to the low number of detected aSWRs, one subject was eliminated from the aSWR-related analysis." What was this number/percentage? In addition, it's stated that "only the subjects with the correct discrimination rate of Novel trials $\geq 85\%$ " were included in the analyses. What was the basis for this exclusion criterion? Was this subject's performance > 2.5 SD's below the mean? It seems arbitrary to make this exclusion based on a single trial type, and I think it would be more appropriate to only exclude this participant if their performance is significantly below the average performance across all trial types (i.e., indicative of poor task performance overall). The specified performance threshold is also very high; please report the reasoning behind this exclusion, the performance value of the excluded subject for Novel trials and cite any relevant literature to motivate this decision. Without sufficient justification, this participant should be included in all analyses.

2) Minor: the paneling in Figure 3 is a little confusing and could benefit from some reorganization (for example, organizing a horizontally as opposed to vertically). Panel E looks like an extension of C. Additionally, the "Beta" label on the Y-axis of Figure 1 could be more informative. The colored diagram to the right is helpful but beta is a very non-descriptive term.

Author rebuttal, first version:

Reviewer #1 (Remarks to the Author: Overall significance):

Zhang et al. investigated the role of awake ripples for encoding of emotional memories. The authors recorded intracranial EEG in 7 patients from the amygdala and hippocampus. They show that arousing stimuli elicit more ripples and lead to better memory performance in a later test. Furthermore, the authors show that ripples correlated with higher neural measures of stimulus specific similarity with a different time course for amygdala and hippocampus. Finally, the authors show a directional influence of stimulus similarity from the amygdala to the hippocampus. Together, these findings draw a potentially interesting and fine grained picture of how ripples mediate emotional memory encoding.

The paper has several strengths but also a couple of weaknesses which I summarize below.

Strengths: Simultaneous recordings of activity in amygdala and hippocampus. Sophisticated analysis of electrophysiological signals. Correlation of ripples and neural similarity metrics with behaviour. Directional measures of information based connectivity metrics. Coherent set of results.

Weaknesses: Low number of patients, i.e. out of the 7 patients only 6 appeared to have yielded usable data. Ripple detection algorithm was based only on ripples only, but appeared to not take into account sharp-waves. Relation between ripples and behaviour could be spuriously driven by arousal. Direct contrast between

hippocampal and amygdala similarity traces is missing which is necessary to support double dissociation claim (Figure 3c).

Reviewer #1 (Remarks to the Author: Strength of the claims):

Major concerns

1. The number of patients that contributed data to the ripple based analysis is 6 which is quite low, even for an intracranial EEG study. I do appreciate that the authors show individual subject data, and I also appreciate that a study that records from multiple regions simultaneously will often have low subject numbers but nevertheless an N=6 does seem a bit low for a study of this scope. If the authors would have additional data, or even if the authors would have multiple sessions in that same patients and show that the effects are reliable across sessions that would alleviate that concern a bit.

Response: We thank the Reviewer for acknowledging the importance and novelty of the present study. We agree that the number of subjects is relatively low, due to paucity of subjects with simultaneous hippocampal and amygdala recordings, outside of seizure zone. While the higher number of subjects enhances the confidence in study interpretation, a number of recent high profile human intracranial studies focused on ripples had similar or lower numbers of subjects (e.g. Vaz et al, 2020, n = 6; Dickey et al., 2021, n = 4). We acknowledge this limitation in the revised manuscript, while also presenting the individual subjects data (Fig. 1c, 2c and Supplementary Fig. 2a, b, 3, 6, 8b, 9, 14), which support the study conclusions. This answer is also related to question by Reviewer #3 (Remarks to the Authors).

Page 13, Lines 8 - 12

Discussion

The number of subjects in the present study is relatively low, due to paucity of subjects with simultaneous amygdala and hippocampus recordings outside of seizure onset zone. However, the effects are consistent across the individual subjects (Fig. 1c, 2c and Supplementary Fig. 2a, b, 3, 6, 8b, 9, 14), mitigating the possibility of outliers affecting the study conclusions.

2. The ripple detection algorithm was based only on the ripple-band filtered data, and hence did not take into account the sharp-wave component of the ripple. To establish the common knowledge, a sharp-wave ripple is composed of two parts (i) the ripple and (ii) the sharp-wave (reflecting a strong input of CA3 to CA1; see Buzsaki 2015 Hippocampus). Therefore, if the authors wish to refer to sharp-wave ripples, evidence for such a sharp-wave needs to be presented or the ripple detection algorithm needs to be adjusted to take into account the sharp-wave (i.e. by checking for an offset at the start of the ripple).

Response: We agree that the presence of a sharp-wave component (typically >100 uV amplitude) is one of defining features of sharp-wave/ripples, as established by the rodent hippocampal recordings. However, the presence of a sharp-wave component depends on the exact electrode position relative to hippocampal layers, as illustrated by the Fig. 1b in Ramirez-Villegas et al. (2015, see references). While the depth electrode position in rodent/non-human primate brains could be optimized with the sub-millimeter precision at any time point, post-implantation electrode position adjustment in the human brain is currently not possible. This is a likely explanation for the absence of pronounced sharp-waves in the present study (Supplementary Fig. 2b), and in other published examples of human ripple waveforms (e.g. Staresina et al., 2015, Fig. 4a; Norman et al., 2021, Fig 1g). Moreover, the ripple detection methodology in the present study is based on the established criteria in rodents and humans (double threshold of z-scored rectified ripple-range envelope; reviewed by Liu et al., 2022), which typically doesn't include sharp-wave detection. As the presence of 'sharp-wave' notation without the explicit attempt for sharp-wave detection might be misleading, we changed the detected events notation from 'sharp-wave/ripple' to 'ripple'. This point is clarified in the revised version of the manuscript.

Page 46, Lines 29-33

Results

The prominence of the sharp-wave component in the ripple waveform depends on the optimal electrode position in the hippocampal layers (Ramirez-Villegas et al., 2015). As the electrode position in human subjects can't be optimized post-surgically, this likely accounts for the absence of prominent sharp-wave component in the ripple waveform in this study, similar to other published examples of human ripples (e.g. Staresina et al., 2015; Vaz et al., 2020; Norman et al., 2021; Sakon and Kahana, 2022).

3. Somewhat related to the above point, there is a concern to which extent the ripples detected here are truly physiological ripples or spurious high-frequency events. The authors do present evidence that the ripples are frequency specific and that they occur during periods of low theta which is reassuring, but they do not present evidence that the number of detected events is actually meaningful in a statistical sense. The problem is that the algorithm looks for a specific pattern (i.e. crossing of a threshold in a band-pass filtered signal). Therefore the algorithm would also detect a number of events in a noise pattern, i.e. a simulated signal containing only Brownian noise but no "true" ripple events. Can the authors show that the number of ripples detected in the physiological data exceeds the number of ripples detected in a simulated signal that does not contain ripples?

Response: We agree that demonstrating the higher ripple detection in hippocampal signal, relative to synthetic signal, would be a useful ripple detection quality control.

To address this question, we applied the ripple detection algorithm on the synthetic signal of the same spectral characteristics as the corresponding hippocampal channels (see Methods description below). Ripple detection in hippocampal channels exceeds the event detection in subject-specific synthetic signals in all the subjects used in the ripple-based analysis (subjects 2-7; Supplementary Fig. 14). The only exception was subject 1, excluded from the ripple-based analysis in the original submission, due to the low ripple detection (z-score < -2, relative to distribution across the subjects). Therefore, the two complementary approaches (low ripple detection relative to other subjects and the subject-specific synthetic signal) justify exclusion of this subject from ripple analysis, considering the recommendations for cautious approach to ripple detection in the human brain (Liu et al., 2022).

Page 47, Line 18 - Page 48, Line 2

Methods

Ripple detection from synthetic signal

The power spectral density (PSD) was calculated for each hippocampal channel used in ripple detection. Next, a channel-specific filter was designed and applied on a random time series following Gaussian distribution, resulting in a synthetic signal with identical PSD as the hippocampal signal. Specifically, the synthetic signal was first transformed to frequency domain by N-point Fourier transform (N denoting the number of datapoints in the signal), followed by multiplication of resulting spectrum by the channel-specific PSD coefficients and application of inverse Fourier transform, to convert the signal back to time domain. The resulting synthetic signal mimicked the hippocampal channel-specific spectral characteristics. Next, the ripple detection procedure (see Ripple detection) was applied on the synthetic signal. Finally, as an additional control, the numbers of detected events were compared between the hippocampal iEEG signal and channel-specific synthetic signal. In all the 6 subjects used in the ripple-based analysis (subjects 2-7), the numbers of detected ripples were higher than the numbers obtained from the subject-specific synthetic signals (Supplementary Fig. 14). Subject 1 was excluded from the analysis based on the two criteria: a) low number of detected putative ripple events (z-score < -2, relative to distribution across the subjects) and b) the number of detected events lower than chance level, defined as the number of detected events in synthetic signal (Supplementary Fig. 14).

Page 37, Lines 1 - 9

Supplementary Fig. 14. Ripple detection in the hippocampal signal and synthetic signal. Comparisons between the numbers of ripples detected in hippocampal channels (left) and number of events detected in subject-specific synthetic signals of the same spectral characteristics (right). In all the subjects included in ripple analysis (subjects 2-7) ripple detection in hippocampal signal was higher than in synthetic signal. Subject 1 was excluded from the ripple-based analysis due to low ripple detection in hippocampal signal (z -score < -2 , relative to distribution across the subjects) and the number of detected putative ripples lower than in synthetic signals of the same spectral characteristics.

4. Since the number of ripples is correlated with both memory performance and arousal it appears possible that the relation between memory and ripples is spurious. For instance, arousal may simply lead to a higher amount of ripples and also better memory performance, but ripples and memory may be totally unrelated. To this end it would be reassuring to see higher rates of ripples for correct vs incorrect memory trials controlled for the level of arousal.

Response: As the post-encoding ripple occurrence is associated with both the stimulus arousal level and later correct Lure discrimination (Fig. 2), we agree it would be important to disentangle these effects. The stimulus arousal modulated magnitude of association between the ripple occurrence and later correct Lure discrimination (Supplementary Fig. 4). Specifically, the difference in post-encoding ripple occurrence between the correctly and incorrectly discriminated Lure stimuli was significantly higher for high-arousal Lure stimuli ($p = 0.007$, chi-square = 7.38; Friedman's two-way ANOVA). In addition, association between the post-encoding ripple occurrence and later correct Lure discrimination was significant for both low- and high-arousing stimuli (Supplementary Fig. 4; low-arousal : $p = 0.046$, $z = -1.68$;

high-arousal : $p = 0.023$, $z = -2.01$; one-tailed paired Wilcoxon test, $p < 0.05$). We elaborate on these results and possible interpretations in the revised manuscript. This answer is also related to Q1, Reviewer #2 and Q5, Reviewer #4.

Page 43, Lines 2 - 5

Methods

The strength of association between the correct Lure discrimination and post-encoding ripple occurrence was compared for low- and high-arousing stimuli using the non-parametric Friedman's two-way ANOVA ($p < 0.05$; Supplementary Fig. 4).

Page 5, Lines 21 - 29

Results

As the stimulus arousal and correct Lure discrimination are both associated with post-encoding ripple occurrence (Fig. 2c), we tested if the association between the ripple occurrence and correct Lure discrimination is modulated by the stimulus arousal level. The magnitude of post-encoding ripple modulation based on the correct Lure discrimination was stronger for high-arousal, relative to low-arousal stimuli (Supplementary Fig. 3; $p = 0.007$, $\chi^2 = 7.38$; Friedman's two-way ANOVA). Ripple occurrence was associated with correct Lure discrimination for both arousal levels (low-arousal: $p = 0.046$, $z = -1.68$; high-arousal: $p = 0.023$, $z = -2.01$; one-tailed paired Wilcoxon test, $p < 0.05$).

Page 10, Lines 12 - 14

Discussion

Significant association between the post-encoding ripples and later correct Lure discrimination was present regardless of the stimulus arousal level (Supplementary Fig. 4), but the effect was stronger for high-arousal stimuli.

Page 10, Lines 20 - 23

Post-encoding stimulus similarity (or reinstatement) is implicated in memory consolidation (Carr et al., 2011; Ben-Yakov et al., 2013; Sols et al., 2017; Schreiner and Staudigl, 2021) and peaks during ripples (Fig. 3). Thus, beside the general role of ripples in memory consolidation, these results also imply ripples as a potential mechanism mediating the effects of arousal on memory consolidation.

Page 27, Lines 1 - 8

Supplementary Fig. 4. Stimulus arousal level modulates the association between the post-encoding ripples and later correct Lure discrimination. The post-encoding ripple modulation by correct Lure discrimination was significantly stronger for high-arousal stimuli ($p = 0.007$, $\chi^2 = 7.38$; Friedman's two-way ANOVA). The association between the post-encoding ripples and later correct Lure discrimination was present regardless of the Lure stimulus arousal level (low-arousal: $p = 0.046$, $z = -1.68$; high-arousal: $p = 0.023$, $z = -2.01$; one-tailed paired Wilcoxon test).

5. The authors report a double-dissociation between amygdala and hippocampus whereby amygdala dissociates between high and low arousing stimuli and hippocampus dissociates between correct and incorrect memory trials. However, this claim is based only on single contrasts that were carried out separately for the amygdala and the hippocampus. For a double dissociation, evidence needs to be shown by way of direct contrasts, that is showing that the difference between high and low arousal trials in the amygdala is stronger than the difference between correct and incorrect trials, and vice versa for the hippocampus (see <https://elifesciences.org/articles/48175>).

Response: We apologize for the unclear definition of regional double dissociation in the original manuscript. This term was meant to denote that the ripple-locked activity in the hippocampus and amygdala are associated with different task aspects - later correct Lure discrimination and the stimulus arousal level, respectively (Fig. 3c). This would be reflected by the different strengths of the association between ripple-locked

activity and arousal or correct Lure discrimination within each region. Therefore, we hypothesized that the regional double-dissociation (arousal vs. correct Lure discrimination) of the amygdala and hippocampus activity during post-encoding ripple windows, would show: 1) stronger effect of stimulus arousal on ripple-locked stimulus similarity in the amygdala, relative to hippocampus and 2) stronger effect of later correct Lure discrimination on the ripple-locked similarity in the hippocampus, relative to amygdala. Specifically, the regional t-value time courses were obtained by comparing the post-encoding ripple-locked (± 250 msec) stimulus similarity during trials with low- and high-arousal stimuli (1) or trials with later correct or incorrect Lure discrimination (2). Next, the t-value time courses for each effect were compared between the amygdala and hippocampus. The association between the post-encoding ripple-locked stimulus similarity and stimulus arousal was significantly stronger in the amygdala than hippocampus (-70 to 20 msec relative to ripple peak, $p < 0.001$, non-parametric cluster-based permutation test, Fig. 3d). On the other hand, the association between the post-encoding ripple-locked stimulus similarity and later correct Lure discrimination was stronger in hippocampus (-60 to 10 msec relative to ripple peak, $p = 0.046$, non-parametric cluster-based permutation test, Fig. 3d). This result suggests the presence of regional double dissociation between the amygdala and hippocampus ripple-locked activity, with the amygdala activity reflecting the stimulus arousal level and the hippocampal activity associated with later correct Lure discrimination. This answer is also related to Q5, Reviewer #2.

Page 50, Lines 10 - 16

Methods

The association strength between the post-encoding ripple-locked stimulus similarity and a) stimulus arousal or b) later correct Lure discrimination was compared between the amygdala and hippocampus. First, the regional t-values were computed, based on the comparison between a) post-encoding ripple-locked stimulus similarity for low- and high-arousal Lure stimuli or b) correctly or incorrectly discriminated Lure stimuli. The regional t-value time courses were then compared using the non-parametric cluster-based permutation test (1000 permutations, $p < 0.05$).

Page 7, Line 29 - Page 8, Line 11

Results

*The amygdala, but not the hippocampus, showed a positive association between ripple-locked **stimulus similarity** and the stimulus-induced arousal (AMY: -80 to -10 msec, $p = 0.035$; HPC: $p > 0.05$, see Methods; Fig. 3c). In contrast, the hippocampus, but not the amygdala, revealed a positive association between ripple-locked **stimulus similarity** and later correct **Lure** discrimination (AMY: $p > 0.05$; HPC: -15 to 90 msec, $p = 0.008$, see Methods; Fig. 3c). *In addition, the post-encoding**

*ripple-locked similarity in the amygdala was more strongly associated with stimulus arousal, than the ripple-locked similarity in the hippocampus (-70 to 20 msec relative to ripple peak, $p < 0.001$, non-parametric cluster-based permutation test, Fig. 3d). In contrast, the ripple-locked similarity in hippocampus was more strongly associated with later correct Lure discrimination than the amygdala ripple-locked similarity (-60 to 10 msec relative to ripple peak, $p = 0.046$, non-parametric cluster-based permutation test, Fig. 3d). To summarize, post-encoding ripple-locked *stimulus similarity* in the amygdala and the hippocampus were associated with reactivation of distinct aspects of encoded stimuli (i.e., the amygdala for stimulus-induced arousal and the hippocampus for later *Lure* discrimination accuracy).*

Fig. 3. d. Double-dissociation between the post-encoding ripple-locked stimulus similarity in hippocampus and amygdala. Left: The association between the stimulus arousal and post-encoding ripple-locked stimulus similarity was stronger in the amygdala, relative to hippocampus (-70 to 20 msec relative to ripple peak, $p < 0.001$, non-parametric cluster-based permutation test). Right: The association between the later correct Lure discrimination and post-encoding ripple-locked stimulus similarity was stronger in the hippocampus, relative to amygdala (-60 to 10 msec relative to ripple peak, $p = 0.046$, non-parametric cluster-based permutation test).

Other comments/concerns

6. I was not convinced that the term reinstatement is warranted here. I understand that the analysis was done at the time immediately following the encoding period, where the stimulus is off-screen but it still is in working memory. Therefore, I wonder if anything needs to be re-instated when it hasn't been out of mind. Maybe it would be better use a different term, i.e. post-encoding stimulus similarity.

Response: We agree that the higher stimulus similarity during post-encoding ripples does not necessarily represent memory reinstatement but could be driven by the working memory mechanisms. As noted by the Reviewer, subjects provided stimulus emotional ratings within 2 sec following the stimulus encoding (Fig. 1a), likely relying on retention of stimulus representation in working memory. This interpretation is also consistent with the proposed role of hippocampal ripples in working memory (Jadhav

et al., 2012; Sasaki et al., 2018; Zhang et al., 2021). Working memory is traditionally conceptualized as persistent stimulus representation during the delay period (Fuster and Alexander, 1971), while more recent models argue for the intermittent dynamics of stimulus representation (Miller et al., 2018). The post-encoding stimulus similarity in the present study shows discrete peaks around ripples (Fig. 3). Therefore, to the extent that post-encoding stimulus similarity is supported by working memory, it is more consistent with intermittent working memory dynamics, as proposed by Miller et al. (2018). Alternatively, rating the emotional valence of encoded stimuli could benefit from ripple-mediated memory retrieval, accompanied by memory reinstatement. This notion is consistent with the association between the memory retrieval in humans and ripple generation, albeit at a longer time-scale (Norman et al., 2019; Vaz et al., 2019; Vaz et al., 2020; Dickey et al., 2022; Sakon and Kahana, 2022). Regardless if the post-encoding stimulus similarity is driven by working memory or reinstatement, it could facilitate consolidation of represented information into a long-term memory. This hypothesis is supported by the association between the post-encoding ripples and later correct Lure discrimination (Fig. 2c). Due to the ambiguous mechanisms supporting the post-encoding stimulus representation, we replaced the term 'reinstatement' with 'post-encoding stimulus similarity', as suggested by the Reviewer. We provide further elaboration in the revised version of the Introduction and Discussion (also related to Q6, Reviewer #4).

Page 3, Lines 11 - 16

Introduction

Based on these findings, we hypothesized that ripples occurring immediately after stimulus encoding (post-encoding) facilitate emotional memory discrimination through the coordinated hippocampal-amygdala memory reinstatement *or by facilitating the retention of stimulus in working memory. Furthermore, we hypothesize that either of these processes would result in increased stimulus similarity during post-encoding ripples.*

Page 12, Lines 18 - 33

Discussion

The post-encoding stimulus similarity occurs during the stimulus valence rating, immediately following the stimulus encoding (Fig. 1a). Therefore, the higher post-encoding similarity might be driven by either the retention of stimulus representation

in working memory or by memory reinstatement. The interpretation that the post-encoding stimulus representation reflects working memory content is consistent with the proposed role of hippocampal ripples in working memory (Jadhav et al., 2012; Sasaki et al., 2018; Zhang et al., 2021). While the working memory is traditionally conceptualized as persistence of stimulus-specific activity (Fuster and Alexander, 1971), the post-encoding similarity in the present study is concentrated around ripple peaks (Fig. 3). Therefore, to the extent that post-encoding similarity is driven by working memory, it is consistent with the intermittent representation of working memory content (Miller et al., 2018). On the other hand, the stimulus valence rating during post-encoding epoch could also rely on memory retrieval, which was associated with ripple emergence (Norman et al., 2019). Regardless of the underlying mechanisms, the post-encoding ripple-locked stimulus similarity is associated with later correct Lure discrimination. This could be due to the contribution of post-encoding ripples to memory consolidation, resulting in a higher fidelity of encoded representation and higher probability of later correct Lure discrimination.

7. What is circular jittering? Please explain in the methods section.

Response: Circular jittering (also known as circular shuffling; e.g. Simonnet and Brecht, 2019; Reitich-Stolero and Paz, 2019) is a method of event time shuffling within a defined time range (up to ± 500 msec in this case) and delimited by the task epoch boundaries. The procedure is described in more detail in the revised version of the manuscript (also shown below).

Page 50, Line 30 - Page 51, Line 3

Methods

Circular jittering denotes the method of event time shuffling within a limited time window. In the context of the present study, the time window is defined by the onset of post-encoding epoch and the offset of subsequent cross-fixation (Fig. 1a). For example, if the ripple occurred 200 msec after the post-encoding onset and the randomly generated shuffled distance is -500 msec, the assigned shuffled ripple timestamp would be -300 msec prior to offset of the cross-fixation epoch.

8. I did not quite understand how different electrodes and time-stamps for SWRs were handled. For instance, where similarity values and SWRs collapsed across electrodes such that an SWR from a hippocampal electrode was used a time stamp for similarity in an electrode in the amygdala? Or were anatomical regions for SWR time stamps and similarity traces used consistently (i.e. hippocampal SWRs for hippocampal similarity, and amygdala SWRs for amygdala similarity).

Response: Similarity values were averaged across the electrodes at the region/subject level, resulting in a subject-specific stimulus similarity for each region

(amygdala and hippocampus). The ripple-locked windows for both the amygdala and hippocampal neural activity were defined relative to hippocampal ripple timestamps. The rationale for focusing on hippocampal ripples was further clarified in the Methods section of the revised manuscript.

Page 47, Lines 10 - 12

Methods

Ripple detection

Ripple-locked windows for both the hippocampal and amygdala activity were performed relative to ripple timestamps from the hippocampal channel with the highest number of detected ripples within a given subject.

Page 50, Lines 7 - 10

Representational Similarity Analysis (RSA)

The region-specific (amygdala and hippocampus) similarity matrices were averaged across trials within individual subjects (*producing the subject/region-specific similarity maps*) and used for group-level statistical analysis.

9. For the similarity analysis Empirical Mode Decomp was used. Why? Wouldn't a simple high-pass filter suffice to reject low frequency components? I understand that the authors write in the methods section that they used EMD to avoid low frequency harmonics, but I don't understand why that would be a problem since the authors are only interested in stimulus specific frequency patterns. I don't think this is critical but I would be interested in a more elaborate justification.

Response: The Reviewer is correct that bandpass filtering would prevent the frequency band of interest (broadband gamma activity) contamination by the lower frequencies activity, but not by their harmonics at higher frequencies, which might impede the detection of stimulus-specific activity patterns. In addition, simple bandpass filtering in the broadband gamma range would result in a disproportionately higher contribution of lower frequencies within this relatively broad frequency range (30-300 Hz), due to the 1/f nature of EEG power spectra. On the other hand, EEMD allows the identification of independent activity modes within the broadband gamma range, followed by normalization within each detected mode prior to averaging across the modes, resulting in a more balanced sampling across the broadband gamma range. While this rationale was not sufficiently clarified in the original version of the manuscript, it is elaborated further in the revised manuscript.

Page 48, Lines 6 - 13

Methods

This frequency range is relatively broad and the simple bandpass-filtering might cause a disproportionate contribution of lower frequencies within the bandpass range to the overall HFA estimate, due to the 1/f nature of EEG power spectra. To avoid this confound, we applied the Ensemble Empirical Mode Decomposition^{7,10} (EEMD; <https://github.com/leeneil/eemd-matlab.git>), to identify the individual characteristic signal modes within the frequency range of interest. Time series representing each individual mode were normalized across time, resulting in a more balanced sampling across the broadband gamma range.

10. Lines 393-394: A Wilcoxon test is a non-parametric test.

Response: We apologize for this error, which is corrected in the revised version of the manuscript.

Page 42, Lines 22 - 23

Methods

Unless stated otherwise, all the statistical tests (e.g., Wilcoxon signed-rank test, t-test) were two-tailed.

11. Lines 492-493: To avoid discontinuities between epochs, epochs were zeroed at beginning and start times. However, this introduces discontinuities itself unless some sort of a windowed approach was used (i.e. Hanning window).

Response: We apologize for the unclear description of this step in the ripple detection algorithm. Zeroing was performed after the Hilbert transformation of ripple range bandpass-filtered signal and before the z-scoring of resulting analytical signal amplitude. The rationale for this approach was that the signal discontinuities normally produce a large broadband power increase. However, ripple detection is based on z-score threshold and it would be affected only if the extremely high-power values resulting from discontinuities lower the z-score values of envelope amplitude during ripple events below the detection thresholds. Therefore, zeroing the narrow time window around signal discontinuities prior to z-scoring prevents the appearance of extreme z-values. Also, zeroing does not affect the ripple windows, as it includes only a narrow window (± 75 msec) around the trial edges. In addition, zeroing is performed only for the purpose of ripple detection, without affecting the representational similarity analysis. We made the method description more precise in the revised version of the manuscript.

Page 46, Lines 21 - 26

Methods

The analytical amplitude was obtained by computing the absolute value of Hilbert-transformed filtered trace (function hilbert.m in Matlab). The analytical amplitude values during periods ± 75 msec around the trial onsets/offsets were set to zero, to avoid the edge effects resulting from concatenating discontinuous traces. Finally, the envelope was z-scored and threshold-based ripple detection was performed on z-scored trace (Supplementary Fig. 13a).

12. For RSA time-bins of 100ms were used (line 550), for Mutual Information time-bins of 200ms (line 631) were used. What is the justification of these different parameters? I would feel more comfortable if these parameters were held constant.

Response: The rationale for using the 100 msec time bin size for RSA is based on the need for high temporal resolution in quantification of ripple-locked stimulus similarity. The high temporal resolution is optimal for this purpose as the ripple-locked memory reactivation is transient and typically limited to ripple duration of less than 100 msec (Buzsaki, 2015; Vaz et al., 2019; Vaz et al., 2020). Therefore, the ability to detect ripple-locked memory reinstatement or ripple-locked stimulus similarity fluctuations might be hampered by using the larger bin sizes. On the other hand, mutual information (MI) typically requires a larger number of datapoints for optimal sensitivity, since the low number of datapoints could inflate the MI estimates (Cohen, 2014). The window sizes used in the MI calculation from EEG signal range from 400 msec (e.g. Helfrich et al., 2019), up to several seconds (Sun et al., 2018). Thus, the 200 msec bin size used for MI estimation in present study was driven by the competing constraints - sufficient number of data points and temporal precision optimal for capturing the transient, ripple-locked dynamics.

Reviewer #1 (Remarks to the Author: Reproducibility):

To enhance reproducibility the authors should make their code used for the analysis publicly available.

Response: All the data and code used in the analysis are deposited in the online repository (10.5281/zenodo.7608723) and will be publicly available upon the paper publication.

Reviewer #2 (Remarks to the Author: Overall significance):

In this manuscript, Zhang and colleagues report an iEEG study (n=7) recording from the hippocampus and the amygdala during an emotional memory paradigm. They find that arousing events are better remembered and are accompanied by higher ripple numbers during the immediate post-encoding window. Reinstatement analyses suggest what encoding activity is reactivated during post-encoding ripples, with a potentially different role of the hippocampus and the amygdala in this process.

The topic of the paper is interesting and timely, but unfortunately the conclusions are not supported by the statistics, which are flawed in many instances.

The manuscript lacks reference and discussion of highly relevant iEEG/ripple papers published recently, e.g. work from Zaghloul and Malach and Halgren labs for wake ripples and work from the Fell group on ripples in hippocampus and amygdala.

Response: We thank the Reviewer for detailed feedback, which we address below. We also apologize for not citing all the relevant literature, due to the citation number limit in the original submission. In the revised version, we cite and discuss the relevant work, including the human awake ripple literature (Norman et al., 2019; Vaz et al., 2019; Vaz et al., 2020; Dickey et al., 2022; Sakon and Kahana, 2022; Sakon et al., 2022).

Reviewer #2 (Remarks to the Author: Strength of the claims):

Major concerns:

1) It is unclear whether post-encoding ripples are driven by memory processes or by stimulus-induced arousal. It is likely that these two factors are strongly correlated and the authors would need to disentangle their respective contributions, e.g., via partial correlations or holding one factor constant while examining the impact of the other.

Response: As the post-encoding ripple occurrence is associated with both the stimulus arousal level and later correct Lure discrimination (Fig. 2), we agree it would be important to disentangle these effects. The stimulus arousal modulated magnitude of association between the ripple occurrence and later correct Lure discrimination (Supplementary Fig. 4). Specifically, the difference in post-encoding ripple occurrence between the correctly and incorrectly discriminated Lure stimuli was significantly higher for high-arousal Lure stimuli ($p = 0.007$, chi-square = 7.38; Friedman's two-way ANOVA). In addition, association between the post-encoding ripple occurrence and later correct Lure discrimination was significant for both low- and high-arousing stimuli (Supplementary Fig. 4; low-arousal : $p = 0.046$, $z = -1.68$; high-arousal : $p = 0.023$, $z = -2.01$; one-tailed paired Wilcoxon test, $p < 0.05$). We elaborate on these results and possible interpretations in the revised manuscript. This answer is also related to Q4, Reviewer #1 and Q5, Reviewer #4.

Page 43, Lines 2 - 5

Methods

The strength of association between the correct Lure discrimination and post-encoding ripple occurrence was compared for low- and high-arousing stimuli using the non-parametric Friedman's two-way ANOVA ($p < 0.05$; Supplementary Fig. 4).

Page 5, Lines 21 - 29

Results

As the stimulus arousal and correct Lure discrimination are both associated with post-encoding ripple occurrence (Fig. 2c), we tested if the association between the ripple occurrence and correct Lure discrimination is modulated by the stimulus arousal level. The magnitude of post-encoding ripple modulation based on the correct Lure discrimination was stronger for high-arousal, relative to low-arousal stimuli (Supplementary Fig. 3; $p = 0.007$, $\chi^2 = 7.38$; Friedman's two-way ANOVA). Ripple occurrence was associated with correct Lure discrimination for both arousal levels (low-arousal: $p = 0.046$, $z = -1.68$; high-arousal: $p = 0.023$, $z = -2.01$; one-tailed paired Wilcoxon test, $p < 0.05$).

Page 10, Lines 12 - 14

Discussion

Significant association between the post-encoding ripples and later correct Lure discrimination was present regardless of the stimulus arousal level (Supplementary Fig. 4), but the effect was stronger for high-arousal stimuli.

Page 10, Lines 20 - 23

Post-encoding stimulus similarity (or reinstatement) is implicated in memory consolidation (Carr et al., 2011; Ben-Yakov et al., 2013; Sols et al., 2017; Schreiner and Staudigl, 2021) and peaks during ripples (Fig. 3). Thus, beside the general role of ripples in memory consolidation, these results also imply ripples as a potential mechanism mediating the effects of arousal on memory consolidation.

Page 27, Lines 1 - 8

Supplementary Fig. 4. Stimulus arousal level modulates the association between the post-encoding ripples and later correct Lure discrimination. The post-encoding ripple modulation by correct Lure discrimination was significantly stronger for high-arousal stimuli ($p = 0.007$, chi-square = 7.38; Friedman's two-way ANOVA). The association between the post-encoding ripples and later correct Lure discrimination was present regardless of the Lure stimulus arousal level (low-arousal: $p = 0.046$, $z = -1.68$; high-arousal: $p = 0.023$, $z = -2.01$; one-tailed paired Wilcoxon test).

2) To make the claim that ripples were predictive of discrimination performance in the post-encoding phase and not the peri-encoding phase, a significant Phase x Discrimination Performance interaction should be demonstrated.

Response: We performed non-parametric Friedman's two-way ANOVA to test if the association between the ripple rate and correct Lure discrimination is dependent on the task epoch. The analysis has shown significant epoch (encoding vs. post-encoding vs. retrieval) x Lure discrimination interaction ($p < 0.001$, chi-square = 13.13, Friedman's two-way ANOVA; Supplementary Fig. 7), suggesting that the later correct Lure discrimination is selectively associated with the ripple rate during post-encoding, but not the encoding or retrieval epochs.

Methods

The epoch-dependence of ripple association with correct Lure discrimination was tested using the non-parametric Friedman's two way ANOVA ($p < 0.05$; Supplementary Fig. 7).

Page 6, Lines 21 - 27

Results

The non-parametric Friedman's two-way ANOVA was used to test if the association between the ripple rate and correct Lure discrimination is task epoch-dependent. We observed significant epoch x discrimination interaction (Supplementary Fig. 7; $p < 0.001$, chi-square = 13.13, Friedman's two-way ANOVA), suggesting that the correct Lure discrimination is selectively associated with the ripple rate during post-encoding, but not during encoding or retrieval epochs.

Page 30, Lines 1 - 10

Supplementary Fig. 7. The association between ripples and correct Lure discrimination is selective for post-encoding epoch. Only the post-encoding ripple rates are associated with correct Lure discrimination (significant epoch x Lure discrimination effect, $p < 0.001$, chi-square = 13.13, Friedman's two-way ANOVA). Post-hoc analysis has shown the significantly higher ripple rates during post-encoding epoch for the correctly discriminated Lure stimuli (Post-Encoding: $p = 0.018$, $z = -2.097$, one tailed Wilcoxon sign-rank test), while there was no significant difference during the encoding or retrieval epochs (Encoding: $p = 0.418$, $z = -0.210$; Retrieval: $p = 0.338$, $z = -0.419$, one tailed Wilcoxon sign-rank test).

3) The statistics in Fig 2c are not convincing. No correction for multiple comparisons is performed. Additionally, the response times post-encoding for high vs low arousal are different. The number of ripples may be higher in high arousal because the window is larger (a mean difference of 200 ms makes the average time window for high arousal trials ~30% larger). The authors should compare ripple rates rather than the raw number of ripple events.

Response: In the revised version, we applied the Benjamini-Hochberg procedure (Benjamini and Hochberg, 1995) to correct for multiple comparisons in each task epoch (two comparisons per epoch). Following this correction, the associations between the post-encoding ripples and stimulus arousal level/correct Lure discrimination remained significant. In addition, this association is consistent across the individual participants (color-coded in Fig. 2c), making the possibility of Type 1 error unlikely. While the Reviewer is correct that mean reaction times (and consequently, the durations of post-encoding epoch) are on average ~200 msec longer on high-arousal trials, this difference is not significant ($p = 0.2$, $z = 0.7$, Wilcoxon signed-rank test). The rationale for using ripple occurrence was based on

the well-established link between the memory reinstatement and memory consolidation (e.g. Deuker et al., 2013, Staresina et al., 2013; Bird et al., 2015), as well as the presence of memory reinstatement during ripples (Axmacher et al., 2008). If the ripple-locked reinstatement facilitates memory consolidation, it is conceivable that a larger number of ripples results in a stronger effect on consolidation. A complementary approach, using the ripple rates, allows the detection of effects limited to specific time-windows that might be missed in the analysis based on ripple occurrence. In the revised version of the manuscript, we demonstrate significant association between post-encoding ripples and correct Lure discrimination/stimulus arousal using both approaches - the ripple occurrence (number of ripples/post-encoding epoch, Fig. 2; Supplementary Fig. 4) and rates (number of ripples/sec, Supplementary Fig. 5). In addition, using the ripple rate comparison, we show that significant association between the ripples and correct Lure discrimination is limited to post-encoding epoch (Supplementary Fig. 7), while not present during the encoding or retrieval epochs (shown in the response to Q2).

Page 42, Lines 25 - 34

Methods

The association between post-encoding ripples and stimulus arousal/correct Lure discrimination was tested using two complementary approaches, ripple occurrence (number of post-encoding ripples/trial, Fig. 2; Supplementary Fig. 4) and ripple rates (number of ripples/sec, Supplementary Fig. 5). The rationale for using ripple occurrence is based on the notion of ripples facilitating memory consolidation as the windows of memory reinstatement (Axmacher et al., 2008). Based on the assumption that a larger number of ripples would result in more instances of memory reinstatement, it would result in a stronger effect on consolidation. A complementary approach, using the ripple rates, allows the detection of effects limited to specific time-windows that might be missed in the analysis based on ripple occurrence.

Page 5, Line 29 - Page 6, Line 6

Results

In addition, we tested if the post-encoding ripple association with stimulus arousal/correct Lure discrimination is limited to specific periods during post-encoding epoch, by performing the conditional comparisons of ripple rates (number of ripples/sec). Post-encoding ripple rates were significantly higher for correctly discriminated, relative to incorrectly discriminated Lure stimuli (Supplementary Fig. 5; $p = 0.005$, -400 to -50 msec relative to response time), and for high-arousal, relative to low-arousal Lure stimuli (Supplementary Fig. 5; $p = 0.035$, -780 to -600 msec; non-parametric cluster-based permutation test). To summarize, the post-encoding ripple associations with stimulus arousal/correct Lure discrimination were present

during distinct, non-overlapping time windows, suggesting the distinct temporal relation between these variables and post-encoding ripples.

Page 10, Lines 6 - 12

Discussion

We demonstrate the association between the post-encoding ripples and stimulus arousal/correct Lure discrimination, using the two complementary methods. Ripple occurrence, defined as the average number of ripples during the post-encoding window, may reflect the higher number of memory reinstatement events or higher fidelity stimulus retention in working memory. Another approach, based on the ripple rate (ripples/sec), revealed the ripple temporal dynamics during the post-encoding epoch.

Page 28, Lines 1 - 12

Supplementary Fig. 5. The time-resolved association between ripple rate and stimulus arousal/correct Lure discrimination across the task epochs. a, During stimulus encoding, there was no significant ripple rate difference between the correctly and incorrectly discriminated Lure stimuli (top; $p > 0.05$, non-parametric cluster-based permutation test) or low-arousal and high-arousal Lure stimuli (bottom; $p > 0.05$, non-parametric cluster-based permutation test). b, During post-encoding, ripple rates were significantly higher for correctly discriminated, relative to incorrectly discriminated Lure stimuli (top, $p = 0.005$, -400 to -50 msec relative to response time), and for high-arousal, relative to low-arousal Lure stimuli (bottom, $p = 0.035$, -780 to -600 msec relative to response time, non-parametric cluster-based permutation test).

4) Likewise, their "region-specific double dissociation" is no double dissociation. They merely found effect A in region 1 and effect B in region 2. A double dissociation warrants a significant 2x2 interaction. That is, they would need to show that effect A is significantly greater in region 1 than in region 2 and that effect B is significantly greater in region 2 than in region 1.

Response: We thank the Reviewer for this suggestion. To test for regional double-dissociation (arousal vs. correct Lure discrimination), the regional t-value time courses were obtained by comparing the post-encoding ripple-locked (± 250 msec) stimulus similarity during trials with low- and high-arousal stimuli (1) or trials with later correct or incorrect Lure discrimination (2). Next, the t-value time courses for each effect were compared between the amygdala and hippocampus. The association between the post-encoding ripple-locked stimulus similarity and stimulus arousal was significantly stronger in the amygdala than hippocampus (-70 to 20 msec relative to ripple peak, $p < 0.001$, non-parametric cluster-based permutation test, Fig. 3d). On the other hand, the association between the post-encoding ripple-locked stimulus similarity and later correct Lure discrimination was significantly stronger in hippocampus (-60 to 10 msec relative to ripple peak, $p = 0.046$, non-parametric cluster-based permutation test, Fig. 3d). This result suggests the presence of regional double dissociation between the amygdala and hippocampus ripple-locked activity, with the amygdala activity reflecting the stimulus arousal level and the hippocampal activity associated with later correct Lure discrimination. This answer is also related to Q4, Reviewer #1.

Page 50, Lines 10 - 16

Methods

The association strength between the post-encoding ripple-locked stimulus similarity and a) stimulus arousal or b) later correct Lure discrimination was compared between the amygdala and hippocampus. First, the regional t-values were computed, based on the comparison between a) post-encoding ripple-locked stimulus similarity for low- and high-arousal Lure stimuli or b) correctly or incorrectly discriminated Lure stimuli. The regional t-value time courses were then compared using the non-parametric cluster-based permutation test (1000 permutations, $p < 0.05$).

Page 7, Line 29 - Page 8, Line 11

Results

The amygdala, but not the hippocampus, showed a positive association between ripple-locked stimulus similarity and the stimulus-induced arousal (AMY: -80 to -10 msec, $p = 0.035$; HPC: $p > 0.05$, see Methods; Fig. 3c). In contrast, the hippocampus, but not the amygdala, revealed a positive association between ripple-locked stimulus similarity and later correct Lure discrimination (AMY: $p > 0.05$; HPC: -15 to 90 msec, $p = 0.008$, see Methods; Fig. 3c). In addition, the post-encoding ripple-locked similarity in the amygdala was more strongly associated with stimulus arousal, than the ripple-locked similarity in the hippocampus (-70 to 20 msec relative to ripple peak, $p < 0.001$, non-parametric cluster-based permutation test, Fig. 3d). In contrast, the ripple-locked similarity in hippocampus was more strongly associated

with later correct Lure discrimination than the amygdala ripple-locked similarity (-60 to 10 msec relative to ripple peak, $p = 0.046$, non-parametric cluster-based permutation test, Fig. 3d). To summarize, post-encoding ripple-locked stimulus similarity in the amygdala and the hippocampus followed distinct temporal dynamics and were associated with reactivation of distinct aspects of encoded stimuli (i.e., the amygdala for stimulus-induced arousal and the hippocampus for later Lure discrimination accuracy).

Page 20, Line 1 - Page 21, Line 13

Fig. 3. d. Double-dissociation between the post-encoding ripple-locked stimulus similarity in hippocampus and amygdala. Left: The association between the stimulus arousal and post-encoding ripple-locked stimulus similarity was stronger in the amygdala, relative to hippocampus (-70 to 20 msec relative to ripple peak, $p < 0.001$, non-parametric cluster-based permutation test). Right: The association between the later correct Lure discrimination and post-encoding ripple-locked stimulus similarity was stronger in the hippocampus, relative to amygdala (-60 to 10 msec relative to ripple peak, $p = 0.046$, non-parametric cluster-based permutation test).

5) The authors state "aSWR-locked joint memory reinstatement was present during the post-encoding period only for correctly discriminated stimuli". To establish the important link between reinstatement and memory, they would need to show that reinstatement is significantly stronger for successful than unsuccessful encoding.

Response: We agree that the significantly stronger reinstatement of the later correctly discriminated Lure stimuli is necessary for arguing the link between the reinstatement and memory. In the original submission, we show that the ripple-locked reinstatement in the hippocampus is significantly stronger for the later correctly discriminated Lure stimuli (lower right plot in the Fig. 3c). We apologize for using unclear terminology ('robust' and 'positive association') to denote the statistical significance, which is replaced by 'significantly higher' and 'significant association' in the revised version. In addition, based on the recommendation from Reviewer #1, the term 'memory reinstatement' is replaced by the 'post-encoding stimulus similarity'.

Page 7, Line 32 - Page 8, Line 2

Results

In contrast, the hippocampus, but not the amygdala, revealed a *significant* association between ripple-locked *post-encoding stimulus similarity* and later correct *Lure* discrimination (AMY: $p > 0.05$; HPC: -15 to 90 msec, $p = 0.008$, see Methods; Fig. 3c).

Page 20, Line 1 - Page 21, Line 5

Fig. 3c. *Ripple-locked post-encoding stimulus similarity* in the hippocampus is *significantly higher* for correctly discriminated *Lure* stimuli (bottom right, $p = 0.008$, see Methods)

6) Overall, the reinstatement results (Fig 3) are not convincing:

a) The correlation is not driven by a specific/circumscribed encoding time window, but based on numerous short intervals, which is at odds with established encoding/retrieval reinstatement findings.

Response: In the revised version, we show the significant stimulus-specific ripple-locked similarity in the hippocampus, based on the single circumscribed cluster, covering ~250 msec of encoding and ripple windows (Supplementary Fig. 11), similar to other reports (e.g. Zhang et al., 2018, Fig. 2; Pacheco Estefan et al., 2019, Fig. 2 and 3). Also, similarity might be computed between the multiple equally long presentations of the same stimulus or the stimulus presentation (encoding) and post-encoding ripple window of unequal duration (2000 msec vs. 500 msec in the present study). Details of experimental design such as this might influence the size/timing of a significant similarity cluster. The method and results are elaborated in more detail in the response to Q6c, Reviewer #2 and Q3, Minor points, Reviewer #3.

Page 8, Lines 19 - 26

Results

The post-encoding ripple-locked neural activity in the hippocampus (-190 - 20 msec, relative to ripple peak) shows the significant stimulus-specific similarity with the activity during stimulus encoding (~500 - 750 msec following the onset of encoding epoch; non-parametric cluster-based permutation test; n = 1000 permutations, p < 0.05; Supplementary Fig. 11). The size/timing of significant stimulus similarity during encoding is consistent with previous reports (Zhang et al., 2018; Pacheco Estefan et al., 2019), but might also be driven by the factors specific to present study, such as the comparison between the encoding and post-encoding ripple activity.

b) The effects of comparing low/high arousal and successful/unsuccessful memory appear to be driven by decreases in similarity for low arousal and unsuccessful encoding, rather than by an increase for high arousal and successful memory, respectively.

Response: To test for the possibility of significant effects shown in Fig. 3c being driven by decreased stimulus similarity, we compared the similarity values during the significance window with the baseline, consisting of the rest of the ripple-locked window. This analysis has shown no significant similarity difference between the baseline and significance window for the low-arousal trials in the amygdala (p = 0.241, t = 0.759, df = 5) or the later incorrect Lure discrimination trials in the hippocampus (p = 0.092, t = 1.541, df = 5). Therefore, in the absence of significant similarity decrease during significance windows, significance is unlikely driven by the decreased similarity.

c) Apart from shuffling ripple times, an important control would be to shuffle encoding trials in order to ensure reinstatement is event-specific.

Response: We agree with the Reviewer that demonstrating stimulus-specificity of post-encoding ripple activity is important to distinguish it from nonspecific memory-related activity. To answer this question, we compared the similarity between the post-encoding ripple windows and encoding epoch on the same trial (S_{same}) or different trials (S_{diff}), while also controlling for the temporal proximity between the encoding and post-encoding epochs on the same or different trials. The method is elaborated in more detail below. This procedure resulted in the stimulus-specific similarity map, showing a cluster of significant stimulus-specific similarity during the window ~500 - 750 msec following the onset of encoding epoch and -190 - 20 msec relative to post-encoding ripple peak (Supplementary Fig. 11). Therefore, the activity during post-encoding ripple windows is specific to stimulus presented on the same trial. This response is also related to Q6a, Reviewer #2 and Q3, Minor points, Reviewer #3.

Page 51, Line 28 - Page 52, Line 13

Methods

The stimulus-specificity of ripple-locked activity in the hippocampus was determined by first computing the similarity between the activity during encoding epoch on i -th trial (Enc_i) and post-encoding ripple window (Ripple) on the same trial ($S_{same} = r(Enc_i, Ripple_i)$). Next, we computed the similarity between other stimuli encoding epochs and ($Enc_{1, 2, \dots, n}$) and post-encoding ripple window on the i -th trial ($S_{diff} = r(Enc_{1, 2, \dots, n}, Ripple_i)$), n denoting the number of different stimuli in the experiment). As the S_{same} might be inflated due to temporal proximity between the Enc_i and $Ripple_i$ (Howard and Kahana, 2002; Polyn et al., 2009), we accounted for the difference in average post-encoding epoch similarity (S_{avg}) between the same and different trials. Specifically, S_{avg} was defined as the difference in similarity between the encoding epoch and the entire post-encoding window on the same trial (S_{same_avg}) or different trials (S_{diff_avg}). For each individual stimulus, the stimulus-specific similarity (S_{spec}) was defined as following:

$$S_{spec} = (S_{same} - S_{diff}) - (S_{same_avg} - S_{diff_avg})$$

Next, the S_{spec} was averaged at subject level and the t -statistics was obtained by comparing the subject-averaged S_{spec} with zero. The similarity null-distribution (S_{shuff}) was created by shuffling the stimulus identity (same vs. different) 1000 times and obtaining the t -statistics as described above. Finally, the cluster statistics was performed by comparing the t -values obtained from S_{spec} with the distribution of t -values obtained from S_{shuff} (non-parametric cluster-based permutation test; $n = 1000$ permutations, $p < 0.05$; Supplementary Fig. 11).

Page 8, Lines 19 - 26

Results

The post-encoding ripple-locked neural activity in the hippocampus (-190 - 20 msec, relative to ripple peak) shows the significant stimulus-specific similarity with the activity during stimulus encoding (~500 - 750 msec following the onset of encoding epoch; non-parametric cluster-based permutation test; $n = 1000$ permutations, $p < 0.05$; Supplementary Fig. 11). The size/timing of significant stimulus similarity during encoding is consistent with previous reports (Zhang et al., 2018; Pacheco Estefan et al., 2019), but might also be driven by the factors specific to present study, such as the comparison between the encoding and post-encoding ripple activity.

Page 34, Lines 1 - 6

Supplementary Fig. 11. The representational similarity map showing the stimulus specificity of the post-encoding ripple representation in the hippocampus. The temporal cluster of significant stimulus-specific similarity (-190 - 20 msec, relative to ripple peak and ~500-750 msec of encoding time) is encircled in black (non-parametric cluster-based permutation test; $n = 1000$ permutations, $p < 0.05$).

d) The time window around the ripple event in Fig 3a overlaps with the peri-encoding stage (-250-0 ms). To claim reinstatement of peri-encoding patterns during post-encoding, data from the peri-encoding phase prior to a ripple in post-encoding cannot be treated as the post-encoding phase.

Response: All the plots in Fig. 3 (including the Fig. 3a) are showing ripple-locked windows, centered around the individual ripple peaks. Therefore, the dashed line in those plots represents the average ripple peak time and not the transition between the encoding and post-encoding epoch. Also, only the parts of the ripple-locked windows non-overlapping with the encoding epoch were used in ripple-locked analysis. We are sorry for not clarifying this detail in the original submission, which is included in the revised version.

Page 50, Line 23 - 25

Methods

To avoid the leakage of encoding epoch activity, only the part of the ripple-locked windows non-overlapping with the encoding epoch was used in the ripple-locked analysis.

e) The authors cannot claim "temporal significant MI difference (AMY-HPC) is present before aSWR peak time (-70 to -30 msec)" in Fig 3e, since cluster-based permutation analysis with cluster-mass is unable to support claims of specific temporal and spatial significance, especially on these time-scales. The same applies to claims of temporal reinstatement profiles in Fig 3b.

Response: Ripples are transient events (<100 msec), so the ripple-locked stimulus similarity changes are expected to be present at similar timescales. In fact, ripple-locked changes extending over much longer periods would be in contrast to established literature on the transient nature or ripple-locked dynamics (e.g. Zhang et al., 2018, Fig. 4d; Vaz et al., 2020, Fig. 4c; Norman et al., 2021, Fig. 6c). From the methodological aspect, cluster-based permutation has been used in detecting the significant brain dynamics changes at comparably brief timescales (~150 msec - Frömer et al., 2018, Fig. 4; ~100 msec, Jas et al., 2018, Fig. 6; 40-125 msec - Rocchi et al., 2021, Fig. 4). Therefore, using cluster statistics to detect ripple-locked stimulus similarity dynamics is valid based on the established sensitivity of cluster-based permutation at these timescales.

Figure 2c appears to show the absolute number of ripples, which seems identical for the 2 s encoding period and the ~ 0.7 s response period. Ripple occurrences would need to be converted into ripple rates (occurrence/time; see comment 3).

Response: We have included the analysis based on ripple rates (events/sec) in the revised version of the manuscript. The results of this analysis are shown below, as well as in answer to Q3 (Reviewer #2) and Q7 (Reviewer #4).

Page 42, Lines 25 - 34

Methods

The association between post-encoding ripples and stimulus arousal/correct Lure discrimination was tested using two complementary approaches, ripple occurrence (number of post-encoding ripples/trial, Fig. 2; Supplementary Fig. 4) and ripple rates (number of ripples/sec, Supplementary Fig. 5). The rationale for using ripple occurrence is based on the notion of ripples facilitating memory consolidation as the windows of memory reinstatement (Buzsaki, 2015). Based on the assumption that a larger number of ripples would result in more instances of memory reinstatement, it would result in a stronger effect on consolidation. A complementary approach, using the ripple rates, allows the detection of effects limited to specific time-windows that might be missed in the analysis based on ripple occurrence.

Page 5, Line 29 - Page 6, Line 6

Results

In addition, we tested if the post-encoding ripple association with stimulus arousal/correct Lure discrimination is limited to specific periods during post-encoding epoch, by performing the conditional comparisons of ripple rates (number of ripples/sec). Post-encoding ripple rates were significantly higher for correctly discriminated, relative to incorrectly discriminated Lure stimuli (Supplementary Fig. 5; $p = 0.005$, -400 to -50 msec relative to response time), and for high-arousal, relative to low-arousal Lure stimuli (Supplementary Fig. 5; $p = 0.035$, -780 to -600 msec; non-parametric cluster-based permutation test). To summarize, the post-encoding ripple associations with stimulus arousal/correct Lure discrimination were present during distinct, non-overlapping time windows, suggesting the distinct temporal relation between these variables and post-encoding ripples.

Page 10, Lines 6 - 12

Discussion

We demonstrate the association between the post-encoding ripples and stimulus arousal/correct Lure discrimination, using the two complementary methods. Ripple occurrence, defined as the average number of ripples during the post-encoding window, may reflect the higher number of memory reinstatement events or higher fidelity stimulus retention in working memory. Another approach, based on the ripple rate (ripples/sec), revealed the ripple temporal dynamics during the post-encoding epoch.

Page 28, Lines 1 - 12

Supplementary Fig. 5. The time-resolved association between ripple rate and stimulus arousal/correct Lure discrimination across the task epochs. a, During stimulus encoding, there was no significant ripple rate difference between the correctly and incorrectly discriminated Lure stimuli (top; $p > 0.05$, non-parametric cluster-based permutation test) or low-arousal and high-arousal Lure stimuli (bottom; $p > 0.05$, non-parametric cluster-based permutation test). b, During post-encoding, ripple rates were significantly higher for correctly discriminated, relative to incorrectly discriminated Lure stimuli (top, $p = 0.005$, -400 to -50 msec relative to response time), and for high-arousal, relative to low-arousal Lure stimuli (bottom, $p = 0.035$, -780 to -600 msec relative to response time, non-parametric cluster-based permutation test).

On that note, it would be important to explicitly state how many trials had one or more ripples during the response window and how the authors speculate memory processes occur on trials without ripples.

Response: In the revised version of the manuscript, we note the percentage of trials containing at least one ripple during post-encoding period, separately for the

correctly and incorrectly discriminated or the low- and high-arousal Lure stimuli. The absence of post-encoding ripples in a subset of trials could be explained by the local ripple amplitude being below the detection threshold or the ripples occurring in the different parts of the hippocampus or in the contralateral hippocampus.

Page 6, Lines 30 - 31

Results

Overall, $30.8 \pm 7.4\%$ (mean \pm SEM) of Lure trials contained one or more ripples during the post-encoding period.

Page 10, Lines 23 - 27

Discussion

It should be noted that a subset of trials contained no ripples during the post-encoding period. During these trials, ripple amplitudes might have been below our detection threshold. In addition, the ripples might have occurred in the different parts of the hippocampus or in the contralateral hemisphere.

Importantly, for the "aSWR-locked joint memory reinstatement" analysis, are there sufficient trials in which both regions show a ripple?

Response: Only hippocampal ripples were used in all the ripple-locked analysis. Also related to the question by Reviewer #3 (Q1), we elaborate below on the rationale for focusing on hippocampal ripples in the ripple-based analysis, as well as on the coincidence between the hippocampal ripples and ripple-like activity in the amygdala. The presence of ripples was reported in the human amygdala, based on the similar detection algorithm as used in the present study (Cox et al., 2020). However, the presence of ripples in extrahippocampal structures, especially in the epileptic brain, is still a subject of debate (see the consensus statement by Liu et al., 2022). Nevertheless, it is possible that the periods of high post-encoding similarity in the amygdala, coinciding with hippocampal ripples (Fig. 3), are driven by the amygdala ripple activity. To better characterize the amygdala high-frequency events during hippocampal ripples, we performed the event detection in amygdala using the same algorithm as used for hippocampal ripple detection. This analysis has shown that hippocampal ripples rarely coincide with ripple-like activity in the amygdala within the ± 50 msec window ($5.89 \pm 1.82\%$, mean \pm SEM). Regardless of the nature of amygdala activity during hippocampal ripples, stimulus similarity in the amygdala and hippocampus both coincide with hippocampal ripples, and this coincidence is also predictive of later correct Lure discrimination (Fig. 4). We have elaborated on this control analysis in the revised version of the manuscript (also related to Q1, Analysis or examples that would strengthen the claims, by Reviewer #3).

Page 46, Lines 15 - 19

Methods

Following the removal of channels with excessive epileptic activity and individual trials containing visually identified interictal epileptic discharges, ripples were detected on the remaining hippocampal channels, using the Freely Moving Animal Toolbox (FMA; <http://fmatoolbox.sourceforge.net/>). *Only the hippocampal channels were used in ripple detection for ripple-based analysis.*

Page 47, Lines 12 - 15

Ripple-like activity in the amygdala was detected using the same algorithm. Coincidence between the hippocampal ripples and ripple-like activity in the amygdala was calculated as the percentage of hippocampal ripples accompanied by amygdala ripple-like activity within the ± 50 msec.

Page 9, Lines 10 - 14

Results

Ripple-like events were also detected in the amygdala, similar to recent reports (Cox et al., 2020). However, only $5.89 \pm 1.82\%$ (mean \pm SEM) of hippocampal ripples were accompanied by ripple-like events in the amygdala within the ± 50 msec window. Thus, the joint increases in post-encoding stimulus similarity during hippocampal ripples (Fig. 4) were not dependent on coincident presence of ripple-like activity in the amygdala.

Page 12, Lines 7 - 17

Discussion

The presence of ripple-like activity, coincident with hippocampal ripples, was recently reported in the human amygdala (Cox et al., 2020) and cortical areas (Vaz et al., 2019; Vaz et al., 2020; Dickey et al., 2022). While the presence of ripples in the extrahippocampal structures, especially in the epileptic brain, is still a subject of debate (Liu et al., 2022), it is possible that the post-encoding stimulus similarity in the amygdala is driven by ripple-like activity. This would be consistent with the co-occurrence of hippocampal ripples and ripple-like activity in temporal cortex during memory retrieval (Dickey et al., 2022). However, only $\sim 5\%$ of hippocampal ripples were accompanied by ripple-like activity in the amygdala. Regardless of the nature of ripple-like activity in the amygdala, the relatively low coincidence with hippocampal ripples suggests the sufficiency of hippocampal ripples alone for coordinating the joint stimulus representation across the structures (Fig. 4a, Supplementary Fig. 12).

Other comments:

1) How many analysed channels are from the hippocampus vs amygdala? It would be useful to see selected channels for analysis highlighted in Fig 2a.

Response: Overall, 17 hippocampal and 23 amygdala channels were processed. After establishing that the number of ripples in subject 1 was too low on all the hippocampal channels, the channels from this subject were excluded from the ripple-based analysis ($n = 3$ hippocampal and 3 amygdala channels). Therefore, the final number of channels used in the ripple-based analysis is 14 hippocampal and 20 amygdala channels. In the revised version of the manuscript, only the channels used in the analysis are shown in Fig. 2a. This point was also clarified in the Results section.

Page 5, Lines 13 - 17

Results

While the behavioral analysis was performed on 7 subjects, one subject was excluded from the ripple-based analysis, due to the low number of recorded ripples. The ripple-based analysis was performed on 14 hippocampal and 20 amygdala electrodes, in 6 subjects. The locations of electrodes used in the analysis are shown in Fig. 2a and the average ripple waveform is shown in Fig. 2b.

2) Why was 150 events chosen as a cut-off to include channels? Did this result in the exclusion of the 7th participant? If so, what effect did that have on results? In a study with so few participants each excluded participant potentially has a large influence on an effect that is already fragile.

Response: We apologize for the mistake when describing the inclusion criteria in the original submission. Instead of the >150 ripple events, we used the z-score threshold > -2 , which identified subject 1 as an outlier with respect to ripple number distribution across all the subjects. In addition, based on the suggestion from Reviewer #1, we ran the ripple detection on the synthetic signal of the same spectral properties as the original hippocampal signal, using the same detection algorithm. This analysis revealed that in all subjects, except subject 1, ripple detection in the hippocampal signal exceeded the event detection in the synthetic signal, providing additional justification for excluding this subject.

Page 47, Lines 2 - 5

Methods

Only the channels with *z-scored number of detected ripples* > -2 and number of detected events higher than in the synthetic signal (see Ripple detection from synthetic signal) were used in the analysis.

Page 47, Line 18 - Page 48, Line 2

Ripple detection from synthetic signal

The power spectral density (PSD) was calculated for each hippocampal channel used in ripple detection. Next, the channel-specific filter was applied on a random signal with Gaussian distribution, resulting in a synthetic signal with identical spectral slope as the hippocampal signal. Specifically, the synthetic signal was first transformed to frequency domain by N-point Fourier transform (N denoting the number of datapoints in the signal), followed by multiplication of resulting spectrum by the PSD coefficients and application of inverse Fourier transform, to convert the signal back to time domain. The resulting synthetic signal mimicked the hippocampal channel-specific spectral characteristics. Next, the ripple detection procedure (see Ripple detection) was applied on the synthetic signal. Finally, as an additional control, the numbers of detected events were compared between the hippocampal iEEG signal and channel-specific synthetic signal. In all the 6 subjects used in the ripple-based analysis (subjects 2-7), the numbers of detected ripples were higher than the numbers obtained from the subject-specific synthetic signals (Supplementary Fig. 14). Subject 1 was excluded from the analysis based on the two criteria: a) low number of detected putative ripple events (z-score < -2, relative to distribution across the subjects) and b) the number of detected events lower than chance level, defined as the number of detected events in synthetic signal (Supplementary Fig. 14).

Supplementary Fig. 14. Ripple detection in the hippocampal signal and synthetic signal. Comparisons between the numbers of ripples detected in hippocampal channels (left) and number of events detected in subject-specific synthetic signals of the same spectral characteristics (right). In all the subjects included in ripple analysis (subjects 2-7) ripple detection in hippocampal signal was higher than in synthetic signal. Subject 1 was excluded from the ripple-based analysis due to low ripple detection in hippocampal signal (z -score < -2 , relative to distribution across the subjects) and the number of detected putative ripples lower than in synthetic signals of the same spectral characteristics.

3) The statement, "The aSWRs probability was significantly higher during low theta power periods (Extended Data Fig 5), consistent with observations that cholinergic tone promotes theta oscillations and suppresses SWRs" comes out of the blue and does not add to the current question.

Response: We agree that invoking the cholinergic tone as a possible mechanism underlying the higher ripple probability during low theta periods (Supplementary Fig. 8 in the revised version) is not warranted in this context. Cholinergic tone is not directly measured in this experiment and the underlying neuromodulatory mechanisms are not critical for the main research question of the study - association between the post-encoding ripples and stimulus arousal/correct Lure discrimination. Therefore, we excluded the mention of cholinergic tone from the revised version.

Page 6, Line 31 - Page 7, Line 1

Results

Ripple probability was significantly higher during low theta power periods (Supplementary Fig. 8), consistent with observations *of ripple suppression during periods of pronounced theta oscillations*^{10,12}.

4) Colour maps should be symmetrical (e.g., Fig 3a).

Response: The asymmetrical color maps in the present study were used for the purpose of limiting the map range to the range of presented data values. Since all the values in Fig. 3a are positive, extending the color map range below zero would impede the efficient visualization. We list below several examples of RSA maps from published literature, containing the asymmetrical color maps covering the entire range of presented data points (Staresina et al., 2016, Fig. 2A; Salmela et al., 2018, Fig. 2d; Pacheco Estefan et al., 2020, Fig. 3b).

5) One-tailed testing in Fig S9 is not justified.

Response: The analysis shown in Fig. 3d was testing if the post-encoding similarity in the amygdala and hippocampus significantly **increases** around the hippocampal ripple peaks. Based on the extensive literature in rodents and humans (e.g. Kudrimoti et al., 1999; O'Neill et al., 2008; Girardeau et al., 2017; Wilber et al., 2017; Zhang et al., 2018; Vaz et al., 2020; Norman et al., 2021), ripples are established as windows of memory reinstatement/reactivation in the hippocampus and extra-hippocampal structures (see Buzsaki, 2015 for review). Memory reinstatement is defined as transiently increased similarity between the neural activity during a given experience (stimulus presentation, in this case) and the activity at any later time point (Genzel et al., 2020). Since the hypothesized change in ripple-locked stimulus similarity was unidirectional (**increase**), we felt justified to use one-tailed testing in this context.

6) The authors may want to stick to SWR and omit the "a" there is now a substantial body of literature using SWR for both wake and sleep ripples and new acronyms are somewhat confusing.

Response: Following the Reviewer's recommendation and trying to reduce the terminological confusion, we removed the acronym 'a', used to denote that ripples analyzed in the present study were recorded in awake subjects.

Reviewer #2 (Remarks to the Author: Reproducibility):

The authors should consider sharing the code required to reproduce the figures and statistics.

Response: The data and code were deposited in the online repository (10.5281/zenodo.7608723) and will be publicly available upon publication.

Reviewer #3 (Remarks to the Author: Overall significance):

Here the authors show that the occurrence of awake sharp-wave ripples in humans is correlated with both the arousal triggered by a visual stimulus and its later successful recollection. Further, they show that the joint reinstatement of hippocampal and amygdala intracranial EEG patterns around the time of the ripple is correlated with the successful recollection of emotional stimuli, suggesting a role for SWR in the preferential encoding of arousing stimuli. This is a new and important finding since we know that emotional valence influences memory formation but, to date, the physiological mechanisms by which this could be mediated are relatively unknown.

Reviewer #3 (Remarks to the Author: Impact):

To my knowledge, this is the first paper that explores the question of encoding and retrieval of emotional material using intracranial recordings in humans. It reveals a potential mechanism for the preferential consolidation of emotional memories and paves the way for future studies in human and animal models.

Reviewer #3 (Remarks to the Author: Strength of the claims):

Overall, the work is convincing as presented, although it might lack statistical power due to the scarcity of data (short trials, few ripples, few patients). However, I think this should be considered in light of the difficulty to obtain intracranial human EEG data and acknowledged as an inevitable drawback that doesn't affect the potential and novelty of the results.

Response: We thank the Reviewer for acknowledging the importance and novelty of the present study. We agree that the number of subjects is relatively low, due to paucity of subjects with simultaneous hippocampal and amygdala recordings outside of seizure zone. While the higher number of subjects enhances the confidence in study interpretation, a number of recent high profile human intracranial studies had similar or lower numbers of subjects (e.g. Vaz et al, 2020, n = 6; Dickey et al., 2021, n = 4). We acknowledge this limitation in the revised manuscript, while also presenting the individual subjects data (Fig. 1c, 2c and Supplementary Fig. 2a, b, 3, 6, 8b, 9, 14), which support the study conclusions. This answer also pertains to a Q1 by Reviewer #1.

Page 13, Lines 8 - 12

Discussion

The number of subjects in the present study is relatively low, due to paucity of subjects with simultaneous amygdala and hippocampus recordings outside of seizure onset zone. However, the effects are consistent across the individual subjects (Fig. 1c, 2c and Supplementary Fig. 2a, b, 3, 6, 8b, 9, 14), mitigating the possibility of outliers affecting the study conclusions.

I have however a few analysis points that I would like to see addressed.

Major points :

1. I am unfamiliar with the reinstatement method the authors use. However, in the reinstatement analysis for the hippocampus, I am worried about a circular confound: indeed, the similarity measure is calculated over a range of high frequencies that include the ripple range, and then the correlation is calculated at times surrounding pre-detected ripples: won't the correlation coefficient necessarily be high around the ripple times? If so, this invalidates the rest of the results that use this reinstatement measure, for example, the joint reinstatement in Fig 3d.

Response: We thank the Reviewer for this point, as it is important to assess all the alternative interpretations of the study results. The method used for quantifying memory reinstatement/post-encoding similarity in the present study is representational similarity analysis (RSA; Kriegeskorte et al., 2008), widely adopted in the analysis of electrophysiological and fMRI signals (e.g. Bruffaerts et al., 2013; Kaneshiro et al., 2015; Salmela et al., 2018), including during the ripple windows (Zhang et al., 2018). The similarity metric is based on correlating the neural activity (broadband gamma range) during a given experience (stimulus encoding epoch) and the neural activity at a later time (post-encoding epoch; Fig. 1a, Supplementary Fig. 15c). As pointed out by the Reviewer, ripple frequency range (80-150 Hz) overlaps with the broadband gamma range used for similarity assessment (30-300 Hz). However, several lines of evidence suggest that the frequency overlap does not inflate the ripple-locked similarity. First, this scenario would not account for the stimulus specificity of ripple-locked activity (Supplementary Fig. 11 in the revised manuscript). The stimulus-specificity denotes the significantly higher post-encoding similarity to stimulus encoded on the same trial (S_{same}), relative to other stimuli (S_{diff} , see the Stimulus-specific representational similarity, Methods). Second, ripple-locked similarity driven by ripple/broadband gamma frequency overlap would not explain the association between ripple-locked similarity and stimulus arousal/correct Lure discrimination (Fig. 3). Finally, ripple probability is lower during periods of higher broadband gamma power (as defined by median-split, Supplementary Fig. 8b), suggesting the distinction between the ripple-locked activity and broadband gamma. These arguments support the interpretation that ripple-locked dynamics during post-encoding epoch is stimulus-specific and relevant for stimulus memory consolidation, rather than reflecting the ripple frequency overlap with broadband gamma. We elaborated on these points in the Discussion section of the revised manuscript.

Supplementary Fig. 8. b, Ripple proportion is lower during the high theta state (top, $p = 0.017$, $z = 2.1$, one-tailed Wilcoxon signed-rank test), or during high gamma state (bottom, $p = 0.028$, $z = 1.9$, one-tailed Wilcoxon signed-rank test). Theta/gamma state classification was based on the power median split (for details, see 'Dual state analysis').

Page 10, Lines 27 - 34

Discussion

The frequency overlap between the broadband gamma (30-300 Hz) and ripples (80-150 Hz) could theoretically result in a circularly inflated stimulus similarity during post-encoding ripples. However, this scenario would not explain the stimulus-specificity of ripple-locked activity (Supplementary Fig. 11), nor the significant association between the post-encoding ripples and stimulus arousal/later correct Lure discrimination (Fig. 3). Therefore, the ripple-locked stimulus similarity during post-encoding epoch is likely functionally relevant for emotional memory consolidation, rather than an epiphenomenon of broadband gamma/ripple frequency overlap.

Minor points :

1. Given the data that is available for analysis in the response time window (less than a second), the data for SWR occurrence is binary (ie either one or zero SWR). I'd suggest it appears more clearly how the authors have accounted for this specificity in their analysis and statistics.

Response: In the original version of the manuscript, only the trials containing ripples were used in ripple-based analysis. Based on the Reviewer's recommendation (Analysis or examples that would strengthen the claims, Q2), we introduced a complementary analysis in the revised version, including the trials without ripples.

Specifically, we tested the association between the post-encoding theta power and correct Lure discrimination. In addition, we computed the subject-average post-encoding ripple times (relative to response time) and used those as reference points for event-locked similarity analysis in the trials without ripples. The results of this analysis are presented as answers to Q3 (Analysis or examples that would strengthen the claims, Page 46).

2. It is unclear whether 6 or 7 subjects have been used since 1 subject has been discarded for having too few aSWR. Was the HF data still analyzed in this patient, although most analyses are restricted to SWR occurring time?

Response: We are sorry for the unclear description. The data from subject 1 (excluded from ripple-based analysis due to low number of ripples) was used in the behavioral analysis (Fig. 1), but not in any of the iEEG analysis, as those were ripple-based. This has been clarified in the revised version of the manuscript.

Page 5, Lines 13 - 17

Results

While the behavioral analysis was performed on 7 subjects, one subject was excluded from the ripple-based analysis, due to the low number of recorded ripples. The ripple-based analysis was performed on 14 hippocampus and 20 amygdala electrodes, in 6 subjects. The locations of electrodes used in the analysis are shown in Fig. 2a and the average ripple waveform is shown in Fig. 2b.

3. In figure 3, the significance of the reinstatement has been calculated for all trials with a SWR (a) but when separating trials according to arousal or correctness, the significance of the reinstatements haven't been tested against the null, shuffled hypothesis (but only between the 2 types of trial). I think this is missing.

Response: We agree that besides comparing the reinstatement (referred to as post-encoding similarity in the revised version) between the different trial types, it would be important to test the stimulus-specificity of ripple-locked activity. The null-hypothesis in this case would be that the similarity between the neural activity during encoding and post-encoding ripple on the same trial would not be different from the similarity between the encoding and post-encoding ripples on different trials. To address this question, we compared the similarity between the post-encoding ripple windows and encoding epoch on the same trial (S_{same}) or different trials (S_{diff}), while also controlling for the temporal proximity between the encoding and post-encoding epochs on the same or different trials. The method is elaborated in more detail below. This procedure resulted in the stimulus-specific similarity map, showing a cluster of significant stimulus-specific similarity during the window ~500 - 750 msec following the onset of encoding epoch and -190 - 20 msec relative to post-encoding

ripple peak (Supplementary Fig. 11). Therefore, the activity during post-encoding ripple windows is specific to stimulus presented on the same trial. This response is also related to Q6a and c, Reviewer #2.

Page 51, Line 28 - Page 52, Line 13

Methods

The stimulus-specificity of ripple-locked activity in the hippocampus was determined by first computing the similarity between the activity during encoding epoch on i -th trial (Enc_i) and post-encoding ripple window (Ripple) on the same trial ($S_{same} = r(Enc_i, Ripple_i)$). Next, we computed the similarity between other stimuli encoding epochs and ($Enc_{1, 2, \dots, n}$) and post-encoding ripple window on the i -th trial ($S_{diff} = r(Enc_{1, 2, \dots, n}, Ripple_i)$, n denoting the number of different stimuli in the experiment). As the S_{same} might be inflated due to temporal proximity between the Enc_i and $Ripple_i$ (Howard and Kahana, 2002; Polyn et al., 2009), we accounted for the difference in average post-encoding epoch similarity (S_{avg}) between the same and different trials. Specifically, S_{avg} was defined as the difference in similarity between the encoding epoch and the entire post-encoding window on the same trial (S_{same_avg}) or different trials (S_{diff_avg}). For each individual stimulus, the stimulus-specific similarity (S_{spec}) was defined as following:

$$S_{spec} = (S_{same} - S_{diff}) - (S_{same_avg} - S_{diff_avg})$$

Next, the S_{spec} was averaged at subject level and the t -statistics was obtained by comparing the subject-averaged S_{spec} with zero. The similarity null-distribution (S_{shuff}) was created by shuffling the stimulus identity (same vs. different) 1000 times and obtaining the t -statistics as described above. Finally, the cluster statistics was performed by comparing the t -values obtained from S_{spec} with the distribution of t -values obtained from S_{shuff} (non-parametric cluster-based permutation test; $n = 1000$ permutations, $p < 0.05$; Supplementary Fig. 11).

Page 8, Lines 19 - 26

Results

The post-encoding ripple-locked neural activity in the hippocampus (-190 - 20 msec, relative to ripple peak) shows the significant stimulus-specific similarity with the activity during stimulus encoding (~500 - 750 msec following the onset of encoding epoch; non-parametric cluster-based permutation test; $n = 1000$ permutations, $p < 0.05$; Supplementary Fig. 11). The size/timing of significant stimulus similarity during encoding is consistent with previous reports (Zhang et al., 2018; Pacheco Estefan et al., 2019), but might also be driven by the factors specific to present study, such as the comparison between the encoding and post-encoding ripple activity.

Supplementary Fig. 11. The representational similarity map showing the stimulus specificity of the post-encoding ripple representation in the hippocampus. The temporal cluster of significant stimulus-specific similarity (-190 - 20 msec, relative to ripple peak and ~500-750 msec of encoding time) is encircled in black (non-parametric cluster-based permutation test; $n = 1000$ permutations, $p < 0.05$).

4. I do not understand the point of the compression analysis in the case of a time-frequency analysis of EEG. My guess is the idea is coming from the fact that neuronal replay during SWR is compressed, which can be calculated in the case of a sequence of activation of individual neurons (duration of replay). How can this apply to iEEG data? Indeed, the authors find that any compression lowers the reinstatement measure, which might be because in that case compressing is just adding noise and has no "physiological" basis. If I am wrong, I would actually be interested in the explanation.

Response: The Reviewer is correct that compression analysis was included in the submitted version of the manuscript in the context of previously published work on compressed single unit reactivation (e.g. Wilson and McNaughton, 1994; Diba and Buzsaki, 2007). We agree that reinstatement compression in the EEG domain has no clear interpretation and the parallel with single unit sequence compression is difficult to establish. Therefore, we excluded this analysis from the revised version of the manuscript.

Analysis or examples that would strengthen the claims :

1. Provide a better description of the amygdala activity overall, and more specifically around the time of hippocampal SWR. A recent paper claims that there are "ripples" in the human amygdala during Non-REM sleep (Staresina 2020) and that they coordinate with hippocampal SWR. Have the authors observed such typical HFOs in their data? And therefore, could the joint pattern reinstatement be mediated by a more clearcut amygdala pattern than the wideband HFA? This could be also discussed.

Response: The presence of ripples was reported in the human amygdala, based on the similar detection algorithm as used in the present study (Cox et al., 2020). However, the presence of ripples in extrahippocampal structures, especially in the epileptic brain, is still a subject of debate (see the consensus statement by Liu et al., 2022). Nevertheless, it is possible that the periods of high post-encoding similarity in the amygdala, coinciding with hippocampal ripples (Fig. 3), are driven by the amygdala ripple activity. To better characterize the amygdala high-frequency events during hippocampal ripples, we performed the event detection in amygdala using the same algorithm as used for hippocampal ripple detection. This analysis has shown that hippocampal ripples rarely coincide with ripple-like activity in the amygdala within the ± 50 msec window ($5.89 \pm 1.82\%$, mean \pm SEM). Regardless of the nature of amygdala activity during hippocampal ripples, stimulus similarity in the amygdala and hippocampus both coincide with hippocampal ripples, and this coincidence is also predictive of later correct Lure discrimination (Fig. 4). We have elaborated on this control analysis in the revised version of the manuscript (also related to Q7 by Reviewer #2).

Page 47, Lines 12 - 15

Methods

Ripple-like activity in the amygdala was detected using the same algorithm. Coincidence between the hippocampal ripples and ripple-like activity in the amygdala was calculated as the percentage of hippocampal ripples accompanied by amygdala ripple-like activity within the ± 50 msec.

Page 9, Lines 10 - 14

Results

Ripple-like events were also detected in the amygdala, similar to recent reports (Cox et al., 2020). However, only $5.89 \pm 1.82\%$ (mean \pm SEM) of hippocampal ripples were accompanied by ripple-like events in the amygdala within the ± 50 msec window. Thus, the joint increases in post-encoding stimulus similarity during hippocampal ripples (Fig. 4) were not dependent on coincident presence of ripple-like activity in the amygdala.

Page 12, Lines 7 - 17

Discussion

The presence of ripple-like activity, coincident with hippocampal ripples, was recently reported in the human amygdala (Cox et al., 2020) and cortical areas (Vaz et al., 2019; Vaz et al., 2020; Dickey et al., 2022). While the presence of ripples in the extrahippocampal structures, especially in the epileptic brain, is still a subject of debate (Liu et al., 2022), it is possible that the post-encoding stimulus similarity in the amygdala is driven by ripple-like activity. This would be consistent with the co-occurrence of hippocampal ripples and ripple-like activity in temporal cortex during memory retrieval (Dickey et al., 2022). However, only ~5% of hippocampal ripples were accompanied by ripple-like activity in the amygdala. Regardless of the nature of ripple-like activity in the amygdala, the relatively low coincidence with hippocampal ripples suggests the sufficiency of hippocampal ripples alone for coordinating the joint stimulus representation across the structures (Fig. 4a, Supplementary Fig. 12).

2. Trials are separated a priori according to the 3 mean criteria (valence, arousal, success) and the data analyzed separately for these (Fig 3d, correct/incorrect). It would be interesting if the hippocampus, amygdala or joint responses can predict the success of the response. Trials without SWR could also be included in this analysis that is more "agnostic", and thus, in my eyes, complementary.

Response: We agree with the Reviewer that the association between the ripple-locked activity in the amygdala and hippocampus or joint activity across the structures and later correct discrimination is critical for understanding the mechanisms promoting the emotional memory. Significant correlation between the ripple-locked stimulus similarity in the hippocampus and correct Lure discrimination is shown in Fig. 3c, while the significant association between the joint ripple-locked similarity in amygdala and hippocampus is shown in Fig. 4a. The analysis of ripple-locked dynamics (Fig. 3, 4 and related Supplementary Figs) was by definition limited to trials containing post-encoding ripples. Following the Reviewer's suggestion, we also included the control analysis of trials not containing post-encoding ripples. As this question overlaps with Q3 from this Reviewer, the results of this analysis are presented in more detail as an answer to Q3 (below).

3. The authors chose to focus almost exclusively their analysis on the trials with SWR, and use control shuffling within these trials. I see a potential issue with shuffling the ripple time in such a short window (ie response time under 800ms, and considered time window for peri-ripple analysis 500ms!).

Response: The time window for shuffling the ripple timestamps consisted of the post-encoding epoch (self-paced, up to 2000 msec) and subsequent fixation epoch (1000 msec; Fig. 1a). We apologize for imprecise description in the original version of the manuscript, which has been clarified in the revised version, along with the

more detailed explanation of the shuffling procedure (also related to Q8 from Reviewer #1).

Page 50, Line 30 - Page 51, Line 3

Methods

Circular jittering denotes the method of event time shuffling within a limited time window. In the context of the present study, the time window is defined by the onset of post-encoding epoch and the offset of subsequent cross-fixation (Fig. 1a). For example, if the ripple occurred 200 msec after the post-encoding onset and the randomly generated shuffled distance is -500 msec, the assigned shuffled ripple timestamp would be -300 msec prior to offset of the cross-fixation epoch.

I suggest making use of the non-SWR trials to develop the analysis and potentially the claims (is there an anti-correlation between the success of the trial and the intensity of the theta power for example?) and also to use as a control.

Response: We thank the Reviewer for these suggestions. We tested the association between the post-encoding theta power and later correct Lure discrimination using the logistic regression. This analysis showed no significant association between the post-encoding theta power and later correct Lure discrimination ($p = 0.851$, $\beta = 0.008$, $t = 0.187$). We added this result in the revised version of the manuscript.

Page 43, Lines 8 - 10

Methods

The association between the post-encoding theta power on ripple channels and later correct Lure discrimination was tested using the logistic regression ($p < 0.05$).

Page 6, Lines 28 - 29

Results

There was no significant association between the post-encoding theta power and later correct Lure discrimination ($p = 0.851$, $\beta = 0.008$, $t = 0.187$; logistic regression).

A possibility would be to analyze those non-SWR trials "as if" they were SWR trials. The "fake" event time used to align the correlation coefficient could be the average time at which the ripple occurs in SWR trials. Also, theta power provides you with a non-binary measure (unlike SWR occurrence) with which to correlate other parameters.

Response: Following the Reviewer's suggestion (also related to Q2), we included the analysis of trials not containing post-encoding ripples. First, we calculated the

subject-averaged post-encoding ripple times (relative to response times). Next, the subject-average post-encoding ripple times were used as the surrogate events in the event-locked analysis on the trials not containing post-encoding ripples. This analysis has shown that the surrogate event-locked post-encoding similarity is not significantly different based on the stimulus arousal level or later correct Lure discrimination (non-parametric cluster-based permutation test, p 's > 0.05; Supplementary Fig. 10).

Page 51, Lines 21 - 25

Methods

As an additional control analysis, we averaged the post-encoding ripple times at subject level, using response times as reference points. The average post-encoding ripple times were used to compare the event-locked stimulus similarity on the trials not containing post-encoding ripples (non-parametric cluster-based permutation test, p 's > 0.05; Supplementary Fig. 10).

Page 8, Lines 13 - 17

Results

The stimulus similarity on the trials not containing post-encoding ripples was not significantly different based on the stimulus arousal level or later correct Lure discrimination (non-parametric cluster-based permutation test, p 's > 0.05; Supplementary Fig. 10). This result further highlights the role of post-encoding ripples/ripple-locked similarity in consolidation of emotional memories.

Page 33, Lines 1 - 6

Supplementary Fig. 10. Surrogate event-locked similarity on the trials not containing post-encoding ripples. Post-encoding similarity in the amygdala (left) and hippocampus (right) was not associated with the stimulus arousal (top) or later correct Lure discrimination (bottom). Non-parametric cluster-based permutation test, p 's > 0.05.

Reviewer #3 (Remarks to the Author: Reproducibility):

The paper, including the methods, is very clearly written and would allow for a reproduction of the work should a researcher get access to this type of recordings in humans.

Response: We thank the Reviewer for acknowledging the presentation clarity. The data/code was deposited at Zenodo online repository ([10.5281/zenodo.7608723](https://zenodo.org/doi/10.5281/zenodo.7608723)) and will be made available upon publication.

1. Increase the size of the axis and legends in the figures. Some of them are almost unreadable even when zooming in.

Response: We have changed the axis and legend sizes across the figures in the revised version, in the attempt to improve the figure readability.

Reviewer #4 (Remarks to the Author: Overall significance):

In the current manuscript, Zhang et al. examined whether sharp-wave ripples in the amygdala and hippocampus at various stages of memory processing relate to accurate memory discrimination of emotional stimuli. They report that higher post-encoding aSWR count was related to stimulus-related arousal but not valence. This neural measure was also related to successful memory discrimination and were distinct from non-specific broadband fluctuations in power. The results also show a dissociation in the relationship between the memory reinstatement processes in the amygdala and hippocampus, with the former relating to stimulus-linked arousal and the latter relating to accurate memory discriminations. Finally, post-encoding amygdala reinstatement processes preceded hippocampal reinstatement by ~100ms and this pattern was related to subsequent memory discrimination, suggesting a directional influence of the amygdala over hippocampal memory reactivation.

I found the results interesting and consistent with prior findings regarding interactions between the amygdala and hippocampus and emotional memory formation. The examination of SWR⁺AEs represents an important extension of existing work both in rodents (during sleep) and in human work on post-encoding patterns of hippocampal/amygdala connectivity related to emotional memory. The current study helps shed new light on the functional role of awake SWR⁺AEs in the prioritization of emotional memories, which will be of broad interest to many readers who study learning and memory.

However, the scholarship could be improved by choosing more appropriate references. Namely, the link between some citations is over-generalized and isn't the best reflection of a given datapoint/claim. One example is the statement, "Stimulus-induced arousal (irrespective of valence) was associated with better memory discrimination, confirming previous reports". Two of these are review papers on broad emotional memory effects (not discrimination, per se), and the only empirical

one (Szollosi and Racsmány, 2020) is reporting differences in Lure discrimination index, which accounts for response biases (see Comment #1);

Response: We apologize for not citing all the relevant work in the original submission, due to the reference number limit, which we attempted to correct in the revised version. In particular, the association between stimulus arousal and memory discrimination was supported by citing multiple empirical studies (Kensinger and Corkin, 2003; Sharot and Yonelinas, 2008; Grider and Malmberg, 2008; Chainay et al., 2012). We also added a literature on the awake ripple recordings in humans (Norman et al., 2019; Vaz et al., 2019; Vaz et al., 2020; Dickey et al., 2022; Sakon and Kahana, 2022; Sakon et al., 2022) and the effects of stimulation in the amygdala-hippocampal circuitry on the emotional memory consolidation (Inman et al., 2018; Qasim et al., 2023).

the LDI has, as of yet, not been computed here.

Response: We thank the Reviewer for this suggestion which has been incorporated in the revised version of the manuscript and presented as the answer to Q1 (below).

In addition, the statement, "consistent with observations that cholinergic tone promotes theta oscillations and suppresses SWRs^{10,12}" is a big leap and has no grounding in this dataset. The noradrenergic system also plays a similar function and has been heavily implicated in emotional memory processes (e.g., Brown et al., 2005; Sara, 2009). There is no direct evidence in the current study linking these processes to the cholinergic system, so this should be removed or at least moved to the discussion along with a broader consideration of other neuromodulatory candidates.

Response: We agree that the neuromodulatory dynamics was not directly tested in the present experiment (as also noted by Reviewer #2) and has no direct relevance for the interpretation of results. Therefore, we removed the proposal of an association between the cholinergic tone and presence of ripples or theta oscillations from the revised version of the manuscript.

Page 6, Line 31 - Page 7, Line 1

Results

Ripple probability was significantly higher during low theta power periods (Supplementary Fig. 8), consistent with observations *of ripple suppression during periods of pronounced theta oscillations*^{10,12}.

It would also be helpful to cite related work, such as Inman et al. (2018), that shows a relationship between post-encoding amygdala stimulation and memory enhancements.

Response: We are sorry for omitting some relevant literature in the original submission. In the revised version, we discuss the relevance of post-encoding amygdala stimulation (Inman et al., 2018) for the memory encoding/retrieval.

Page 11, Line 30 - Page 12, Line 6

Discussion

Ripples are associated with synchronous activation of neuronal ensembles in the hippocampus and connected structures (Jadhav et al., 2016; Girardeau et al., 2017; Wilber et al., 2017). The onset of ripple-locked similarity in the amygdala prior to ripple peak (Fig. 3c) suggests a sequential process whereby amygdala activation triggers the hippocampal ripple, followed by the cascade of ripple-associated plasticity (King et al., 1999; Rolotti et al., 2022), resulting in higher probability of later correct Lure discrimination. Similarly, electrical stimulation of the amygdala could potentially induce the hippocampal recruitment equivalent to endogenous ripples and facilitate the consolidation of recently encoded content, as demonstrated by Inman et al. (2018). On the other hand, high frequency electrical stimulation of the hippocampus during encoding impairs the emotional memory consolidation (Qasim et al., 2023), suggesting the disruption of consolidation-related hippocampal dynamics.

Reviewer #4 (Remarks to the Author: Impact):

I do think this paper is a strong theoretical contribution to the field of learning and memory. However, my opinion does hinge on additional analyses that will help clarify some of the results. It will also be important for put forth a stronger/clearer interpretation of what post-encoding process is being supported by aSWR in the service of subsequent discriminations (i.e., is about a high-fidelity item representation? context? etc.)

Response: We thank the Reviewer for acknowledging the contribution of the present work for the field of learning and memory. We addressed the issues raised by the Reviewer below.

Reviewer #4 (Remarks to the Author: Strength of the claims):

1) Emotional memory discrimination paradigms often compute a "Lure discrimination index" (LDI) to correct for potential response biases in old/new memory decisions, such as an over tendency to simple endorse most items as "New" (e.g., Leal et al.,

2014). Here, the commonly reported/used behavioral outcome measure is "correct discrimination", which appears to collapse accurate responses across all three trial types. Please compute the LDI for each subject/emotional valence and report the statistical results so that the current results can be generalized to the broader emotional memory literature.

Response: In the revised version, we clarified the scope of the term 'correct discrimination', by specifying the trial category it applies to in a given context (e.g. 'correct Lure discrimination' or 'correct repeat discrimination', also related to Q2 and 3). As pointed out by the Reviewer, the Lure discrimination index (LDI), defined as the difference in the probability of 'Lure' and 'repeat' stimuli being classified as 'New' ($p(\text{'New'}|\text{'Lure'}) - p(\text{'New'}|\text{'repeat'})$), corrects for the general tendency of classifying the stimuli as 'New' (Leal et al., 2014). In the revised version, we present the LDI values across individual subjects and emotional valences (Supplementary Fig. 2, also shown below). In addition, we show no significant effect of stimulus valence on LDI ($p = 0.396$, $F(2, 18) = 0.980$; one-way ANOVA; Supplementary Fig. 2). The effect of stimulus arousal on LDI was tested by comparing the LDI for low- and high-arousal stimuli (as defined using median-split). LDI was significantly higher for high-arousal stimuli ($p = 0.043$, $t(6) = -2.058$, one-tailed paired t-test; Supplementary Fig. 2), reflecting the stronger tendency of the high-arousal stimuli to be classified as 'New'.

Page 45, Lines 4 - 9

Methods

Lure discrimination index (LDI) is defined as the difference in the probability of 'Lure' and 'repeat' stimuli being classified as 'New' ($p(\text{'New'}|\text{'Lure'}) - p(\text{'New'}|\text{'repeat'})$). This procedure corrects for the general tendency of classifying the stimuli as 'New' (Leal et al., 2014). The effect of valence on LDI was tested using one-way ANOVA ($p < 0.05$). The LDI comparison between the low- and high-arousal stimuli was performed using the one-tailed paired t-test ($p < 0.05$).

Page 4, Lines 24 - 32

Results

The Lure discrimination index (LDI, see Methods) is a procedure used to correct for the general tendency of classifying the stimuli as 'New' 14. There was no significant effect of valence on LDI ($p = 0.396$, $F(2, 18) = 0.980$; one-way ANOVA). The effect of valence on LDI shows a considerable inter-subject variability (Supplementary Fig. 2a), both in the terms of absolute values, as well as the distribution across the valences. The reported relations between the valence and LDI are mixed, including both the higher and lower LDI for emotional stimuli (Szöllősi and Racsmany, 2020; Leal et al., 2014; Zheng et al., 2019). LDI was significantly higher

for high-arousal stimuli ($p = 0.043$, $t(6) = -2.058$, one-tailed paired t -test), reflecting the tendency for classifying the high-arousal stimuli as 'New'.

Page 25, Lines 1 - 6

Supplementary Fig. 2. a, Lure discrimination index (LDI) is not significantly associated with the stimulus valence, while it is significantly higher for the high-arousal stimuli reflecting the tendency for high-arousal stimuli to be classified as 'New'. **b**, Valence: $p = 0.396$, $F(2, 18) = 0.980$; one-way ANOVA; Arousal: $p = 0.043$, $t(6) = -2.058$, one-tailed paired t -test. The data from individual subjects are color-coded.

2) It is unclear when the term "correct/accurate discrimination" specifically refers to Lure trials or to any accurate discrimination ("Old" for Repeats and "New" for Novel/Lures, as in Figure 1B). This makes it challenging to unpack some of the findings and should be made very clear throughout.

Response: We apologize for the lack of terminological clarity. For the analysis in Fig. 1b, the term 'correct discrimination' applied to all the trial categories (novel, repeat and Lure), with the category-specific data presented separately. In the revised version, whenever the 'correct discrimination' was limited to the particular trial category, it was explicitly indicated as either the 'correct Lure discrimination' or 'correct repeat discrimination'. This issue also pertains to Q3 (below).

3) If some of the aSWR analyses refer to discrimination accuracy across multiple trial types (presumably Repeat + Lure, because Novel items from encoding don't repeat in phase): Mnemonic discrimination studies use the LDI to index the process of pattern separation, whereas old-item recognition is more often used to index pattern completion. Collapsing these measures together by only assessing "accurate discrimination" as one bin (if indeed correct responses for Repeat + Lure trials are combined) might therefore obscure two opposing memory processes, making it unclear what operations are occurring in the amygdala and hippocampus at retrieval. That could potentially account for null retrieval effect.

Response: Only the Lure trials were included in ripple-based analysis, so there was no lumping of different trial categories in the correct discrimination behavioral metrics used in ripple analysis. We apologize for the unclear description in the original submission, which was clarified in the revised version. In addition, we made a clear distinction between the ‘correct discrimination’ that includes any trial category (Fig. 1b) and the ‘correct discrimination’ of a specific trial category (‘correct Lure discrimination’ or ‘correct repeat discrimination’; also related to Q2).

Page 47, Lines 15 - 16

Methods

Only the Lure trials were used in ripple-based analysis.

It is also unclear what the post-encoding period reflects about the quality of memory consolidation. Are SWR’s capturing a representation in high fidelity to aid a subsequent discrimination? Why is this process supporting discrimination as opposed to other recognition processes? The specific behavioral significance of the post-encoding aSWRs should be discussed in more detail.

Response: We agree that the link between post-encoding ripples and later correct Lure discrimination deserves a more detailed treatment. We elaborated on these points in the Discussion of the revised version (also related to Q8 response).

Page 11, Lines 17 - 29

Discussion (combined with the Discussion related to Q8)

Post-encoding ripples, as defined in the study, occur immediately following the stimulus encoding and might represent the initial stage of stimulus memory consolidation. This stage could be particularly relevant in the present task setting, since the stimulus consolidation might be interfered by the presentation of consecutive stimuli. Transient peaks of post-encoding stimulus similarity during ripples (Fig. 3) might strengthen the connectivity between the neurons participating in stimulus representation, both within the hippocampus and in the hippocampus/amygdala circuitry (Fig. 4). In addition, the better discriminability of high arousal Lure stimuli with low within Lure pair similarity (Supplementary Table 2) might reflect the higher encoding fidelity and/or more efficient consolidation of arousing stimuli, as suggested by the higher ripple-locked post-encoding stimulus

similarity for arousing stimuli (Fig. 3, 4). These mechanisms might result in a higher fidelity stimulus representation at retrieval, allowing a more reliable discrimination from the corresponding Lure stimulus.

Page 13, Lines 1 - 7

Lure stimulus discrimination is arguably a more difficult cognitive task, relative to recognition of novel or repeat stimuli, as reflected by the performance difference between these trial categories (Fig. 1b). While it is possible that post-encoding ripples also support these forms of recognition, it could be obscured by the ceiling effect, as the performance on novel or repeat trials might be high regardless of the ripple presence. In addition, a small number of incorrect discrimination novel or repeat stimuli would not provide enough statistical power to answer this question in the current setting.

4) The logistic regression results in Figure 1C help reveal how emotional parameters relate to Lure discrimination. It would be helpful to perform this same analysis for the "Repeat" trials to demonstrate that there is: a) specificity to rejecting Lures, and b) not driven by response biases. Hypothetically, it could be the case that these results simply reflect endorsements of "New" during the memory test, in which case it would be important to verify that you don't see the same pattern for incorrect discriminations for "Repeat" trials.

Response: We thank the Reviewer for this suggestion. We applied a logistic linear mixed-effects model (LME) to illuminate the effect of stimulus arousal and valence on discrimination of repeat stimuli. There was no significant effect of arousal ($p = 0.785$, $t = -0.27$, $df = 233$, $\beta = -0.024$) or valence ($p = 0.126$, $t = 1.54$, $df = 233$, $\beta = 0.216$) on the correct repeat discrimination, suggesting the selective effect of arousal on discrimination of Lure stimuli. This could be due to the lower difficulty of repeat trials, as the facilitating effect of high stimulus arousal was not detectable due to a very high performance (Fig. 1b).

Page 45, Lines 11 - 12

Methods

The effects of stimulus arousal and valence on the correct discrimination were tested using the linear mixed-effects model (LME; $p < 0.05$).

Page 4, Lines 17 - 22

Results

Neither the stimulus arousal ($p = 0.785$, $t = -0.27$, $df = 233$, $\beta = -0.024$) nor valence ($p = 0.126$, $t = 1.54$, $df = 233$, $\beta = 0.216$) were significantly associated with correct repeat discrimination, supporting the selective effect of arousal on

correct Lure discrimination. This could be due to the lower difficulty of repeat trials, as the correct repeat discrimination performance was already very high (Fig. 1b), limiting the discrimination-enhancing effect of high stimulus arousal.

5) One of the major take-homes from this study is that aSWRs relate to enhanced memory for emotional stimuli. Yet this conclusion isn't fully supported by the current analyses, which show separate main effects for successful memory discrimination and stimulus-related arousal. Although arousal and memory discrimination are correlated, the current results cannot speak to whether these mechanisms/processes simply co-occur and are independent of each other. To argue that these neural processes relate to emotional memory, it would be necessary to identify a memory-by-arousal interaction effect using 2 (Memory Discrimination: successful, unsuccessful) x 2 (Arousal: high, low) repeated-measure ANOVA. I recommend this approach be used wherever possible in the analyses, including those with valence. If there is also a motivation to make specific predictions regarding Lure discrimination versus recognition, then trial type (Type: repeat, Lure) should also be modeled.

Response: As the post-encoding ripple occurrence is associated with both the stimulus arousal level and later correct Lure discrimination (Fig. 2), we agree it would be important to disentangle these effects. The stimulus arousal modulated magnitude of association between the ripple occurrence and later correct Lure discrimination (Supplementary Fig. 4). Specifically, the difference in post-encoding ripple occurrence between the correctly and incorrectly discriminated Lure stimuli was significantly higher for high-arousal Lure stimuli ($p = 0.007$, chi-square = 7.38; Friedman's two-way ANOVA). In addition, association between the post-encoding ripple occurrence and later correct Lure discrimination was significant for both low- and high-arousing stimuli (Supplementary Fig. 4; low-arousal : $p = 0.046$, $z = -1.68$; high-arousal : $p = 0.023$, $z = -2.01$; one-tailed paired Wilcoxon test, $p < 0.05$). We elaborate on these results and possible interpretations in the revised manuscript. This answer is also related to Q4, Reviewer #1 and Q1, Reviewer #2.

Page 43, Lines 2 - 5

Methods

The strength of association between the correct Lure discrimination and post-encoding ripple occurrence was compared for low- and high-arousing stimuli using the non-parametric Friedman's two-way ANOVA ($p < 0.05$; Supplementary Fig. 4).

Page 5, Lines 21 - 29

Results

As the stimulus arousal and correct Lure discrimination are both associated with post-encoding ripple occurrence (Fig. 2c), we tested if the association between the

ripple occurrence and correct Lure discrimination is modulated by the stimulus arousal level. The magnitude of post-encoding ripple modulation based on the correct Lure discrimination was stronger for high-arousal, relative to low-arousal stimuli (Supplementary Fig. 3; $p = 0.007$, $\chi^2 = 7.38$; Friedman's two-way ANOVA). Ripple occurrence was associated with correct Lure discrimination for both arousal levels (low-arousal: $p = 0.046$, $z = -1.68$; high-arousal: $p = 0.023$, $z = -2.01$; one-tailed paired Wilcoxon test, $p < 0.05$).

Page 10, Lines 12 - 14

Discussion

Significant association between the post-encoding ripples and later correct Lure discrimination was present regardless of the stimulus arousal level (Supplementary Fig. 4), but the effect was stronger for high-arousal stimuli.

Page 10, Lines 20 - 23

Post-encoding stimulus similarity (or reinstatement) is implicated in memory consolidation (Carr et al., 2011; Ben-Yakov et al., 2013; Sols et al., 2017; Schreiner and Staudigl, 2021) and peaks during ripples (Fig. 3). Thus, beside the general role of ripples in memory consolidation, these results also imply ripples as a potential mechanism mediating the effects of arousal on memory consolidation.

Page 27, Lines 1 - 8

Supplementary Fig. 4. Stimulus arousal level modulates the association between the post-encoding ripples and later correct Lure discrimination. *The post-encoding ripple modulation by correct Lure discrimination was significantly stronger for high-arousal stimuli ($p = 0.007$, $\chi^2 = 7.38$; Friedman's two-way ANOVA). The association between the post-encoding ripples and later correct Lure discrimination was present regardless of the Lure stimulus arousal level (low-arousal: $p = 0.046$, $z = -1.68$; high-arousal: $p = 0.023$, $z = -2.01$; one-tailed paired Wilcoxon test).*

6) From a conceptual perspective, is it accurate to call the valence rating phase "retrieval"? The rating happens immediately after stimulus presentation, which suggests this is a working memory process (which has also been linked to aSWRs; Jadhav et al., 2012). Further, participants may have already made their valence decisions during image presentation, because they are aware of this instruction and are exposed to many trials. The claim that these neural processes index a retrieval mechanism needs to be clarified and/or alternative mechanisms (e.g., WM) should be considered.

Response: The retrieval task epoch is defined as the period 10-15 minutes following the initial encoding, when the subjects are asked to classify the presented stimuli as 'New' or 'Old' (Fig. 1a). The Reviewer might also be pointing out the usage of the term 'reinstatement' for describing the stimulus similarity during ripple windows. We agree that the higher stimulus similarity during post-encoding ripples does not necessarily represent memory reinstatement, but could be driven by the working memory mechanisms. As noted by the Reviewer, subjects provided stimulus emotional ratings within 2 sec following the stimulus encoding (Fig. 1a), possibly relying on retention of stimulus representation in working memory. This interpretation is also consistent with the proposed role of rodent/human hippocampal ripples and working memory (Jadhav et al., 2012; Sasaki et al., 2018; Zhang et al., 2021). Working memory is traditionally conceptualized as persistent stimulus representation during the delay period (Fuster and Alexander, 1971), while more recent models argue for the intermittent dynamics of stimulus representation (Miller et al., 2018). The post-encoding stimulus similarity in the present study shows discrete peaks around ripples (Fig. 3). Therefore, to the extent that post-encoding stimulus similarity is supported by working memory, it is more consistent with intermittent working memory dynamics, as proposed by Miller et al. (2018). Alternatively, rating the emotional valence of encoded stimuli could benefit from ripple-mediated memory retrieval, accompanied by memory reinstatement. This notion is consistent with the association between the memory retrieval in humans and ripple generation, albeit at a longer time-scale (Norman et al., 2019; Vaz et al., 2019; Vaz et al., 2020; Dickey et al., 2022; Sakon and Kahana, 2022). Regardless if the post-encoding stimulus similarity is driven by working memory or reinstatement, it could facilitate consolidation of represented information into a long-term memory. This hypothesis is supported by the association between the post-encoding ripples and later correct

Lure discrimination (Fig. 2c). Due to the ambiguous mechanisms supporting the post-encoding stimulus representation, we replaced the term 'reinstatement' with 'post-encoding stimulus similarity', as suggested by the Reviewer #1. We provide further elaboration in the revised version of the Introduction and Discussion (also related to Q6, Reviewer #1).

Page 3, Lines 11 - 16

Introduction

Based on these findings, we hypothesized that ripples occurring immediately after stimulus encoding (post-encoding) facilitate emotional memory discrimination through the coordinated hippocampal-amygdala memory reinstatement *or by facilitating the retention of stimulus in working memory. Furthermore, we hypothesize that either of these processes would result in increased stimulus similarity during post-encoding ripples.*

Page 12, Lines 18 - 33

Discussion

The post-encoding stimulus similarity occurs during the stimulus valence rating, immediately following the stimulus encoding (Fig. 1a). Therefore, the higher post-encoding similarity might be driven by either the retention of stimulus representation in working memory or by memory reinstatement. The interpretation that the post-encoding stimulus representation reflects working memory content is consistent with the proposed role of rodent/human hippocampal ripples in working memory (Jadhav et al., 2012; Sasaki et al., 2018; Zhang et al., 2021). While the working memory is traditionally conceptualized as persistence of stimulus-specific activity (Fuster and Alexander, 1971), the post-encoding similarity in the present study is concentrated around ripple peaks (Fig. 3). Therefore, to the extent that post-encoding similarity is driven by working memory, it is consistent with the intermittent representation of working memory content (Miller et al., 2018). On the other hand, the stimulus valence rating during post-encoding epoch could also rely on memory retrieval, which was associated with ripple emergence (Norman et al., 2019; Vaz et al., 2020). Regardless of the underlying mechanisms, the post-encoding ripple-locked stimulus similarity is associated with later correct Lure discrimination. This could be due to the contribution of post-encoding ripples to memory consolidation, resulting in a higher fidelity of encoded representation and higher probability of later correct Lure discrimination.

The prior work that is cited for post-encoding reactivation (e.g., Sols et al., 2017 and Ben-Yakov et al., 2013) involves a task switch (and perceptual) or a blank screen. Neither of these explicitly pertain to a judgement about the preceding image/stimulus, so the nuances here might matter.

Response: In the revised version, we cited additional work, showing a more direct involvement of post-encoding reactivation in memory consolidation (Carr et al., 2011; Schreiner and Staudigl, 2021). While the stimulus rating might contribute to post-encoding ripple emergence, our hypothesis is that the presence of post-encoding ripples and ripple-locked similarity contributes to memory consolidation, regardless of the processes driving the post-encoding ripple generation.

7) Although the RT's didn't differ by arousal or discrimination, it still seems like variability in trial-level RT's need to be addressed. Furthermore, please report whether the RT's differed by emotional valence.

Response: We tested the association between the emotional valence and response times using one-way ANOVA, which has shown no significant effect of emotional valence on response times ($p = 0.749$, $F(2,18) = 0.290$).

Page 4, Lines 22 - 24

Results

Response times were not significantly associated with the stimulus emotional valence ($p = 0.749$, $F(2,18) = 0.290$, one-way ANOVA).

I encourage the authors to use SWR rate (ripple per unit time) instead of a raw count to normalize the number of SWR events based on the actual response window of a given trial (if the goal is to isolate a decision-making process).

Response: We agree that comparing the ripple rate time course during the post-encoding epoch would help identifying the specific periods of significant ripple association with stimulus arousal/correct Lure discrimination. This question is also related to Q3 by Reviewer #2. In the revised version of the manuscript, we demonstrate significant association between post-encoding ripples and later correct Lure discrimination/stimulus arousal for both the post-encoding ripple occurrence (number of post-encoding ripples/trial, Fig. 2, Supplementary Fig. 4) and ripple rates (events/sec, Supplementary Fig. 5).

Page 42, Lines 25 - 34

Methods

The association between post-encoding ripples and stimulus arousal/correct Lure discrimination was tested using two complementary approaches, ripple occurrence

(number of post-encoding ripples/trial, Fig. 2; Supplementary Fig. 4) and ripple rates (number of ripples/sec, Supplementary Fig. 5). The rationale for using ripple occurrence is based on the notion of ripples facilitating memory consolidation as the windows of memory reinstatement (Buzsaki, 2015). Based on the assumption that a larger number of ripples would result in more instances of memory reinstatement, it would result in a stronger effect on consolidation. A complementary approach, using the ripple rates, allows the detection of effects limited to specific time-windows that might be missed in the analysis based on ripple occurrence.

Page 5, Line 29 - Page 6, Line 6

Results

In addition, we tested if the post-encoding ripple association with stimulus arousal/correct Lure discrimination is limited to specific periods during post-encoding epoch by performing the conditional comparisons of ripple rates (number of ripples/s). Post-encoding ripple rates were significantly higher for correctly discriminated, relative to incorrectly discriminated Lure stimuli (Supplementary Fig. 5; $p = 0.005$, -400 to -50 msec relative to response time), and for high-arousal, relative to low-arousal Lure stimuli (Supplementary Fig. 5; $p = 0.035$, -780 to -600 msec; non-parametric cluster-based permutation test). To summarize, the post-encoding ripple associations with stimulus arousal/correct Lure discrimination were present during distinct, non-overlapping time windows, suggesting the distinct temporal relation between these variables and post-encoding ripples.

Page 10, Lines 6 - 12

Discussion

We demonstrate the association between the post-encoding ripples and stimulus arousal/correct Lure discrimination, using the two complementary methods. Ripple occurrence, defined as the average number of ripples during the post-encoding window, may reflect the higher number of memory reinstatement events or higher fidelity stimulus retention in working memory. Another approach, based on the ripple rate (ripples/sec), revealed the ripple temporal dynamics during the post-encoding epoch.

Supplementary Fig. 5. The time-resolved association between ripple rate and stimulus arousal/correct Lure discrimination across the task epochs. a, During stimulus encoding, there was no significant ripple rate difference between the correctly and incorrectly discriminated Lure stimuli (top; $p > 0.05$, non-parametric cluster-based permutation test) or low-arousal and high-arousal Lure stimuli (bottom; $p > 0.05$, non-parametric cluster-based permutation test). b, During post-encoding, ripple rates were significantly higher for correctly discriminated, relative to incorrectly discriminated Lure stimuli (top, $p = 0.005$, -400 to -50 msec relative to response time), and for high-arousal, relative to low-arousal Lure stimuli (bottom, $p = 0.035$, -780 to -600 msec relative to response time, non-parametric cluster-based permutation test).

8) The logistic regression (Fig. 1C) should include all two-way and the three-way interaction term for similarity, valence, and arousal. Similarity is disregarded in all the main brain analyses, but it would be useful to demonstrate that there is no significant interaction with similarity to justify collapsing high and low similarity in the arousal, valence, and discrimination analyses. The results in Figure 1D qualitatively suggest that there might be an interaction; regardless, how do the authors interpret the finding that Lure discrimination is best for high arousal trials that are low vs. high in similarity? This seems to be an important effect and warrants more scrutiny.

Response: We thank the Reviewer for this suggestion. In the revised version, we used a logistic linear mixed-effect model with interaction terms (similarity x valence, similarity x arousal, valence x arousal) and correct Lure discrimination as dependent variable. This analysis has shown the main effects of arousal ($p < 0.001$, $t = 15.782$) and similarity ($p < 0.001$, $t = 50.562$) on the correct Lure discrimination, as well as the significant interaction between the arousal and similarity ($p < 0.001$, $t = 10.327$). The main effect of valence on correct Lure discrimination, as well as the interactions between the similarity and valence or valence and arousal were not significant (p 's < 0.05 ; Supplementary Table 2). This analysis confirmed that the correct Lure discrimination is indeed highest for the high-arousal stimuli of low similarity with the corresponding Lure stimulus. This effect could be due to the stimulus-induced arousal resulting in a higher fidelity encoding, as reflected in higher similarity between the neural activity during encoding epoch and post-encoding ripples (Fig. 3, 4). Stimulus representation during post-encoding ripples might further facilitate the consolidation process, resulting in a more detailed representation available for later comparison with Lure stimuli.

Page 4, Line 33 - Page 5, Line 5

Results

Correct Lure discrimination was significantly associated with stimulus arousal ($p < 0.001$, $t = 15.782$) and similarity ($p < 0.001$, $t = 50.562$), while there was no significant association with valence ($p = 0.308$, $t = 1.020$). In addition, there was a significant interaction between the arousal and similarity, ($p < 0.001$, $t = 10.327$), reflecting the highest correct Lure discrimination for the high-arousal stimuli of low similarity. There was no other significant interaction between the experimental variables (arousal x valence, similarity x valence, arousal x similarity x valence; Supplementary Table 2).

Page 11, Lines 17 - 29

Discussion (combined with the Discussion related to Q3)

Post-encoding ripples, as defined in the study, occur immediately following the stimulus encoding and might represent the initial stage of stimulus memory

consolidation. This stage could be particularly relevant in the present task setting, since the stimulus consolidation might be interfered by the presentation of consecutive stimuli. Transient peaks of post-encoding stimulus similarity during ripples (Fig. 3) might strengthen the connectivity between the neurons participating in stimulus representation, both within the hippocampus and in the hippocampus/amygdala circuitry (Fig. 4). In addition, the better discriminability of high arousal Lure stimuli with low within Lure pair similarity (Supplementary Table 2) might reflect the higher encoding fidelity and/or more efficient consolidation of arousing stimuli, as suggested by the higher ripple-locked post-encoding stimulus similarity for arousing stimuli (Fig. 3, 4). These mechanisms might result in a higher fidelity stimulus representation at retrieval, allowing a more reliable discrimination from the corresponding Lure stimulus.

9) Stimuli. Please report more information about the image stimuli, including the database they were selected from. Furthermore, it seems that the normative emotion ratings were based on a previous study (Leal et al., 2014). If this is true, please cite that paper.

Response: The stimuli were collected as a freely available online content. Normative emotional valence, arousal and similarity ratings were obtained from healthy volunteers and the same ratings were indeed used in Leal et al. (2014), which are cited in the revised version.

Page 44, Lines 5 - 7

Methods

The same set of stimuli was used across subjects. In addition, the valence, arousal and similarity of each stimulus were rated by separate cohorts of healthy subjects (*also used in Leal et al., 2014*).

One additional issue is that the raters in Leal et al. (2014) reported negative stimuli to be significantly more arousing than positive stimuli; this may confound any interpretations related to the effects being purely driven by arousal vs. valence (i.e., it may be the case that the reported effects are a mixture of both, with arousal being the prevailing factor). This limitation should be addressed in the methods or discussion.

Response: We thank the Reviewer for pointing out this potential confound, which is acknowledged in the revised version of the Discussion.

Page 10, Lines 14 - 18

Discussion

The higher arousal ratings of negative stimuli (Leal et al., 2014) could potentially confound the effects of arousal and valence in the present study. While we can't entirely rule out this interpretation, there was no significant effect of stimulus valence on the correct Lure discrimination (Fig. 1) or encoding/post-encoding ripples (Supplementary Fig. 3).

10) Extended Data Figure 1. It's unclear whether these ratings are from the additional samples or from the subjects in the current study. Please specify. It's also unclear in the main text whether the trial types (e.g., defined by valence) are based on the normative ratings or ratings from the current subjects - though it seems to be the former. I'd encourage the authors to instead define the conditions based on the actual subjects' ratings from this study as opposed to the normative rating-related categorizations, especially given that some stimuli do not fit clearly within the predefined categories (e.g., several neutral datapoints would be classified as Positive based on their valence ratings in the Figure).

Response: We are sorry for the unclear method description. The ratings in the Supplementary Fig. 1a are based on the normative ratings obtained from healthy subjects (also used in Leal et al., 2014). This has been specified in the Supplementary Fig. 1 caption (revised version). The ratings collected from study subjects were categorical, as collecting the continuous ratings on the sliding scale would affect the response times, depending on the rating position on the scale. In addition, continuous ratings from study subjects were not collected outside the task context due to their time limitation in the epilepsy monitoring unit. Finally, the correspondence between the continuous (healthy subjects) and categorical (study subjects) ratings was ~85% (Supplementary Fig. 1b). The rationale for using the normative ratings from healthy subjects was that the correlation with neurophysiological signals is more feasible for continuous, relative to categorical behavioral variables. This rationale was elaborated in the revised version of the manuscript.

Page 44, Lines 16 - 22

Methods

The rationale for obtaining the categorical ratings from study subjects was the need of using the sliding scale for obtaining the continuous ratings, which would introduce systematic difference in response times, depending on the scale distance. The continuous ratings from healthy subjects were used for behavioral/neurophysiological correlation based on the: a) better feasibility of continuous behavioral variables for correlation with neural signals and b) high correspondence between the continuous (healthy subjects) and categorical (study subjects) ratings (~85%; Supplementary Fig. 1b).

Results

Supplementary Fig. 1. a, Positive and negative valenced stimuli (*based on healthy subject ratings*) are associated with higher stimulus-induced arousal, relative to neutral valence stimuli ($***p < 0.001$, Wilcoxon rank-sum test). **b**, Stimuli valence ratings of study subjects are highly similar to the healthy population (match rate = $85.3 \pm 1.3\%$).

Reviewer #4 (Remarks to the Author: Reproducibility):

NOTE: Most of my comments related to statistical analyses and appropriateness are described in the "Strengths..." section.

1) Participant exclusions. It is stated that "due to the low number of detected aSWRs, one subject was eliminated from the aSWR-related analysis." What was this number/percentage?

Response: We apologize for the mistake when describing the exclusion criteria in the original submission. The exclusion threshold was not based on the absolute number of detected events, but relative to the distribution of ripple numbers across all the subjects. Specifically, we used the z-score threshold < -2 , which identified subject 1 as an outlier. In addition, based on the suggestion from Reviewer #1, we run the ripple detection on the synthetic signal with the same spectral properties as the original hippocampal signal and using the same detection algorithm (Supplementary Fig. 14). This analysis has shown that in all the subjects, except subject 1, the number of detected ripples in the hippocampal signal exceeded the number of detected events in the synthetic signal, providing additional justification for excluding this subject. This response is also related to Q3 (Reviewer #1) and Q2 (Reviewer #2).

Methods

Only the channels with *z-scored number of detected ripples* > -2 and number of detected events higher than in the synthetic signal (see Ripple detection from synthetic signal) were used in the analysis.

Ripple detection from synthetic signal

The power spectral density (PSD) was calculated for each hippocampal channel used in ripple detection. Next, the channel-specific filter was applied on a random signal with Gaussian distribution, resulting in a synthetic signal with identical spectral slope as the hippocampal signal. Specifically, the synthetic signal was first transformed to frequency domain by N-point Fourier transform (N denoting the number of datapoints in the signal), followed by multiplication of resulting spectrum by the PSD coefficients and application of inverse Fourier transform, to convert the signal back to time domain. The resulting synthetic signal mimicked the hippocampal channel-specific spectral characteristics. Next, the ripple detection procedure (see Ripple detection) was applied on the synthetic signal. Finally, as an additional control, the numbers of detected events were compared between the hippocampal iEEG signal and channel-specific synthetic signal. In all the 6 subjects used in the ripple-based analysis (subjects 2-7), the numbers of detected ripples were higher than the numbers obtained from the subject-specific synthetic signals (Supplementary Fig. 14). Subject 1 was excluded from the analysis based on the two criteria: a) low number of detected putative ripple events (z-score < -2, relative to distribution across the subjects) and b) the number of detected events lower than chance level, defined as the number of detected events in synthetic signal (Supplementary Fig. 14).

Supplementary Fig. 14. Ripple detection in the hippocampal signal and synthetic signal. Comparisons between the numbers of ripples detected in hippocampal channels (left) and number of events detected in subject-specific synthetic signals of the same spectral characteristics (right). In all the subjects included in ripple analysis (subjects 2-7) ripple detection in hippocampal signal was higher than in synthetic signal. Subject 1 was excluded from the ripple-based analysis due to low ripple detection in hippocampal signal (z -score < -2 , relative to distribution across the subjects) and the number of detected putative ripples lower than in synthetic signals of the same spectral characteristics.

In addition, it's stated that "only the subjects with the correct discrimination rate of novel trials $\geq 85\%$ " were included in the analyses. What was the basis for this exclusion criterion? Was this subject's performance > 2.5 SD's below the mean? It seems arbitrary to make this exclusion based on a single trial type, and I think it would be more appropriate to only exclude this participant if their performance is significantly below the average performance across all trial types (i.e., indicative of poor task performance overall). The specified performance threshold is also very high; please report the reasoning behind this exclusion, the performance value of the excluded subject for Novel trials and cite any relevant literature to motivate this decision. Without sufficient justification, this participant should be included in all analyses.

Response: A critical prerequisite for collecting the high-quality data in cognitive experiments is that the subjects show no attention lapses. This assumption would be supported by the very high performance on the novel trials - the easiest trial category, requiring classification of stimuli presented for the first-time and not similar to any previous stimuli as 'New'. The only subject excluded based on the behavioral

performance correctly discriminated only 60.5% of novel trials ($z = -2.34$), while all the subjects included in behavioral analysis exceeded 85% ($93.3 \pm 1.6\%$, mean \pm SEM). In addition, stimulus discrimination accuracy over all 3 trial types collapsed was very close to the chance level (51.7%, $z = -2.43$; chance level = 50%), suggesting possible attention lapses and questioning the utility of data collected from this subject. We have elaborated on this exclusion rationale in the Methods section of the revised manuscript.

Page 42, Lines 8 - 15

Methods

The behavioral inclusion threshold (>85% performance on novel trials) was used as a sensitive indicator of subjects attention level. Correct performance on novel trials required classifying the stimuli encountered for the first time and with no similarity to previously encountered stimuli as 'New'. One subject was excluded from the behavioral analysis due to low performance on novel trials (60.5%, $z = -2.34$), while the rest of the subjects performed at much higher level ($93.3 \pm 1.6\%$, mean \pm SEM). In addition, this subject performed close to the chance level (50%) across all the trial types combined (51.7%, $z = -2.43$).

2) Minor: the paneling in Figure 3 is a little confusing and could benefit from some reorganization (for example, organizing a horizontally as opposed to vertically). Panel E looks like an extension of C. Additionally, the "Beta" label on the Y-axis of Figure 1 could be more informative. The colored diagram to the right is helpful but beta is a very non-descriptive term.

Response: We thank the Reviewer for these suggestions. Fig. 1 and 3 were changed accordingly in the revised version. In addition, we defined the beta label in the figure caption. Finally, we tried to achieve the optimal readability of the figures added during revision.

References:

Vaz, A.P., Inati, S.K., Brunel, N. and Zaghoul, K.A., 2019. Coupled ripple oscillations between the medial temporal lobe and neocortex retrieve human memory. *Science*, 363(6430), pp.975-978.

Dickey, C.W., Sargsyan, A., Madsen, J.R., Eskandar, E.N., Cash, S.S. and Halgren, E., 2021. Travelling spindles create necessary conditions for spike-timing-dependent plasticity in humans. *Nature communications*, 12(1), pp.1-15.

- Ramirez-Villegas, J.F., Logothetis, N.K. and Besserve, M., 2015. Diversity of sharp-wave–ripple LFP signatures reveals differentiated brain-wide dynamical events. *Proceedings of the National Academy of Sciences*, 112(46), pp.E6379-E6387.
- Staresina, B.P., Bergmann, T.O., Bonnefond, M., Van Der Meij, R., Jensen, O., Deuker, L., Elger, C.E., Axmacher, N. and Fell, J., 2015. Hierarchical nesting of slow oscillations, spindles and ripples in the human hippocampus during sleep. *Nature neuroscience*, 18(11), pp.1679-1686.
- Norman, Y., Raccach, O., Liu, S., Parvizi, J. and Malach, R., 2021. Hippocampal ripples and their coordinated dialogue with the default mode network during recent and remote recollection. *Neuron*, 109(17), pp.2767-2780.
- Liu, A.A., Henin, S., Abbaspoor, S., Bragin, A., Buffalo, E.A., Farrell, J.S., Foster, D.J., Frank, L.M., Gedankien, T., Gotman, J. and Guidera, J.A., 2022. A consensus statement on detection of hippocampal sharp wave ripples and differentiation from other fast oscillations. *Nature communications*, 13(1), pp.1-14.
- Vaz, A.P., Wittig Jr, J.H., Inati, S.K. and Zaghoul, K.A., 2020. Replay of cortical spiking sequences during human memory retrieval. *Science*, 367(6482), pp.1131-1134.
- Sakon, J.J. and Kahana, M.J., 2022. Hippocampal ripples signal contextually mediated episodic recall. *Proceedings of the National Academy of Sciences*, 119(40), p.e2201657119.
- Carr, M.F., Jadhav, S.P. and Frank, L.M., 2011. Hippocampal replay in the awake state: a potential substrate for memory consolidation and retrieval. *Nature neuroscience*, 14(2), pp.147-153.
- Ben-Yakov, A., Eshel, N. and Dudai, Y., 2013. Hippocampal immediate poststimulus activity in the encoding of consecutive naturalistic episodes. *Journal of Experimental Psychology: General*, 142(4), p.1255.
- Sols, I., DuBrow, S., Davachi, L. and Fuentemilla, L., 2017. Event boundaries trigger rapid memory reinstatement of the prior events to promote their representation in long-term memory. *Current Biology*, 27(22), pp.3499-3504.
- Schreiner, T., Petzka, M., Staudigl, T. and Staresina, B.P., 2021. Endogenous memory reactivation during sleep in humans is clocked by slow oscillation-spindle complexes. *Nature communications*, 12(1), pp.1-10.

Jadhav, S.P., Kemere, C., German, P.W. and Frank, L.M., 2012. Awake hippocampal sharp-wave ripples support spatial memory. *Science*, 336(6087), pp.1454-1458.

Zhang, Y., Cao, L., Varga, V., Jing, M., Karadas, M., Li, Y. and Buzsáki, G., 2021. Cholinergic suppression of hippocampal sharp-wave ripples impairs working memory. *Proceedings of the National Academy of Sciences*, 118(15), p.e2016432118.

Sasaki, T., Piatti, V.C., Hwaun, E., Ahmadi, S., Lisman, J.E., Leutgeb, S. and Leutgeb, J.K., 2018. Dentate network activity is necessary for spatial working memory by supporting CA3 sharp-wave ripple generation and prospective firing of CA3 neurons. *Nature neuroscience*, 21(2), pp.258-269.

Fuster, J.M. and Alexander, G.E., 1971. Neuron activity related to short-term memory. *Science*, 173(3997), pp.652-654.

Miller, E.K., Lundqvist, M. and Bastos, A.M., 2018. Working Memory 2.0. *Neuron*, 100(2), pp.463-475.

Norman, Y., Yeagle, E.M., Khuvis, S., Harel, M., Mehta, A.D. and Malach, R., 2019. Hippocampal sharp-wave ripples linked to visual episodic recollection in humans. *Science*, 365(6454), p.eaax1030.

Simonnet, J. and Brecht, M., 2019. Burst firing and spatial coding in subicular principal cells. *Journal of Neuroscience*, 39(19), pp.3651-3662.

Reitich-Stolero, T. and Paz, R., 2019. Affective memory rehearsal with temporal sequences in amygdala neurons. *Nature neuroscience*, 22(12), pp.2050-2059.

Buzsáki, G., 2015. Hippocampal sharp wave-ripple: A cognitive biomarker for episodic memory and planning. *Hippocampus*, 25(10), pp.1073-1188.

Cohen, M.X., 2014. *Analyzing neural time series data: theory and practice*. MIT press.

Helfrich, R.F., Lendner, J.D., Mander, B.A., Guillen, H., Paff, M., Mnatsakanyan, L., Vadera, S., Walker, M.P., Lin, J.J. and Knight, R.T., 2019. Bidirectional prefrontal-hippocampal dynamics organize information transfer during sleep in humans. *Nature communications*, 10(1), pp.1-16.

Sun, C., Yang, F., Wang, C., Wang, Z., Zhang, Y., Ming, D. and Du, J., 2018. Mutual information-based brain network analysis in post-stroke patients with different levels of depression. *Frontiers in Human Neuroscience*, 12, p.285.

Dickey, C.W., Verzhbinsky, I.A., Jiang, X., Rosen, B.Q., Kajfez, S., Eskandar, E.N., Gonzalez-Martinez, J., Cash, S.S. and Halgren, E., 2022. Cortical ripples during NREM sleep and waking in humans. *Journal of Neuroscience*, 42(42), pp.7931-7946.

Sakon, J.J., Halpern, D.J., Schonhaut, D.R. and Kahana, M.J., 2022. Human hippocampal ripples signal encoding of episodic memories. *bioRxiv*, pp.2022-10.

Benjamini, Y. and Hochberg, Y., 1995. Controlling the false discovery rate: a practical and powerful approach to multiple testing. *Journal of the Royal statistical society: series B (Methodological)*, 57(1), pp.289-300.

Deuker, L., Olligs, J., Fell, J., Kranz, T.A., Mormann, F., Montag, C., Reuter, M., Elger, C.E. and Axmacher, N., 2013. Memory consolidation by replay of stimulus-specific neural activity. *Journal of Neuroscience*, 33(49), pp.19373-19383.

Staresina, B.P., Alink, A., Kriegeskorte, N. and Henson, R.N., 2013. Awake reactivation predicts memory in humans. *Proceedings of the National Academy of Sciences*, 110(52), pp.21159-21164.

Bird, C.M., Keidel, J.L., Ing, L.P., Horner, A.J. and Burgess, N., 2015. Consolidation of complex events via reinstatement in posterior cingulate cortex. *Journal of Neuroscience*, 35(43), pp.14426-14434.

Axmacher, N., Elger, C.E. and Fell, J., 2008. Ripples in the medial temporal lobe are relevant for human memory consolidation. *Brain*, 131(7), pp.1806-1817.

Zhang, H., Fell, J. and Axmacher, N., 2018. Electrophysiological mechanisms of human memory consolidation. *Nature communications*, 9(1), pp.1-11.

Pacheco Estefan, D., Sánchez-Fibla, M., Duff, A., Principe, A., Rocamora, R., Zhang, H., Axmacher, N. and Verschure, P.F., 2019. Coordinated representational reinstatement in the human hippocampus and lateral temporal cortex during episodic memory retrieval. *Nature communications*, 10(1), p.2255.

Frömer, R., Maier, M. and Abdel Rahman, R., 2018. Group-level EEG-processing pipeline for flexible single trial-based analyses including linear mixed models. *Frontiers in neuroscience*, 12, p.48.

Jas, M., Larson, E., Engemann, D.A., Leppäkangas, J., Taulu, S., Hämäläinen, M. and Gramfort, A., 2018. A reproducible MEG/EEG group study with the MNE software: recommendations, quality assessments, and good practices. *Frontiers in neuroscience*, 12, p.530.

Rocchi, L., Di Santo, A., Brown, K., Ibáñez, J., Casula, E., Rawji, V., Di Lazzaro, V., Koch, G. and Rothwell, J., 2021. Disentangling EEG responses to TMS due to cortical and peripheral activations. *Brain stimulation*, 14(1), pp.4-18.

Costa, M., Lozano-Soldevilla, D., Gil-Nagel, A., Toledano, R., Oehr, C.R., Kunz, L., Yebra, M., Mendez-Bertolo, C., Stieglitz, L., Sarnthein, J. and Axmacher, N., 2022. Aversive memory formation in humans involves an amygdala-hippocampus phase code. *Nature Communications*, 13(1), pp.1-16.

Cox, R., Rüber, T., Staresina, B.P. and Fell, J., 2020. Sharp wave-ripples in human amygdala and their coordination with hippocampus during NREM sleep. *Cerebral cortex communications*, 1(1), p.tgaa051.

Staresina, B.P., Michelmann, S., Bonnefond, M., Jensen, O., Axmacher, N. and Fell, J., 2016. Hippocampal pattern completion is linked to gamma power increases and alpha power decreases during recollection. *Elife*, 5, p.e17397.

Salmela, V., Salo, E., Salmi, J. and Alho, K., 2018. Spatiotemporal dynamics of attention networks revealed by representational similarity analysis of EEG and fMRI. *Cerebral Cortex*, 28(2), pp.549-560.

Kriegeskorte, N., Mur, M. and Bandettini, P.A., 2008. Representational similarity analysis-connecting the branches of systems neuroscience. *Frontiers in systems neuroscience*, p.4.

Bruffaerts, R., Dupont, P., Peeters, R., De Deyne, S., Storms, G. and Vandenberghe, R., 2013. Similarity of fMRI activity patterns in left perirhinal cortex reflects semantic similarity between words. *Journal of Neuroscience*, 33(47), pp.18597-18607.

Kaneshiro, B., Perreau Guimaraes, M., Kim, H.S., Norcia, A.M. and Suppes, P., 2015. A representational similarity analysis of the dynamics of object processing using single-trial EEG classification. *Plos one*, 10(8), p.e0135697.

Wilson, M.A. and McNaughton, B.L., 1994. Reactivation of hippocampal ensemble memories during sleep. *Science*, 265(5172), pp.676-679.

Diba, K. and Buzsáki, G., 2007. Forward and reverse hippocampal place-cell sequences during ripples. *Nature neuroscience*, 10(10), pp.1241-1242.

Kensinger, E.A. and Corkin, S., 2003. Memory enhancement for emotional words: Are emotional words more vividly remembered than neutral words?. *Memory & cognition*, 31(8), pp.1169-1180.

Sharot, T. and Yonelinas, A.P., 2008. Differential time-dependent effects of emotion on recollective experience and memory for contextual information. *Cognition*, 106(1), pp.538-547.

Sharot, T. and Yonelinas, A.P., 2008. Differential time-dependent effects of emotion on recollective experience and memory for contextual information. *Cognition*, 106(1), pp.538-547.

Grider, R.C. and Malmberg, K.J., 2008. Discriminating between changes in bias and changes in accuracy for recognition memory of emotional stimuli. *Memory & cognition*, 36(5), pp.933-946.

Chainay, H., Michael, G.A., Vert-Pré, M., Landré, L. and Plasson, A., 2012. Emotional enhancement of immediate memory: Positive pictorial stimuli are better recognized than neutral or negative pictorial stimuli. *Advances in Cognitive Psychology*, 8(3), p.255.

Inman, C.S., Manns, J.R., Bijanki, K.R., Bass, D.I., Hamann, S., Drane, D.L., Fasano, R.E., Kovach, C.K., Gross, R.E. and Willie, J.T., 2018. Direct electrical stimulation of the amygdala enhances declarative memory in humans. *Proceedings of the National Academy of Sciences*, 115(1), pp.98-103.

Qasim, S.E., Mohan, U.R., Stein, J.M. and Jacobs, J., 2023. Neuronal activity in the human amygdala and hippocampus enhances emotional memory encoding. *Nature Human Behaviour*, pp.1-11.

Jadhav, S.P., Rothschild, G., Roumis, D.K. and Frank, L.M., 2016. Coordinated excitation and inhibition of prefrontal ensembles during awake hippocampal sharp-wave ripple events. *Neuron*, 90(1), pp.113-127.

King, C., Henze, D.A., Leinekugel, X. and Buzsáki, G., 1999. Hebbian modification of a hippocampal population pattern in the rat. *The Journal of physiology*, 521(Pt 1), p.159.

Rolotti, S.V., Blockus, H., Sparks, F.T., Priestley, J.B. and Losonczy, A., 2022. Reorganization of CA1 dendritic dynamics by hippocampal sharp-wave ripples during learning. *Neuron*, 110(6), pp.977-991.

Leal, S.L., Tighe, S.K., Jones, C.K. and Yassa, M.A., 2014. Pattern separation of emotional information in hippocampal dentate and CA3. *Hippocampus*, 24(9), pp.1146-1155.

Stiernströmer, E.S., Wolgast, M., Johansson, M., Innes-Ker, Å. and Cardeña, E., 2018. The effect of variations of emotional expressions on mnemonic discrimination

and traditional recognition memory. *Journal of Cognitive Psychology*, 30(5-6), pp.547-557.

Zheng, J., Stevenson, R.F., Mander, B.A., Mnatsakanyan, L., Hsu, F.P., Vadera, S., Knight, R.T., Yassa, M.A. and Lin, J.J., 2019. Multiplexing of theta and alpha rhythms in the amygdala-hippocampal circuit supports pattern separation of emotional information. *Neuron*, 102(4), pp.887-898.

Szóllósi, Á. and Racsmány, M., 2020. Enhanced mnemonic discrimination for emotional memories: the role of arousal in interference resolution. *Memory & cognition*, 48, pp.1032-1045.

Reviewer comments, second version:

Reviewer #1 (Remarks to the Author: Overall significance):

This is a revised version of a manuscript that I have reviewed earlier. My assessment of the overall significance has not changed. If anything, it has slightly improved since the authors have done a very good job in addressing my previous concerns. The only comment they were not able to address refers to the low number of subjects. I was hoping to see that more than one session was recorded per subject and that the results were consistent across these sessions. However, I understand this was not the case and instead, the authors show single subject data and show stability of patterns across subjects, which is reassuring and addresses my concern. I am also very happy to see that the number of ripples detected in the real signal is consistently higher than in a synthetic signal. Overall, all my previous concerns have been addressed and I have no further concerns about this manuscript.

Reviewer #2 (Remarks to the Author: Overall significance):

The rebuttal from Zhang et al. has not addressed all previous concerns. Additionally, their analyses have raised new concerns. There are multiple results which are inconsistent and analyses which are not presented clearly. Although the topic of the manuscript is interesting and timely, I am not convinced that the results support the authors' claims.

Reviewer #2 (Remarks to the Author: Strength of the claims):

1) In Supp Fig 4, the authors used ripple number and not ripple rate. This is a consistent issue throughout the manuscript (e.g., Fig 2c, Supp Fig 3, Supp Fig 6). All reported differences in ripple occurrence between conditions or time windows need to control for differences in response time. The reported differences in ripple number may have been the result of minor differences in response time (as the effects are small). Response time can be controlled for using ripple rate or by using response time as a control variable in a regression model.

2) The ripple counts in Supp Fig 4 do not align with the ripple counts in Fig 2c bottom-right. Supp Fig 4 shows ripple count ~ 0.3 while Fig 2c bottom-right shows ripple count of ~ 0.55 . Since these figures consist of the same ripples in the same time windows and conditions, the ripple counts should be similar.

3) If the authors want to claim that arousal and lure-discrimination both have main effects, they need to show a statistically significant effects of arousal for high and low lure-discrimination (not shown in Supp Fig 4). Further, Supp Fig 4 suggests there may be an interaction effect between lure-discrimination and arousal. If this is the case, it requires a different interpretation than two main effects.

4) In Fig 2c, the ripple counts are similar for the stimulus encoding and the post-encoding phases.

However, the average time of the post-encoding phase (~700 msec) is less than half the time of the stimulus encoding phase (2000 msec). Therefore, one expects the ripple rates in Supp Fig 5 right panels (post-encoding) to be roughly twice the ripple rates in the left panels (stimulus encoding). This is further exemplified by Supp Fig 4 which shows (although not statistically) ripple rates for post-encoding are higher than for stimulus-encoding. This is not reflected in Supp Fig 5, which is concerning.

5) I do not understand the shaded areas in Fig 3d. The authors report that they find the t-values of the comparison between lure and arousal conditions for hippocampus and amygdala. These t-values (over the -250 – 250 msec time around the ripple) are calculated across participants. Therefore, there should only be one value presented in Fig 3d for each time period and no shaded area (as it is already across participants). To show the double dissociation in Fig 3d, the authors should have found the effect (difference between the two lure/arousal conditions) for hippocampus and amygdala for each participant, and compared the difference between those effects across participants with cluster-based permutation analysis.

6) I'm concerned about Fig 3b which shows the similarity between the activity patterns in amygdala and hippocampus around the time of the real ripples and the jittered pseudo-ripples. Firstly, the figure does not have a labelled y-axis. Secondly, it is unclear how exactly the null-distribution using jittered ripples is made. I expect it is generated by finding new jittered ripple-times for the ripples for each participant and then calculating a new mean and 95% confidence interval across participants for amygdala and hippocampus. However, the authors do not clarify this in sufficient detail.

7) To stress the importance of Fig 3b, the results relating to Fig 3a, Fig 3c, and Fig 3d are not supported without it. Without clear methodological descriptions and statistical evidence showing similarity increases selectively around ripple events, the authors cannot claim there are significant increases in similarity around ripples (Fig 3a). Nor can they claim there are differences in similarity around ripples related to arousal (Fig 3c top), or related to memory (Fig 3c bottom), or that there is a regional double dissociation (Fig 3d).

8) Fig 3a top and Fig 3c top-right are not consistent. Both figures consist of the same ripple similarity time course, with Fig 3a top containing high and low arousal trials and Fig 3c top-right splitting by arousal condition. In Fig 3a top, there is a dip in similarity around 0 msec from ripple. In Fig 3c top-right, there is an increase in similarity around 0 msec from ripple. The authors need to explain why this dip in Fig 3a top becomes a peak in similarity in Fig 3c top-right.

9) The authors claim they Z-scored the similarity in Fig 3c, using the mean and standard deviation of a null-distribution derived from jittered ripples (line 1201), however, they provide no reason for doing so. Why not use the raw similarity values from Fig 3a, split by arousal/lure condition.

10) In Supp Fig 11, the cluster is incorrectly described as being at ~500-750 ms after encoding. The figure shows it to be at ~1.1 – 1.3 s after encoding. This incorrect description of the results is repeated multiple times in the text.

11) The authors claim two effects are sequential in time (line 34: "Ripple-locked stimulus similarity appeared earlier in the amygdala than in hippocampus..."). Comparing the times of the significant clusters is not sufficient to make this claim. The authors need to show evidence that the effects are significantly different in time (e.g., statistically comparing the peak times of the effects).

Reviewer #3 (Remarks to the Author: Overall significance):

The authors adequately responded to my concerns; i have no further comments on this manuscript and consider it ready for publication.

Reviewer #3 (Remarks to the Author: Impact):

see previous comments.

Reviewer #3 (Remarks to the Author: Strength of the claims):

see previous comments.

Reviewer #3 (Remarks to the Author: Reproducibility):

see previous comments.

Author rebuttal, second version:

We are pleased to learn that Reviewers #1 and #3 have accepted our responses. We also thank the editor for giving us the opportunity to address the remaining comments from Reviewer #2. The responses to Reviewer's comments are shown in blue font, while the changes in manuscript are shown in blue italic.

Reviewer #2 (Remarks to the Author: Overall significance):

The rebuttal from Zhang et al. has not addressed all previous concerns. Additionally, their analyses have raised new concerns. There are multiple results which are inconsistent and analyses which are not presented clearly. Although the topic of the manuscript is interesting and timely, I am not convinced that the results support the authors' claims.

Response: We apologize that the Reviewer's concerns were not entirely addressed in the previous rebuttal. Below we present the additional analysis related to Reviewer's concerns and clarify likely sources of confusion. This includes entirely replacing the ripple occurrence (or ripple count)-based analysis with ripple rates, clarifying some methodological details, and improving the figure readability.

Reviewer #2 (Remarks to the Author: Strength of the claims):

1) In Supp Fig 4, the authors used ripple number and not ripple rate. This is a consistent issue throughout the manuscript (e.g., Fig 2c, Supp Fig 3, Supp Fig 6). All reported differences in ripple occurrence between conditions or time windows need to control for differences in response time. The reported differences in ripple number may have been the result of minor differences in response time (as the effects are small). Response time can be controlled for using ripple rate or by using response time as a control variable in a regression model.

Response: The rationale for using ripple occurrence (events/trial epoch) was based on the notion that individual ripples are involved in memory encoding/consolidation, as suggested by memory-impairing effects of ripple suppression (Girardeau et al., 2009; Ego-Stengel et al., 2009) and memory-enhancing effects of ripple prolongation (Fernandez-Ruiz et al., 2019). Therefore, we hypothesized that the larger number of post-encoding ripples would result in a stronger subsequent memory effect. Nevertheless, to address the Reviewer's concern about the confounding effect of conditional differences in post-encoding epoch duration, we entirely removed the ripple occurrence-based analysis from the manuscript, replacing it with the ripple rate-based analysis (events/sec; Fig. 2c, Supplementary Fig. 3, 4 and 6). In addition, Supplementary Fig. 5 and 7 were already presenting ripple rate-based analysis in the previous version.

Overall, ripple rate-based analysis confirms the association between the post-encoding ripples and stimulus-induced arousal/late correct Lure discrimination. Specifically, the post-encoding ripple rate is significantly higher following the high-arousal, compared to low-arousal Lure stimuli ($p = 0.046$, $z = -1.99$; Wilcoxon signed-rank test; Fig. 2c). The post-encoding ripple rate is also higher for correctly discriminated Lure stimuli ($p = 0.028$, $z = -2.20$; Wilcoxon signed-rank test; Fig. 2c). In addition, the post-encoding ripple rate was not associated with the stimulus valence ($p = 0.483$, $F(2, 15) = 0.76$; one-way ANOVA; Supplementary Fig. 3). Since the stimulus arousal is predictable of later

correct Lure discrimination, we disentangled the relations between those individual variables and post-encoding ripple rate by performing a two-way ANOVA (Supplementary Fig. 4). This analysis revealed the main effects of stimulus-induced arousal ($p = 0.038$; $F(1,20) = 4.93$) and later correct Lure discrimination ($p = 0.009$, $F(1,20) = 8.32$), with no significant interaction ($p = 0.619$, $F(1,20) = 0.26$). These effects are specific to the post-encoding window, as there is no significant association between the ripple rate and stimulus-induced arousal or later memory discrimination during encoding or retrieval ($p > 0.05$, Wilcoxon signed-rank test; Fig. 2c, Supplementary Fig. 6). To summarize, the ripple rate-based analysis confirms the association between post-encoding ripples and stimulus-induced arousal/later correct Lure discrimination, alleviating the concern about the possible confound induced by conditional differences in post-encoding epoch duration.

Results

Page 5, Lines 147-150

We tested the association of post-encoding ripple *rate* (the number of ripple *events/second*) with the stimulus emotional content (stimulus-induced arousal and valence) and correct Lure discrimination during retrieval.

Page 5, Lines 155-170

Higher post-encoding ripple *rate* was associated with stimulus-induced arousal ($p = 0.046$, $z = -1.99$; Wilcoxon signed-rank test, Fig. 2c) and also predicted correct Lure discrimination during retrieval ($p = 0.028$, $z = -2.20$, Wilcoxon signed-rank test, Fig. 2c), but was not associated with stimulus valence ($p = 0.187$, $F(2, 15) = 1.88$; one-way ANOVA; Supplementary Fig. 3). As the stimulus arousal and correct Lure discrimination are *correlated* (Fig. 1c), *while* both *being* associated with post-encoding ripple *rate* (Fig. 2c), we tested if the association between the ripple *rate* and correct Lure discrimination is modulated by the stimulus arousal level. *This analysis revealed the main effects of stimulus-induced arousal ($p = 0.038$; $F(1,20) = 4.93$) and later correct Lure discrimination ($p = 0.009$, $F(1,20) = 8.32$), with no significant interaction ($p = 0.619$, $F(1,20) = 0.26$, two-way ANOVA; Supplementary Fig. 4). This result suggests that stimulus-induced arousal and later correct Lure discrimination have an independent association with post-encoding ripples.* In addition, we tested if the post-encoding ripple association with stimulus arousal/correct Lure discrimination is limited to specific periods during post-encoding epoch by performing the conditional comparisons of *time-resolved* ripple rates (number of ripples/sec).

Page 6, Lines 186-191

Therefore, the associations between stimulus-induced arousal or correct Lure discrimination and post-encoding ripple *rates* were unrelated to post-encoding duration.

Associations between ripple *rate* and stimulus-induced arousal/later correct *Lure* discrimination accuracy were selective for the post-encoding time window. These relationships were absent for the stimulus encoding or the retrieval task stage ($p > 0.05$, Wilcoxon signed-rank test; Fig. 2c, Supplementary Fig. 6).

Discussion

Page 10, Lines 307-310

Our study reveals an association of higher ripple *rate* with stimulus-induced arousal and subsequent correct stimulus discrimination, providing direct evidence for ripple-mediated strengthening of emotional memory. Interestingly, the *increase in ripples* has been shown in rodents after exposure to a novel or reward-associated context²⁵.

Fig. 2. The post-encoding ripple *rate* predicts the stimulus-induced arousal and memory discrimination. **c.** The ripple *rate* (events/sec) is significantly higher following encoding of arousing (top right; $*p = 0.046$, $z = -2.0$, Wilcoxon signed-rank test) and later correctly discriminated stimuli (bottom right, $*p = 0.028$, $z = -2.2$, Wilcoxon signed-rank test). The ripple *rate* was showing no conditional differences during stimulus encoding (left column, p 's > 0.05 , Benjamini-Hochberg correction for multiple comparisons⁴⁷).

a
Supplementary Fig. 3. Stimulus valence is not significantly associated with ripple *rate* during the encoding or post-encoding epochs. Stimulus encoding phase: $F(2, 15) = 2.44$, $p = 0.121$; Post-encoding: $F(2, 15) = 1.88$, $p = 0.187$, One-way ANOVA). The data from individual subjects are color-coded.

a
Supplementary Fig. 4. *Stimulus-induced arousal and later correct Lure discrimination*

independently associated with the post-encoding ripple rate. The post-encoding ripple rate was significantly associated with both stimulus-induced arousal ($p=0.038$, $F(1,20) = 4.93$) and later correct Lure discrimination ($p = 0.009$, $F(1,20) = 8.32$), without significant interaction ($p = 0.619$, $F(1,20) = 0.26$; two-way ANOVA).

Supplementary Fig. 6. The ripple rate during retrieval task stage is not associated with stimulus-induced arousal, valence or correct Lure discrimination. Arousal: Stimulus presentation (top row), $p = 0.116$, $z = 1.57$; Response (bottom row), $p = 0.249$, $z = 1.15$, Wilcoxon signed-rank test. **Valence:** Stimulus presentation, $p = 0.298$, $F(2, 15) = 1.32$; Response, $p = 0.220$, $F(2, 15) = 1.68$, One-way ANOVA). **Correct Lure discrimination:** Stimulus presentation: $p = 0.600$, $z = -0.52$; Response: $p = 0.345$, $z = 0.94$, Wilcoxon signed-rank test).

Methods

Statistics

Pages 44-45, Lines 946-957

The association between post-encoding ripples and stimulus arousal/correct Lure discrimination was tested using *ripple rates (number of ripples/sec, Fig. 2; Supplementary Fig. 4, 5, 6 and 7)*. Except for the Wilcoxon signed-rank test analysis shown in Fig. 2c, ripple rates were normalized at individual subject level using z-score. Conditional comparisons of ripple rates (correct/incorrect Lure discrimination or high/low arousal; Fig. 2c) were done using the Wilcoxon signed rank test ($p < 0.05$). Associations between the stimulus-induced arousal/late correct Lure discrimination and ripple rate were analyzed using two-way ANOVA (*anovan.m* function in Matlab, $p < 0.05$; Supplementary Fig. 4).

The epoch-dependence of ripple association with correct Lure discrimination was tested using the *two-way ANOVA* ($p < 0.05$; Supplementary Fig. 7). *Post-hoc tests were done using the multcompare.m function in Matlab.*

2) The ripple counts in Supp Fig 4 do not align with the ripple counts in Fig 2c bottom-right. Supp Fig 4 shows ripple count ~ 0.3 while Fig 2c bottom-right shows ripple count of ~ 0.55 . Since these figures consist of the same ripples in the same time windows and conditions, the ripple counts should be similar.

Response: We apologize for the insufficient method description. The analysis shown in Fig. 2c was based on the raw ripple counts and the conditional comparisons were performed using the Wilcoxon test, a non-parametric test for paired data. On the other hand, the analysis in Supplementary Fig. 4 was based on Friedman's two-way ANOVA, which is a non-paired test that could be confounded by inter-subject variability, justifying the normalization of ripple counts within-subject. In the revised version, we clarified the usage of normalized ripple rates both in the Methods section and the y-axis label of Supplementary Fig. 4 (shown below). Also, based on Reviewer's comment #1, ripple count-based analysis in Supplementary Fig. 4 was replaced by ripple rate-based analysis.

Methods

Page 44, Lines 948-950

Except for the Wilcoxon signed-rank test analysis shown in Fig. 2c, ripple rates were normalized at individual subject level using z-score.

Supplementary Fig. 4. Stimulus-induced arousal and later correct Lure discrimination independently associated with the post-encoding ripple rate. The post-encoding ripple rate was significantly associated with both stimulus-induced arousal ($p=0.038$, $F(1,20) = 4.93$) and later correct

Lure discrimination ($p = 0.009$, $F(1,20) = 8.32$), without significant interaction ($p = 0.619$, $F(1,20) = 0.26$; two-way ANOVA).

3) If the authors want to claim that arousal and lure-discrimination both have main effects, they need to show a statistically significant effects of arousal for high and low lure-discrimination (not shown in Supp Fig 4). Further, Supp Fig 4 suggests there may be an interaction effect between lure-discrimination and arousal. If this is the case, it requires a different interpretation than two main effects.

Response: We thank the Reviewer for this suggestion. To address this issue, we had to apply a statistical method that computes both main effects and interaction. Therefore, we applied the two-way ANOVA, instead of previously used Friedman's two-way ANOVA. This analysis has shown the significant main effects of both stimulus-induced arousal ($p = 0.038$, $F(1,20) = 4.93$) and later correct Lure discrimination ($p = 0.009$, $F(1,20) = 8.32$), but no significant interaction ($p = 0.619$, $F(1,20) = 0.26$; two-way ANOVA). Therefore, the results suggest that both stimulus-induced arousal and later correct Lure discrimination have an independent association with post-encoding ripples.

For consistency, we also applied two-way ANOVA for the analysis shown in Supplementary Fig. 7 (shown below), testing the epoch-selectivity of the association between the ripple rate and later correct Lure discrimination. This analysis revealed significant main effects of task epoch ($p < 0.001$, $F(2, 30) = 103.91$) and correct Lure discrimination ($p = 0.004$, $F(1, 30) = 9.67$) and significant epoch x discrimination interaction ($p < 0.001$, $F(2, 30) = 10.97$; two-way ANOVA; Supplementary Fig. 7). Post-hoc comparisons have shown the significantly higher ripple rates during post-encoding epoch for the correctly discriminated Lure stimuli (post-encoding: $p < 0.001$, $M = -1.70$, 95% CI = [-1.08, -0.47]), while there were no significant conditional differences during the encoding or retrieval epochs (p 's > 0.05 ; post-hoc tests performed using `multcompare.m` function in Matlab). To summarize, the analysis reveals that association between the correct Lure discrimination and ripple rate is epoch-specific, as it selectively occurs during post-encoding.

Results

Page 5, Lines 159-167

As the stimulus arousal and correct Lure discrimination are *correlated* (Fig. 1c), while both *being* associated with post-encoding ripple *rate* (Fig. 2c), we tested if the association between the ripple *rate* and correct Lure discrimination is modulated by the stimulus arousal level. *This analysis revealed the main effects of stimulus-induced arousal ($p = 0.038$; $F(1,20) = 4.93$) and later correct Lure discrimination ($p = 0.009$, $F(1,20) = 8.32$), with no significant interaction ($p = 0.619$, $F(1,20) = 0.26$, two-way ANOVA; Supplementary Fig. 4). This result suggests that stimulus-induced arousal and later correct Lure discrimination have an independent association with post-encoding ripples.*

Page 6, Lines 191-202

Two-way ANOVA was used to test if the association between the ripple rate and correct Lure

discrimination is task epoch-dependent. The analysis shows significant main effects of task epoch ($p < 0.001$, $F(2, 30) = 103.91$) and correct Lure discrimination ($p = 0.004$, $F(1, 30) = 9.67$). In addition, we observed significant epoch x discrimination interaction (Supplementary Fig. 7; $p = 0.0003$, $F(2, 30) = 10.97$, two-way ANOVA). Post-hoc comparisons revealed the significantly higher ripple rates during post-encoding epoch for the correctly discriminated Lure stimuli (post-encoding: $p < 0.001$, $M = -1.70$, 95% CI = [-1.08, -0.47]), with no significant conditional differences during the encoding or retrieval epochs (p 's > 0.05 ; `multcompare.m` function in Matlab). To summarize, the analysis shows that the correct Lure discrimination is selectively associated with the ripple rate during post-encoding, but not during encoding or retrieval epochs.

Supplementary Fig. 4. **Stimulus-induced arousal and later correct Lure discrimination independently associated with the post-encoding ripple rate.** The post-encoding ripple rate was significantly associated with both stimulus-induced arousal ($p=0.038$, $F(1,20) = 4.93$) and later correct Lure discrimination ($p = 0.009$, $F(1,20) = 8.32$), without significant interaction ($p = 0.619$, $F(1,20) = 0.26$; two-way ANOVA).

Supplementary Fig. 7. The association between ripples and correct Lure discrimination is selective for post-encoding epoch. Only the post-encoding ripple rates are associated with correct Lure discrimination (significant epoch x Lure discrimination interaction effect, $p = 0.0003$, $F(2,30) = 10.97$, two-way ANOVA). Post-hoc analysis has shown the significantly higher ripple rates during post-encoding epoch for the correctly discriminated Lure stimuli (post-encoding: $p < 0.001$, $M = -1.70$ with 95% CI = [-1.08, -0.47]), while there was no significant difference during the encoding or retrieval epochs (p 's > 0.05 ; post-hoc tests performed using `multcompare.m` function in Matlab). * $p < 0.001$

Methods

Pages 44-45, Lines 952-957

Associations between the stimulus-induced arousal/ later correct Lure discrimination and ripple rate were analyzed using two-way ANOVA (anovan.m function in Matlab, $p < 0.05$; Supplementary Fig. 4 and 6). The epoch-dependence of ripple association with correct Lure discrimination was tested using the two-way ANOVA ($p < 0.05$; Supplementary Fig. 7). Post-hoc tests were done using the `multcompare.m` function in Matlab.

4) In Fig 2c, the ripple counts are similar for the stimulus encoding and the post-encoding phases. However, the average time of the post-encoding phase (~700 msec) is less than half the time of the stimulus encoding phase (2000 msec). Therefore, one expects the ripple rates in Supp Fig 5 right panels (post-encoding) to be roughly twice the ripple rates in the left panels (stimulus encoding). This is further exemplified by Supp Fig 4 which shows (although not statistically) ripple rates for post-encoding are higher than for stimulus-encoding. This is not reflected in Supp Fig 5, which is concerning.

Response: The Supplementary Fig. 5 shows average ripple rates (events/sec) for the post-encoding windows across the entire distribution of post-encoding window lengths up to 1500 msec. Ripples tend to occur close to response time (~80% occur within 600 msec prior to response time, as shown in Rebuttal Fig. 1). This is also reflected by an almost two-fold increased ripple rate during the last 600 msec of the post-encoding window (Supplementary Fig. 5). The tendency of ripples to occur close to response time also accounts for a relatively low ripple rate on the left side of Supplementary Fig. 5, which reflects the early parts (further away from response time) of the longer post-encoding windows. Importantly, ripple rate collapsed across the entire post-encoding window (Fig. 2c) is disproportionately affected by the ripple rates from shorter post-encoding epochs, as ~60% of post-encoding windows are <600 msec long (Rebuttal Fig. 2). To summarize, the majority of response times (and consequently, the post-encoding windows) are short, so the parts of post-encoding window in Supplementary fig. 5 that show relatively lower ripple rates (-1.5 - 0.5 sec prior to response time) are contributed to by a small number of trials. Therefore, collapsing across all the trials (regardless of the post-encoding window length), the average post-encoding ripple rate is higher than during encoding (as shown, for example, in Fig. 2 or Supplementary Fig. 3).

Rebuttal Fig. 1. Distribution of ripple peak times, relative to post encoding response time. Ripples are clustered close to response time (~80% occur within 600 msec prior to response time).

Rebuttal Fig. 2. Response time distribution, corresponding to the duration of post-encoding windows, is skewed towards the shorter response time, reflecting the ~60% of response times (and consequently, post-encoding window lengths) being <600 msec.

Methods

Page 49, Lines 1107-1114

Time-resolved ripple rates

The time-resolved ripple rates (Supplementary Fig. 5) were calculated for individual epochs (encoding, post-encoding) and conditions (low and high stimulus-induced arousal, correct and incorrect Lure discrimination) using 1 msec time bins, smoothed with a Gaussian kernel ($\sigma = 150$ msec) and averaged across trials. The time-resolved ripple rate was compared based on the low vs. high stimulus-induced arousal and later correct vs. incorrect Lure discrimination contrasts, using non-parametric cluster-based permutation test ($p < 0.05$).

5) I do not understand the shaded areas in Fig 3d. The authors report that they find the t-values of the comparison between lure and arousal conditions for hippocampus and amygdala. These t-values

(over the -250 – 250 msec time around the ripple) are calculated across participants. Therefore, there should only be one value presented in Fig 3d for each time period and no shaded area (as it is already across participants).

Response: We apologize for the insufficient explanation of the procedure used for computing the data points shown in Fig. 3d. First, the subject-level t-values were calculated by comparing the stimulus similarity between the low- and high-arousal trials within-subject, reflecting the magnitude and direction of the arousal effect on representational similarity. The same within-subject calculations were made for the correctly or incorrectly discriminated Lure stimuli. The data points shown in Fig. 3d represent the mean \pm SEM (line and shaded area) of individual subject t-values at a given time point during a ripple-locked window. We have added this clarification in the Methods section and figure caption of the revised version.

Methods

Page 52, Lines 1208-1214

The association strength between the post-encoding ripple-locked stimulus similarity and a) stimulus arousal or b) later correct Lure discrimination was compared between the amygdala and hippocampus. First, the regional t-values were computed, based on *within-subject* comparison between the post-encoding ripple-locked stimulus similarity for low- and high-arousal Lure stimuli or for the correctly or incorrectly discriminated Lure stimuli. The regional t-value time courses were then compared using the non-parametric cluster-based permutation test (1000 permutations, $p < 0.05$).

Fig 3.d, Double-dissociation between the post-encoding ripple-locked stimulus representation in hippocampus and amygdala. Left: The association between the stimulus arousal and post-encoding ripple-locked stimulus similarity was stronger in the amygdala (-70 to 20 msec relative to ripple peak, $p < 0.001$, non-parametric cluster-based permutation test). Right: The association between the later correct Lure discrimination and post-encoding ripple-locked stimulus similarity was stronger in the hippocampus (-60 to 10 msec relative to ripple peak, $p = 0.046$, non-parametric cluster-based permutation test). The line and shaded areas represent the mean \pm SEM of the individual subject t-values, respectively.

To show the double dissociation in Fig 3d, the authors should have found the effect (difference between the two lure/arousal conditions) for hippocampus and amygdala for each participant, and compared the difference between those effects across participants with cluster-based permutation analysis.

Response: We performed additional testing of regional double-dissociation of stimulus-induced arousal and later correct Lure discrimination, based on the complementary approach suggested by the Reviewer (listed above). Specifically, we obtained the subject- and condition-specific t-value time series, by comparing the low vs. high stimulus-induced arousal trials and correct vs. incorrect Lure discrimination trials, using the non-parametric cluster-based permutation, separately for amygdala and hippocampus. This analysis revealed that the association between the stimulus-induced arousal and ripple-locked similarity is significantly stronger than the association between the later correct Lure discrimination and ripple-locked similarity in the amygdala (-42 msec to 0 msec relative to ripple peak, $p = 0.034$). On the other hand, the association between the later correct Lure discrimination and post-encoding ripple-locked stimulus similarity was stronger in the hippocampus (-83 msec to 10 msec relative to ripple peak, $p = 0.047$, non-parametric cluster-based permutation test; Supplementary Fig. 9). Therefore, the regional double-dissociation between the stimulus-induced arousal and later correct Lure discrimination in the amygdala and hippocampus was demonstrated using two complementary approaches.

Results

Page 8, Lines 247-258

In addition, the regional double-dissociation of stimulus-induced arousal and later correct Lure discrimination was tested by comparing the low vs. high stimulus-induced arousal trials and correct vs. incorrect Lure discrimination trials, separately for amygdala and hippocampus. This analysis shows that the association between the stimulus-induced arousal and ripple-locked similarity is significantly stronger than the association between the later correct Lure discrimination and ripple-locked similarity in the amygdala (-42 msec to 0 msec relative to ripple peak, $p = 0.034$). The opposite pattern was present in the hippocampus, where the association between the later correct Lure discrimination and post-encoding ripple-locked stimulus similarity was significantly stronger (-83 msec to 10 msec relative to ripple peak, $p = 0.047$, non-parametric cluster-based permutation test; Supplementary Fig. 9).

Supplementary Fig. 9. Double-dissociation between the post-encoding ripple-locked stimulus similarity in hippocampus and amygdala. Left: The association between the stimulus arousal and post-encoding ripple-locked stimulus similarity was stronger in the amygdala (-42 msec to 0 msec relative to ripple peak, $p = 0.034$, non-parametric cluster-based permutation test). Right: The association between the later correct Lure discrimination and post-encoding ripple-locked stimulus similarity was stronger in the hippocampus (-83 msec to 10 msec relative to ripple peak, $p = 0.047$, non-parametric cluster-based permutation test). The line and shaded areas represent the mean \pm SEM of the individual subject t-values, respectively.

Methods

Page 54, Lines 1259-1265

In addition, regional double-dissociation was tested by computing the region-specific (amygdala and hippocampus) t-value time series, obtained by comparing the low vs. high stimulus-induced arousal trials and correct vs. incorrect Lure discrimination trials within-subject. Next the condition-specific t-values were compared separately for each region, between the stimulus-induced arousal and later correct Lure discrimination, using the non-parametric cluster-based permutation ($p < 0.05$; Supplementary Fig. 9).

6) I'm concerned about Fig 3b which shows the similarity between the activity patterns in amygdala and hippocampus around the time of the real ripples and the jittered pseudo-ripples. Firstly, the figure does not have a labelled y-axis.

Secondly, it is unclear how exactly the null-distribution using jittered ripples is made. I expect it is generated by finding new jittered ripple-times for the ripples for each participant and then calculating a new mean and 95% confidence interval across participants for amygdala and hippocampus. However, the authors do not clarify this in sufficient detail.

Response: We apologize for this omission. The y-axis has been labeled in the revised version of the manuscript (also shown below). Regarding the ripple jittering procedure, the null-distribution was created by circularly jittering the ripple timestamps within a ± 500 msec window, relative to ripple peak. Next, the stimulus similarity trace around jittered timestamps was averaged within subjects

and the grand average was calculated across subjects. The procedure was repeated for 1000 times, resulting in an empirical null-distribution of stimulus similarity. Regional similarity trace windows exceeding 95th percentile of null-distribution were considered as periods of statistically significant ripple-locked stimulus similarity. The shaded areas in Fig. 3b represent 95th percentile of null-distribution (as noted in the Fig. 3b caption: 'Shaded areas denote the null-distribution 95% confidence interval.'). The more elaborated description of this procedure is provided in the Methods section of the revised version.

Fig. 3. Post-encoding stimulus similarity in the hippocampus and amygdala around ripple. b, Post-encoding stimulus similarity is greatest around the time of ripples as shown by comparison with the null-distribution (within ± 250 msec). Shaded areas denote the null-distribution 95% confidence interval. Similarity in the hippocampus overlaps with ripple peak (orange), while similarity in the amygdala peaks prior to and after the ripples (magenta).

Methods

Page 53, Lines 1228-1240

Specifically, we circularly randomly jittered the ripple peak times within ± 500 msec window for 1000 times, obtaining an empirical null distribution of stimulus similarity. Circular jittering denotes the method of event time shuffling within a limited time window. In the context of the present study, the time window is defined by the onset of post-encoding epoch and the offset of subsequent cross-fixation (Fig. 1a). For example, if the ripple occurred 200 msec after the post-encoding onset and the randomly generated shuffled distance is -500 msec, the assigned shuffled ripple timestamp would be -300 msec prior to offset of the cross-fixation epoch. *Next, the stimulus similarity trace around jittered timestamps was averaged within subjects and the grand average was calculated across subjects. The procedure was repeated for 1000 times, resulting in an empirical null-distribution of stimulus similarity. Regional similarity trace windows exceeding 95th percentile of null-distribution were considered windows of statistically significant ripple-locked stimulus similarity.*

7) To stress the importance of Fig 3b, the results relating to Fig 3a, Fig 3c, and Fig 3d are not supported without it. Without clear methodological descriptions and statistical evidence showing similarity increases selectively around ripple events, the authors cannot claim there are significant increases in similarity around ripples (Fig 3a). Nor can they claim there are differences in similarity around ripples related to arousal (Fig 3c top), or related to memory (Fig 3c bottom), or that there is a regional double dissociation (Fig 3d).

Response: We agree about the importance of Fig. 3b, as this analysis is the foundation for further analysis in the manuscript. Therefore, we further elaborate the analysis presented in the figure, including the approach for generating the ripple null-distribution (see the answer to #6).

8) Fig 3a top and Fig 3c top-right are not consistent. Both figures consist of the same ripple similarity time course, with Fig 3a top containing high and low arousal trials and Fig 3c top-right splitting by arousal condition. In Fig 3a top, there is a dip in similarity around 0 msec from ripple. In Fig 3c top-right, there is an increase in similarity around 0 msec from ripple. The authors need to explain why this dip in Fig 3a top becomes a peak in similarity in Fig 3c top-right.

Response: Figure 3a (top) shows the ripple-locked similarity in **amygdala**, while Figure 3c (top-right) shows the ripple-locked similarity in **hippocampus**. Therefore, these two time courses are expected to be different. In addition to regional labels present in the original figure axis labels and captions, we included additional labels in the revised version, to maximize the figure readability.

Fig. 3. Post-encoding stimulus similarity in the hippocampus and amygdala around ripple. **a**, Ripple-locked similarity in the amygdala (top) and hippocampus (bottom) during the post-encoding period (line and shaded areas represent the mean \pm SEM). **b**, Post-encoding stimulus similarity is greatest around the time of ripples as shown by comparison with the null-distribution (within ± 250 msec). Shaded areas denote the null-distribution 95% confidence interval. Similarity in the hippocampus overlaps with ripple peak (orange), while similarity in the amygdala peaks prior to and after the ripples (magenta). **c**, Ripple-locked post-encoding stimulus similarity in the amygdala is significantly higher for arousing stimuli (top left, $p = 0.035$, see Methods) but is not associated with subsequent discrimination (bottom left, $p = 0.066$). Ripple-locked post-encoding stimulus similarity in the hippocampus is significantly higher for correctly discriminated Lure stimuli (bottom right, $p =$

0.008, see Methods) but does not depend on stimulus-induced arousal (top right, $p > 0.1$). **d**, Double-dissociation between the post-encoding ripple-locked stimulus representation in hippocampus and amygdala. Left: The association between the stimulus arousal and post-encoding ripple-locked stimulus similarity was stronger in the amygdala (-70 to 20 msec relative to ripple peak, $p < 0.001$, non-parametric cluster-based permutation test). Right: The association between the later correct Lure discrimination and post-encoding ripple-locked stimulus similarity was stronger in the hippocampus (-60 to 10 msec relative to ripple peak, $p = 0.046$, non-parametric cluster-based permutation test).

9) The authors claim they Z-scored the similarity in Fig 3c, using the mean and standard deviation of a null-distribution derived from jittered ripples (line 1201), however, they provide no reason for doing so. Why not use the raw similarity values from Fig 3a, split by arousal/lure condition.

Response: We apologize for the insufficient justification of the ripple-locked similarity normalization approach used for the analysis presented in Fig. 3c. We hypothesized that stimulus similarity increases during ripples and that this increase is different between the conditions (low vs. high arousal and correct vs. incorrect memory discrimination). As the baseline levels of similarity could differ both between-subjects and across individual trials within-subject, the purpose of normalization is to control for this potential confound. In this case, we applied trial-level normalization, providing the most sensitive index of ripple-locked change in stimulus similarity. Normalization of ripple-locked activity prior to statistical comparisons is commonly used when examining ripple-locked neural activity, as illustrated by several examples listed below.

1. "...the majority (22 of 29) of neurons showed a small, but significant, decrease in activity coinciding with ripple events (**peri-event histogram trough z score** < -3.28 and $P < 0.001$, Wilcoxon signed-rank test; Fig. 1c,d)." Wang et al., 2015. Mesopontine median raphe regulates hippocampal ripple oscillation and memory consolidation. *Nature Neuroscience*, 18(5), pp.728-735.
2. "This procedure was repeated for each surrogate distribution, and **the true mean ripple-related power response was z-scored relative to the 1000 mean surrogate responses.**" Cox et al., 2020. Sharp wave-ripples in human amygdala and their coordination with hippocampus during NREM sleep. *Cerebral Cortex Communications*, 1(1)
3. "We quantified these differences by comparing the session-averaged **SPW-R modulation scores (mean z-scored - 50 ms to 150 ms around ripple peak)** for each hippocampal site and each area, yielding significant differences between septal and posterior sites (Fig. 6C)." Nitzan et al., 2022. Brain-wide interactions during hippocampal sharp wave ripples. *Proceedings of the National Academy of Sciences*, 119(20).

10) In Supp Fig 11, the cluster is incorrectly described as being at ~500-750 ms after encoding. The figure shows it to be at ~1.1 – 1.3 s after encoding. This incorrect description of the results is

repeated multiple times in the text.

Response: This impression is caused by the flipped y-axis in the original version, causing the timing reported in the text to appear incorrect, as noted by the Reviewer. We confirm that the cluster timing as reported in the text is correct and we apologize for the figure error, which has been corrected. For the comparison, we show both the flipped and correct version of the figure below.

Below is the old version of the figure, with flipped y-axis.

This version has been replaced by the new version (below), with corrected y-axis. Please note that the numbers in the caption and text were correct and therefore do not change as the result of this correction. Please note that due to addition of Supplementary Figures in this version of revision, the former Supplementary Fig. 11 is now numbered as Supplementary Fig. 12.

Supplementary Fig. 12. The representational similarity map showing the stimulus specificity of the post-encoding ripple representation in the hippocampus. The temporal cluster of significant stimulus-specific similarity (-190 - 20 msec, relative to ripple peak and ~500-750 msec of encoding time) is encircled in black (non-parametric cluster-based permutation test; $n = 1000$ permutations, $p < 0.05$).

11) The authors claim two effects are sequential in time (line 34: “Ripple-locked stimulus similarity appeared earlier in the amygdala than in hippocampus...”). Comparing the times of the significant clusters is not sufficient to make this claim. The authors need to show evidence that the effects are significantly different in time (e.g., statistically comparing the peak times of the effects).

Response: Following Reviewer’s suggestion, we compared the timing of peak joint stimulus similarity in the hippocampus and amygdala. First, the peak similarity timings were computed during post-encoding ripple windows following later correctly discriminated stimuli. Next, peak timings were compared between the hippocampus and amygdala. The peak similarity in the amygdala occurred significantly earlier than in the hippocampus ($p = 0.006$, $t(5) = -3.89$; one-tail paired t-test). In addition, earlier amygdala peak similarity is present across all individual subjects (Supplementary Fig. 13a). Therefore, the earlier presence of ripple-locked stimulus similarity in the amygdala is demonstrated by several converging approaches, increasing the confidence in this result.

Results

Page 9, Lines 276-279

The peak ripple-locked stimulus similarity occurred significantly earlier in the amygdala, than in the hippocampus (difference: -18 ± 11 msec, mean \pm SEM; $p = 0.006$, $t(5) = -3.89$; one-tail paired t-test), with the timing difference being consistent across the subjects (Supplementary Fig. 13).

Supplementary Fig. 13. Ripple-locked similarity occurs in the amygdala, prior to hippocampus. a, Stimulus similarity during post-encoding ripple windows peaks earlier in the amygdala (red), relative to hippocampus (blue). The data points represent individual subject mean \pm SD. The earlier amygdala similarity peak is consistent across the individual subjects. **b,** The difference in ripple-locked peak similarity timing (amygdala - hippocampus). Negative values denote the earlier similarity peak in the amygdala (-18 ± 11 msec, mean \pm SEM; $p = 0.006$, $t(5) = -3.89$; one-tail paired t-test). ****** $p < 0.01$

Methods

Page 45, Lines 972-976

To compare the timing of ripple-locked stimulus similarity between the hippocampus and amygdala,

the peak similarity timings were computed during post-encoding ripple windows, following the encoding of later correctly discriminated stimuli. Next, the peak similarity timings were compared between the regions using the one-tail paired t-test ($p < 0.05$; Supplementary Fig. 13).

Reviewer comments, Third version:

Reviewer #2 (Remarks to the Author: Overall significance):

Zhang and colleagues have made significant improvements to the manuscript and have made a good effort to integrate previous feedback. All major concerns have been addressed by the corrections. There are a few minor points which are mentioned below which should still be addressed.

- a) Fig 2 is now missing annotations for panels a & b.
- b) Fig 3a is described as the similarity during the post-encoding period, but the axes state "Time during Stimulus Encoding". This is unclear, the axes should be corrected to say "Time after Stimulus Encoding".
- c) Fig 4b is described as having top and bottom panels when it actually has left and right panels.

Author rebuttal, Third version:

Reviewer #2 (Remarks to the Author: Overall significance):

Zhang and colleagues have made significant improvements to the manuscript and have made a good effort to integrate previous feedback. All major concerns have been addressed by the corrections. There are a few minor points which are mentioned below which should still be addressed.

Response: We are pleased to learn that Reviewer #2 has accepted our responses. We also thank the editor for giving us the opportunity to publish the revised manuscript. The responses to Reviewer's comments are shown in blue font, and the changes in the manuscript are presented accordingly.

a) Fig 2 is now missing annotations for panels a & b.

Response: We thank the Reviewer for this suggestion. These missing annotations have been added to the figure legend.

Fig. 2. The post-encoding ripple rate predicts the stimulus-induced arousal and memory discrimination. **a,** Reconstructed locations of hippocampal (blue) and amygdala electrodes (red). **b,** The ripple grand average waveform ($n = 4689$ ripples in 6 hippocampal channels, 6 participants). Line and shaded areas represent the mean \pm SEM. **c,** The ripple rate (events/sec) is significantly higher following encoding of arousing (top right; $*p = 0.046$, $z = -2.0$, $n=6$, Two-sided Wilcoxon signed-rank test) and later correctly discriminated stimuli (bottom right, $*p = 0.028$, $z = -2.2$, $n=6$, Two-sided Wilcoxon signed-rank test). The ripple rate was showing no conditional differences during stimulus encoding (left column, p 's > 0.05 , Benjamini-Hochberg correction for multiple comparisons⁴⁷). Box and bar indicate mean \pm SEM.

b) Fig 3a is described as the similarity during the post-encoding period, but the axes state “Time during Stimulus Encoding”. This is unclear, the axes should be corrected to say “Time after Stimulus Encoding”.

Response: We apologize for the insufficient figure description. The figure shows stimulus similarity during the post-encoding ripple window (-250 to 250 ms). The x-axis represents the time relative to the post-encoding ripple peak, while the y-axis represents the time during stimulus encoding. Thus, the color plot shows the stimulus similarity matrix for every time point pair between the stimulus encoding window and the post-encoding ripple window. We confirmed that the axis labels were correct while we elaborated the figure description.

Fig. 3. Post-encoding stimulus similarity in the hippocampus and amygdala around ripple. a, Ripple-locked similarity in the amygdala (top) and hippocampus (bottom) during the post-encoding period (line and shaded areas represent the mean \pm SEM). The x-axis represents the time relative to the post-encoding ripple peak and the y-axis represents the time during stimulus encoding.

c) Fig 4b is described as having top and bottom panels when it actually has left and right panels.

Response: We thank the Reviewer for this suggestion. The legend has been corrected.

Fig. 4. Synchronously increased ripple-locked post-encoding stimulus similarity in the hippocampus and amygdala predicts the correct Lure discrimination.

b, Mutual information (MI) difference for the amygdala (AMY) and hippocampal (HPC) stimulus similarity time-courses, during the post-encoding ripple windows (correct Lure discrimination - top, incorrect Lure discrimination - bottom). Positive values denote stronger AMY \rightarrow HPC directionality. A temporal cluster of significant MI difference (AMY \rightarrow HPC) is present before ripple peak time (-70 to -30 msec) after encoding of correctly discriminated Lure stimuli (top; $p = 0.038$, see Methods), indicating that hippocampal stimulus similarity is better predictable by amygdalastimulus similarity than vice versa. This effect is present only during the post-encoding period for correctly discriminated Lure stimuli (left), but not for the incorrectly discriminated Lure stimuli (right). The line and shaded areas represent the mean \pm SEM of the individual participant MI difference.